# A Unified Framework for the Transportability of Population-Level Causal Measures

**Ahmed Boughdiri**
Premedical Inria
Univ. Montpellier, France
ahmed.boughdiri@inria.fr

**Clément Berenfeld**
Premedical Inria
Univ. Montpellier, France
clement.berenfeld@inria.fr

**Julie Josse**
Premedical Inria
Univ. Montpellier, France
julie.josse@inria.fr

**Erwan Scornet**
Sorbonne Université and Université Paris Cité, CNRS
Laboratoire de Probabilités, Statistique et Modélisation
F-75005 Paris, France
erwan.scornet@sorbonne-universite.fr

## Abstract

Generalization methods offer a powerful solution to one of the key drawbacks of randomized controlled trials (RCTs): their limited representativeness. By enabling the transport of treatment effect estimates to target populations subject to distributional shifts, these methods are increasingly recognized as the future of meta-analysis, the current gold standard in evidence-based medicine. Yet most existing approaches focus on the risk difference, overlooking the diverse range of causal measures routinely reported in clinical research. Reporting multiple effect measures—both absolute (e.g., risk difference, number needed to treat) and relative (e.g., risk ratio, odds ratio)—is essential to ensure clinical relevance, policy utility, and interpretability across contexts. To address this gap, we propose a unified framework for transporting a broad class of first-moment population causal effect measures under covariate shift. We provide identification results for both continuous and binary outcome under two conditional exchangeability assumptions, derive both classical and semiparametric estimators, and evaluate their performance through theoretical analysis, simulations, and real-world applications. Our analysis shows the specificity of different causal measures and thus the interest of studying them all: for instance, two common approaches (one-step, estimating equation) lead to similar estimators for the risk difference but to two distinct estimators for the odds ratio.

## 1 Introduction

Generalization methods [33, 43, 12, 10, 14] have emerged as a powerful response to the restricted external validity [36] of RCTs: due to stringent inclusion and exclusion criteria, RCT populations often exclude key segments of the real-world patient population—such as individuals with comorbidities, pregnant women, or other vulnerable groups—resulting in trial samples that are poorly representative. Consequently, the findings of many RCTs may lack relevance for broader clinical or policy applications. Generalization techniques address this gap by exploiting treatment effect heterogeneity—that is, the fact that treatment efficacy can vary systematically with patient characteristics. By adjusting for

---

Code & Python library: `Causalipy`. `https://github.com/BoughdiriAhmed/Causalipy`.

39th Conference on Neural Information Processing Systems (NeurIPS 2025).

differences in the distribution of these covariates between the trial population and a target population, these methods can estimate treatment effects beyond the original study population. This is especially valuable given the high costs, long timelines, and operational complexity of conducting new trials. As such, generalization approaches are increasingly viewed as a pivotal step toward rethinking the role of clinical trials in modern evidence generation. The implications are far-reaching. For instance, when drug reimbursement decisions are partly tied to estimated real-world efficacy [17], the ability to predict treatment benefits across diverse populations could influence pricing, access, and healthcare policy. Moreover, recent works suggest that generalization methods may also redefine the role of meta-analysis [11, 37, 20], traditionally viewed as the top of the pyramid of evidence-based medicine.

Most generalization methods are dedicated to estimating the average treatment effect (ATE) via the risk difference (RD), owing to its linearity and analytical convenience. Yet this focus is incomplete. Clinical guidelines and regulatory bodies explicitly recommend reporting both absolute and relative effect measures [39, 31], as they capture complementary aspects of treatment impact. Among absolute measures, the RD remains standard, but the number needed to treat (NNT), a direct, clinically intuitive transformation of the RD, offers additional interpretability [29]. On the relative scale, measures such as the risk ratio (RR), widely used in public health research [28], and the odds ratio (OR), very popular in epidemiology [18], play a critical role in framing treatment efficacy. Presenting multiple causal estimands is not simply recommended—it is essential. A treatment effect that appears homogeneous on one scale may reveal heterogeneity on another, a phenomenon that may seem counterintuitive but carries significant implications for personalized decision-making. Furthermore, the perceived magnitude of benefit can shift dramatically depending on the baseline risk and the effect measure used. Consider a treatment that reduces mortality from 3% to 1%: the RD of 0.02 may suggest a minor effect, yet the RR reveals a threefold increase in risk for the untreated, reframing the impact as clinically substantial. This striking contrast illustrates how the choice of causal measure directly shapes interpretation and ultimately, policy and clinical decisions.

**Contributions.** In Section 2, we present a unified framework for transporting a broad class of *first-moment population causal effect measures*, defined as functionals of the expectations of potential outcomes. This class, composed of more than a dozen widely-used estimands, includes collapsible (RD, RR, etc.) and non-collapsible measures (odds ratio-OR, NNT etc.) the latter posing unique generalization challenges. Building on this formalism, we develop generic identification strategies under two key assumptions: (i) exchangeability in mean (Section 3), and (ii) exchangeability in effect measure (Section 4). The latter assumption, though weaker, requires access to control outcomes in the target population—a condition met, for instance, when treatment has not yet been deployed. Crucially, our identification results extend to non-collapsible measures even under this weaker condition, which to our knowledge has not been previously derived. Within each setting, we derive two broad families of estimators. The first approach adapts classical methods—such as weighting (Horvitz-Thompson) and regression-based strategies (G-formula)—for which we establish asymptotic properties and derive closed-form variance expressions. To the best of our knowledge, no prior work has formally studied these properties using density ratio estimation. The second leverages semiparametric efficiency theory to construct new doubly-robust estimators using either one-step or estimating equation approaches. While these often coincide in the linear case (e.g., RD), we highlight that for nonlinear measures such as the RR or OR, they diverge—leading to fundamentally different estimators. Finally, in Section 5 we conduct an extensive empirical evaluation of all estimators on synthetic and a real-world dataset.

**Related work.** Most generalization work focused on RD. While the idea of weighting randomized controlled trials (RCTs) is not new, using external data can be traced back to the foundational work of [5]. [14, 10] provided a comprehensive survey of generalization techniques, and [6] contributed additional consistency results for the main classes of estimators: regression-based, weighting-based, and hybrid approaches. [8] further investigated the inverse probability of sampling weighting (IPSW) estimator, deriving finite-sample bias and variance, as well as an upper bound on its risk under the assumption that the covariates $X$ are categorical. Other studies have addressed the generalization problem by modeling the outcome directly [27, 13]. [11] extended this line of research to settings involving multiple source datasets (i.e., several RCTs) and a set of covariates drawn from a distinct target distribution. They position this framework as a natural evolution of traditional meta-analysis, highlighting its potential to unify evidence across studies. Although the present work focuses on a single RCT and a single target population, all of our results seamlessly extend to the multi-RCT case. This generality allows us to offer a compelling alternative to classical meta-analyses—one

that supports the generalization of both absolute and relative treatment effect measures in a unified framework. We also highlight recent works on the analysis of externally controlled single-arm trials and hybrid trials [2, 30], which aim to enhance the precision of RCT-based estimands by incorporating external information. Similarly, [15] incorporate observational data to RCTs using the prediction-powered inference framework. While these approaches address distinct inferential questions, they share structural similarities with ours in using auxiliary data to improve estimation. Another line of work has focused on estimating alternative causal effect measures beyond the risk difference. For instance, [1] proposes several strategies for estimating the risk ratio (RR) in observational settings, without addressing the generalization to a target population. [7] examined several key properties (e.g., collapsibility) of different causal effect measures (e.g., RD, RR, and OR) to identify which estimands are less sensitive to distributional shifts. While their work emphasizes which causal estimands are easier to identify and generalize, ours provides a unified framework that generalizes all first-moment causal measures, focusing on estimation strategies and efficiency for practical deployment with theoretical guarantees. Their focus is on distinguishing treatment effects from baselines; ours highlights the shared structure among estimands, enabling unified identification and estimation. [34, 44, 40] introduce methods for estimating conditional risk ratios, including approaches based on causal forests. Yet, it is crucial to emphasize that due to the non-linearity of relative measures, estimating the CATE does not directly yield the ATE.

## 2 Problem setting: notation and identification assumptions

Following the potential outcomes framework [38, 41], we consider random variables $(X, S, A, Y^{(0)}, Y^{(1)})$, where $X \in \mathbb{R}^p$ denotes patient covariates, $S \in \{0, 1\}$ indicates sample membership ($S = 1$ for the source population and $S = 0$ for the target population), and $A \in \{0, 1\}$ denotes treatment assignment ($A = 1$ if treatment is administered and $A = 0$ otherwise). Potential outcomes $Y^{(0)}$ and $Y^{(1)}$ represent outcomes under control and treatment respectively, of which only one is observed per individual depending on the assigned treatment, yielding the observed outcome

$$Y = AY^{(1)} + (1 - A)Y^{(0)}.$$

We observe data from two distinct populations: a Randomized Controlled Trial (RCT) dataset $(X_i, A_i, Y_i)_{i \in [n]}$ from the source population, where treatment assignments and outcomes are recorded, and a target dataset $(X_i)_{i \in [m]}$, where only covariates are observed. This reflects the practical constraint that treatment and outcome data are collected only for individuals enrolled in the trial, while covariate information is available for both populations. We model this two datasets as stemming from one probability distribution $P_{\text{obs}}$ and we observe a $N$-sample:

$$(Z_i)_{i \in [N]} := (S_i, X_i, S_i A_i, S_i Y_i)_{i \in [N]} \sim P_{\text{obs}}^{\otimes N},$$

Let $\alpha$ be the Bernoulli parameter of the random variable $S$. Thus, $(n, m)$ follow a multinomial distribution with parameter $N$ and probability $(\alpha, 1 - \alpha)$. Let $P_{\text{S}}$ and $P_{\text{T}}$ denote the conditional distributions of $Z$ given $S = 1$ and $S = 0$, respectively, with corresponding expectations $\mathbb{E}_{\text{S}}[\cdot]$ and $\mathbb{E}_{\text{T}}[\cdot]$.

### 2.1 Causal estimands of interest

Causal effect measures, as formalized by Pearl [32], are expressed in terms of the joint distribution of potential outcomes. Individual-level causal effects rely on this joint distribution, which is generally not identifiable from observed data. Therefore, in this paper, we focus on a specific subclass: the first-moment population causal measures [16]. This class includes commonly employed estimands—such as RD, RR, and OR—and less frequently used quantities, including the Switch Relative Risk [22], Excess Risk Ratio (ERR) [4], Survival Ratio (SR), and Relative Susceptibility (RS), and Log Odds Ratio (log-OR), see Appendix A for a detailed enumeration of measures falling into this class.

**Definition 1** (First moment population causal measures)**.** *Let $P$ denote the joint distribution of potential outcomes $(Y^{(0)}, Y^{(1)})$. The quantity $\tau^P$ is a first moment population causal measure if there exists an* effect measure $\Phi : D_\Phi \to \mathbb{R}$, with $D_\Phi \subset \mathbb{R}^2$, *such that for all distributions $P$ with* $(\mathbb{E}_P[Y^{(0)}], \mathbb{E}_P[Y^{(1)}]) \in D_\Phi$,

$$\tau^P := \Phi\left(\mathbb{E}_P[Y^{(1)}], \mathbb{E}_P[Y^{(0)}]\right). \tag{1}$$

*We require that for all $\psi_0 \in \mathbb{R}$, the map $\psi_1 \mapsto \Phi(\psi_1, \psi_0)$ is injective on its definition domain. It's inverse, when it exists, is denoted by $\Gamma(\cdot, \psi_0)$ and is called the* effect function. *This definition extends to subpopulations by conditioning on covariates $X$, yielding, when it exists, the conditional effect*

$$\tau^P(X) := \Phi\left(\mathbb{E}_P[Y^{(1)}|X], \mathbb{E}_P[Y^{(0)}|X]\right). \tag{2}$$

*Example* 1. For these well-known causal measures, the effect measures and functions are given by:

| Measure | Effect Measure $\Phi$ | Effect Function $\Gamma$ |
|---|---|---|
| **Risk Difference (RD)** | $\Phi(\psi_1, \psi_0) = \psi_1 - \psi_0$ | $\Gamma(\tau, \psi_0) = \psi_0 + \tau$ |
| **Risk Ratio (RR)** | $\Phi(\psi_1, \psi_0) = \frac{\psi_1}{\psi_0}$ | $\Gamma(\tau, \psi_0) = \tau \cdot \psi_0$ |
| **Odds Ratio (OR)** | $\Phi(\psi_1, \psi_0) = \frac{\psi_1}{1-\psi_1} \cdot \frac{1-\psi_0}{\psi_0}$ | $\Gamma(\tau, \psi_0) = \frac{\tau \cdot \psi_0}{1 + \tau \cdot \psi_0 - \psi_0}$ |

The existence of $\Gamma$ ensures that for a fixed baseline, distinct treatment responses yield different causal measures. In the following, we use $\tau_\Phi^P$ to denote any first moment population causal measure evaluated under distribution $P$. Our objective is to estimate $\tau_\Phi^{\mathrm{T}} := \Phi\left(\mathbb{E}_{\mathrm{T}}[Y^{(1)}], \mathbb{E}_{\mathrm{T}}[Y^{(0)}]\right)$, for all effect measures $\Phi$, where the expectations are taken with respect to the target population distribution.

## 2.2 Identification assumptions

Because the effect $\tau_\Phi^{\mathrm{T}}$ is defined in terms of potential outcomes in the target population, it is not directly identifiable from trial data alone. The core difficulty lies in the fact that while treatment assignment in the trial is randomized, the trial sample may not be representative of the broader target population. To ensure identification of the estimand from observed data, we introduce standard causal inference assumptions (Assumption 1) and assume covariate overlap (Assumption 2).

**Assumption 1** (Trial's Internal Validity). *The RCT is assumed to be internally valid, that is:*

    A. ***Ignorability:*** $(Y^{(1)}, Y^{(0)}) \perp\!\!\!\perp A \mid S = 1$*;*

    B. ***Stable Unit Treatment Value Assumption (SUTVA):*** $Y = AY^{(1)} + (1 - A)Y^{(0)}$*;*

    C. ***Positivity and Randomized Assignment:*** $A \sim \mathcal{B}(\pi)$*, where $0 < \pi < 1$ (typically $\pi = 0.5$).*

**Assumption 2** (Overlap). *For all $x \in \mathrm{supp}(P_{\mathrm{T}})$, we have $\mathbb{P}(S = 1 \mid X = x) > 0$.*

## 3 Generalization under exchangeability in mean

In addition to internal validity and covariate overlap, identification under this setting requires a transportability condition that links the distribution of potential outcomes between the trial and target populations. This condition can be expressed as conditional exchangeability in mean, which states that, conditional on covariates, the average potential outcomes are the same across populations.

**Assumption 3** (Exchangeability in mean). *For all $x \in \mathrm{supp}(P_{\mathrm{S}}) \cap \mathrm{supp}(P_{\mathrm{T}})$ and for all $a \in \{0, 1\}$, we have $\mu_{(a)}^{\mathrm{S}}(x) = \mu_{(a)}^{\mathrm{T}}(x)$ where $\mu_{(a)}^P(x) = \mathbb{E}_P[Y^{(a)} \mid X = x]$.*

Leveraging observations from the randomized controlled trial and under Assumption 1 to 3, three identification formulas that express $\mathbb{E}_{\mathrm{T}}[Y^{(a)}]$ in terms of source population quantities can be derived:

$$\mathbb{E}_{\mathrm{T}}[Y^{(a)}] = \mathbb{E}_{\mathrm{T}}\left[\mathbb{E}_{\mathrm{S}}[Y^{(a)}|X]\right] \qquad \textit{(transporting conditional outcomes)} \tag{3}$$

$$= \mathbb{E}_{\mathrm{S}}\left[\frac{P_{\mathrm{T}}(X)}{P_{\mathrm{S}}(X)}Y^{(a)}\right] \qquad \textit{(weighting outcomes)} \tag{4}$$

$$= \mathbb{E}_{\mathrm{S}}\left[\frac{P_{\mathrm{T}}(X)}{P_{\mathrm{S}}(X)}\mathbb{E}_{\mathrm{S}}[Y^{(a)}|X]\right] \qquad \textit{(weighting conditional outcomes)}, \tag{5}$$

see Appendix B.1 for details. In the next section, we present several estimators of any first moment population causal measure, based on the three identification formulas above.

## 3.1 Weighting and regression strategies under exchangeability in mean

The Horvitz–Thompson estimator [21] is probably one of the most simple estimators to use in a RCT. Based on equation (4), we construct the weighted Horvitz-Thompson estimator. The following assumption is also required for technical reasons.

**Assumption 4.** *For $a \in \{0, 1\}$, we assume that $Y^{(a)}$ and $P_{\mathrm{T}}(X)Y^{(a)}/P_{\mathrm{S}}(X)$ are square-integrable.*

**Definition 2** (Weighted Horvitz-Thompson). *For any first moment population causal measure $\Phi$, we define the weighted Horvitz-Thompson estimator $\widehat{\tau}_{\Phi,wHT}$ as follows:*

$$\widehat{\tau}_{\Phi,wHT} = \Phi\left(\frac{1}{n}\sum_{S_i=1}\widehat{r}(X_i)\frac{A_i Y_i}{\pi}, \frac{1}{n}\sum_{S_i=1}\widehat{r}(X_i)\frac{(1-A_i)Y_i}{1-\pi}\right), \tag{6}$$

*where $\widehat{r}(X)$ is any estimator of the density ratio between the target and source covariate distributions.*

When $\widehat{r}$ is replaced by the oracle quantity $r$, we show that $\widehat{\tau}_{\Phi,wHT}$ is asymptotically normal and an unbiased estimator of $\tau_\Phi^{\mathrm{T}}$ (see Proposition 8 in Appendix B.2). Since

$$r(X) = \frac{P_{\mathrm{T}}(X)}{P_{\mathrm{S}}(X)} = \frac{\mathbb{P}(S=1)\,\mathbb{P}(S=0|X=x)}{\mathbb{P}(S=0)\,\mathbb{P}(S=1|X=x)},$$

one can estimate $r(X)$ by estimating $\mathbb{P}(S=1 \mid X=x)$ via a logistic regression model. By doing so, we avoid imposing a parametric model, such as a Gaussian distribution, on the distribution of $X$ [see also 24]. In this context, the $M$-estimation framework [42] can then be applied to derive asymptotic variances for this estimator.

**Logistic model.** *We assume $\mathbb{E}[XX^\top]$ is positive definite, $X$ is Sub-Gaussian, and $\exists \beta_\infty = (\beta_\infty^0, \beta_\infty^1) \in \mathbb{R}^{p+1}$ s.t. $\mathbb{P}(S=1|X) = \sigma(X, \beta_\infty) = \{1 + \exp(-X^\top\beta_\infty^1 - \beta_\infty^0)\}^{-1}$, a.s.*

**Proposition 1** (asymptotic normality of weighted Horvitz-Thompson estimator). *Grant Assumption 1 to 4. Let $\widehat{\beta}_N$ denote the maximum likelihood estimate (MLE) obtained from logistic regression of the selection indicator $S$ on covariates $X$. Define the estimated density ratio as*

$$r(x, \widehat{\beta}_N) = \frac{n}{N-n} \cdot \frac{1 - \sigma(x, \widehat{\beta}_N)}{\sigma(x, \widehat{\beta}_N)}, \qquad with \quad n = \sum_{i=1}^N S_i. \tag{7}$$

*Under the logistic model, the Horvitz-Thompson estimator $\widehat{\tau}_{\Phi,HT}$ weighted by the estimated ratio $r(x, \widehat{\beta}_N)$ is asymptotically unbiased and satisfies $\sqrt{N}\left(\widehat{\tau}_{\Phi,wHT} - \tau_\Phi^{\mathrm{T}}\right) \xrightarrow{d} \mathcal{N}\left(0, V_{\Phi,HT}\right)$.*

Appendix B.3 provides full proofs and asymptotic variances for all first moment population causal measures, including results for the Neyman estimator with estimated treatment probabilities—results that, to our knowledge, are novel even for the RD.

Alternatively, using the identification results in equations (3) and (5), one can adapt the regression-based approach of Robins [35] to derive weighted or transported G-formula estimators. While the transported version appears in Dahabreh et al. [12] for the RD, the weighted version, to the best of our knowledge, has not been previously presented.

**Definition 3** (Weighted/Transported G-formula). *For any first moment population causal measure $\Phi$, we define the weighted G-formula estimator $\widehat{\tau}_{\Phi,wG}$ and the transported G-formula estimator $\widehat{\tau}_{\Phi,tG}$ as:*

$$\widehat{\tau}_{\Phi,wG} = \Phi\left(\frac{1}{n}\sum_{S_i=1}\widehat{r}(X_i)\,\widehat{\mu}_{(1)}^{\mathrm{S}}(X_i), \frac{1}{n}\sum_{S_i=1}\widehat{r}(X_i)\,\widehat{\mu}_{(0)}^{\mathrm{S}}(X_i)\right), \tag{8}$$

$$\widehat{\tau}_{\Phi,tG} = \Phi\left(\frac{1}{N-n}\sum_{S_i=0}\widehat{\mu}_{(1)}^{\mathrm{S}}(X_i), \frac{1}{N-n}\sum_{S_i=0}\widehat{\mu}_{(0)}^{\mathrm{S}}(X_i)\right), \tag{9}$$

*where $\widehat{r}, \widehat{\mu}_{(1)}^{\mathrm{S}}, \widehat{\mu}_{(0)}^{\mathrm{S}}$ are any estimators of $r, \mu_{(1)}^{\mathrm{S}}, \mu_{(0)}^{\mathrm{S}}$.*

Using the $M$-estimation framework [42], one can derive asymptotic variance estimates for both G-formula estimators. Furthermore, assuming a linear model for the outcomes, we prove that regression adjustment leads to a lower asymptotic variance compared to the weighted Horvitz-Thompson.

**Linear model 1.** *For all $a, s \in \{0, 1\}$, let $Y_s^{(a)} = V^\top \beta_s^{(a)} + \epsilon_s^{(a)}$, where $V^\top = [1, X^\top]$ with $\mathbb{E}[\epsilon_s^{(a)} \mid X] = 0$ and $Var(\epsilon_s^{(a)} \mid X) = \sigma^2$.*

**Proposition 2** (Asymptotic Normality of weighted and Transported G-formula Estimators). *Let $\widehat{\tau}_{\Phi,\mathrm{tG}}$ and $\widehat{\tau}_{\Phi,\mathrm{wG}}$ denote respectively the weighted G-formula where $\widehat{r}$ is a logistic regression estimator (7) and $\widehat{\mu}_{(1)}^S, \widehat{\mu}_{(0)}^S$ are ordinary least squares estimator. Then, under Assumptions 1 to 4,*

$$\sqrt{N} \left( \widehat{\tau}_{\Phi,\mathrm{wG}} - \tau_\Phi^\mathrm{T} \right) \xrightarrow{d} \mathcal{N} \left( 0, V_{\Phi,\mathrm{wG}}^{OLS} \right), \qquad \sqrt{N} \left( \widehat{\tau}_{\Phi,\mathrm{tG}} - \tau_\Phi^\mathrm{T} \right) \xrightarrow{d} \mathcal{N} \left( 0, V_{\Phi,\mathrm{tG}}^{OLS} \right).$$

*Besides, $V_{\Phi,\mathrm{tG}}^{OLS} \leq V_{\Phi,\mathrm{wG}}^{OLS} \leq V_{\Phi,HT}$, where all variances are explicit Appendix B.4.*

Given these results, one might wonder whether it's better to avoid estimating the density ratio altogether, as the transported G-formula is more efficient under correct specification. However, when models are misspecified, weighting the outcomes may perform better. This motivates doubly robust estimators, which retain consistency if either model is correctly specified.

## 3.2 Semiparametric efficient estimators under exchangeability in mean

A common approach to constructing doubly robust estimators relies on semiparametric efficiency theory. By deriving the efficient influence function (EIF) of the target parameter [see, e.g. 25], one can build estimators that are not only robust to misspecification, but also achieve the lowest possible asymptotic variance among unbiased estimators. We denote by $\varphi_\Phi(Z, \eta, \psi_1^\mathrm{T}, \psi_0^\mathrm{T})$ the EIF of $\tau_\Phi$, which depends on $(i)$ the nuisance parameters $\eta = (\mu_{(0)}, \mu_{(1)}, r)$ and $(ii)$ the values of $\psi_1^\mathrm{T}$ and $\psi_0^\mathrm{T}$.

Based on estimators $\widehat{\eta}, \widehat{\psi}_1^\mathrm{T}, \widehat{\psi}_0^\mathrm{T}$, one can use the EIF framework via one of the following two techniques to build new estimators of $\tau_\Phi^\mathrm{T}$ whose properties are described below:

(i) **One-step estimators** $\widehat{\tau}_\Phi^\mathrm{OS}$ consist in applying a first-order bias correction to an initial plug-in estimator $\widehat{\tau}_\Phi^\mathrm{T} = \Phi(\widehat{\psi}_1^\mathrm{T}, \widehat{\psi}_0^\mathrm{T})$, resulting in $\widehat{\tau}_\Phi^\mathrm{OS} = \widehat{\tau}_\Phi^\mathrm{T} + \frac{1}{N} \sum_{i=1}^N \varphi_\Phi(Z_i, \widehat{\eta}, \psi_1^\mathrm{T}, \widehat{\psi}_0^\mathrm{T})$.

(ii) **Estimating equation estimators** are obtained by setting the empirical mean of the EIF to zero. This amounts to finding an estimators $\widehat{\tau}_\Phi^\mathrm{EE} = \Phi(\widehat{\psi}_1^\mathrm{T}, \widehat{\psi}_0^\mathrm{T})$ such that $\widehat{\psi}_1^\mathrm{T}, \widehat{\psi}_0^\mathrm{T}$ are solutions to $\sum_{i=1}^N \varphi_\Phi(Z_i, \widehat{\eta}, \widehat{\psi}_1^\mathrm{T}, \widehat{\psi}_0^\mathrm{T}) = 0$ (Estimating Equation).

In practice and in both case, it is usual to resort to crossfitted techniques [3] by estimating $\widehat{\tau}_\Phi$ and $\widehat{\eta}$ and evaluating the EIF $\varphi_\Phi$ on two different datasets to enforce independence. We drop the T superscript in the rest of this section for the sake of clarity. Both above approaches require knowing the EIF $\varphi_\Phi$. As $\tau_\Phi = \Phi(\psi_1, \psi_0)$, letting $\varphi_1$ (resp. $\varphi_0$) be the EIF of $\psi_1$ (resp. $\psi_0$), standard calculations on EIF functions (see e.g. [26, Sec 3.4.3]) show that

$$\varphi_\Phi(Z, \eta, \psi_1, \psi_0) = \partial_1 \Phi(\psi_1, \psi_0) \varphi_1(Z, \eta, \psi_1) + \partial_0 \Phi(\psi_1, \psi_0) \varphi_0(Z, \eta, \psi_0).$$

With this equality, and the following proposition, one can compute the influence function $\varphi_\Phi$.

**Proposition 3.** *Grant Assumption 1 to 3. For all $a \in \{0, 1\}$, we have*

$$\varphi_a(Z, \eta, \psi_a) := \frac{1 - S}{1 - \alpha}(\mu_{(a)}(X) - \psi_a) + \frac{S \mathbb{1}\{A = a\}}{\alpha P(A = a)} r(X)(Y - \mu_{(a)}(X)).$$

The exact expressions of one-step and estimating equation estimators depend on $\varphi_\Phi$, which in turns depends on the effect measure $\Phi$. While the first approach leads to explicit expressions, estimating equation estimators are defined implicitly through the estimating equation in $(ii)$, which writes

$$\partial_1 \Phi(\widehat{\psi}_1, \widehat{\psi}_0) \frac{1}{N} \sum_{i=1}^N \varphi_1(Z, \widehat{\eta}, \widehat{\psi}_1) + \partial_0 \Phi(\widehat{\psi}_1, \widehat{\psi}_0) \frac{1}{N} \sum_{i=1}^N \varphi_0(Z, \widehat{\eta}, \widehat{\psi}_0) = 0. \qquad (10)$$

One way — though not the only — to satisfy (10) is to find two estimators $\widehat{\psi}_1, \widehat{\psi}_0$ that cancels the empirical mean of their EIF (second and fourth term in (10)), that is the corresponding EE estimator for $\psi_1$ and $\psi_0$. We end up with a plug-in estimator of the form $\Phi(\widehat{\psi}_1^\mathrm{EE}, \widehat{\psi}_0^\mathrm{EE})$, whose precise expression is given below.

**Proposition 4** (Estimating equation estimators.). *Given estimators $\widehat{\mu}_{(a)}$ (resp. $\widehat{r}$) of $\mu_{(a)}$ (resp. $r$), an estimating equation estimator $\widehat{\tau}_\Phi^{\mathrm{EE}}$ of $\tau_\Phi$ is given by $\widehat{\tau}_\Phi^{\mathrm{EE}} = \Phi(\widehat{\psi}_1^{\mathrm{EE}}, \widehat{\psi}_0^{\mathrm{EE}})$ where for all $a \in \{0, 1\}$*

$$\widehat{\psi}_a^{\mathrm{EE}} := \frac{1}{m} \sum_{S_i=0} \widehat{\mu}_{(a)}(X_i) + \frac{1-\alpha}{m\alpha} \sum_{S_i=1} \frac{\mathbb{1}\{A = a\}}{\mathbb{P}(A = a)} \widehat{r}(X_i)(Y - \widehat{\mu}_{(a)}(X_i)). \tag{11}$$

A similar estimator already appears in [12]. We prove below that $\widehat{\tau}_\Phi^{\mathrm{EE}}$ is *(weakly) doubly-robust*.

**Proposition 5.** *Let $\widehat{\mu}_{(a)}$ and $\widehat{r}$ be two estimators independent from $Z_1, \ldots, Z_n$. Then, under boundedness of $\widehat{\mu}^{(a)}$, $\widehat{r}$ and $Y$, and assuming that $\Phi$ is continuous, the estimator $\widehat{\tau}_\Phi^{\mathrm{EE}} = \Phi(\widehat{\psi}_1^{\mathrm{EE}}, \widehat{\psi}_0^{\mathrm{EE}})$ is consistent as soon as either $\widehat{\mu}_{(a)} = \mu$ for all $a \in \{0, 1\}$ or $\widehat{r} = r$.*

It would be easy to extend this result to a *strong* robustness property, meaning that $\sqrt{N}(\widehat{\tau}_\Phi^{\mathrm{EE}} - \tau_\Phi^{\mathrm{T}}) \xrightarrow{d} \mathcal{N}(0, \mathrm{Var}\, \varphi_\Phi(Z))$ under mild convergence requirements of $\widehat{\mu}_{(a)}$ and $\widehat{r}$ towards $\mu_{(a)}$ and $r$. Concerning the OS estimators, we cannot derive any formal robustness results in full generality, and this property needs to be assessed on a case-by-case basis. For instance, [1] shows that the OS estimator for the RR is doubly robust based on usual requirements on the estimation of the nuisance parameters and some extra assumptions on $\widehat{\psi}_0^{\mathrm{T}}$. In light of these observations, we recommend using EE estimators rather than OS estimators, when possible, as they also usually lead to better results empirically, see Figure 1.

When $\Phi$ is a linear functional (e.g., RD), the two approaches, one-step and estimating equations yields the same estimators, up to a scaling term depending on $\alpha$ and $N$. For nonlinear functionals however (such as the RR or OR), the estimators generally differ (see below and Appendix B.6).

(RD)  For the risk difference, the estimating equation approach yields $\widehat{\tau}_{\mathrm{RD}}^{\mathrm{EE}} = \widehat{\psi}_1^{\mathrm{EE}} - \widehat{\psi}_0^{\mathrm{EE}}$. The one step approach, based on initial estimators $\widehat{\psi}_1$ and $\widehat{\psi}_0$ yields an estimate of the form:

$$\widehat{\tau}_{\mathrm{RD}}^{\mathrm{OS}} = \frac{m}{N(1-\alpha)} \widehat{\tau}_{\mathrm{RD}}^{\mathrm{EE}} + \left(1 - \frac{m}{N(1-\alpha)}\right) \widehat{\tau}_{\mathrm{RD}}.$$

In particular, estimators $\widehat{\psi}_a$ of the form G-formula or weighted Horwitz-Thomson yield a final one-step estimator that has the same structure as the estimating equation estimator.

(RR)  For the risk ratio, the estimating equation approach yield $\widehat{\tau}_{\mathrm{RR}}^{\mathrm{EE}} = \widehat{\psi}_1^{\mathrm{EE}} / \widehat{\psi}_0^{\mathrm{EE}}$. In contrast, the one-step approach, based on initial estimators $\widehat{\psi}_1$ and $\widehat{\psi}_0$ yields

$$\widehat{\tau}_{\mathrm{RR}}^{\mathrm{OS}} = \frac{\widehat{\psi}_1}{\widehat{\psi}_0} + \frac{1}{\widehat{\psi}_0} \frac{m}{N(1-\alpha)} (\widehat{\psi}_1^{\mathrm{EE}} - \widehat{\psi}_1) - \frac{\widehat{\psi}_1}{\widehat{\psi}_0^2} \frac{m}{N(1-\alpha)} (\widehat{\psi}_0^{\mathrm{EE}} - \widehat{\psi}_0).$$

In general, $\widehat{\tau}_{\mathrm{RR}}^{\mathrm{OS}} \neq \widehat{\tau}_{\mathrm{RR}}^{\mathrm{EE}}$, unless of course we initially picked $\widehat{\psi}_a = \widehat{\psi}_a^{\mathrm{EE}}$.

## 4  Generalization under exchangeability in effect measure

Estimators in Section 3 were derived under the exchangeability in mean assumption (Assumption 3). Under this assumption, generalizing necessitates full access to all prognostic covariates whose distributions is shifted between the source and target populations [8]. A weaker assumption consists in transportability of conditional treatment effects.

**Assumption 5** (Exchangeability in effect measure). $\forall x \in \mathrm{supp}(P_{\mathrm{T}}) \cap \mathrm{supp}(P_{\mathrm{S}})$, $\tau_\Phi^{\mathrm{S}}(x) = \tau_\Phi^{\mathrm{T}}(x)$.

Under Assumption 5, the effect of the treatment depends on the patient's features in the same way in the source and target population. While exchangeability in mean implies exchangeability of the effect measure for any $\Phi$, the reverse does not generally hold.

### 4.1  Transporting causal measures under exchangeability in effect measure

To ensure identification, we typically require a relationship between the conditional average treatment effect (CATE) and the average treatment effect (ATE). A classical concept that supports this

relationship is collapsibility: a measure is said to be collapsible if it can be expressed as a weighted average of conditional effects [see, e.g., 19, 23]. However, some measure (like OR) are not collapsible, thus questioning their transportability under this Assumption 5. However, it turns out that under Assumption 5, any first-moment causal measure is identifiable assuming access to the control potential outcomes $Y^{(0)}$ for all individuals in the target population:

$$\tau_\Phi^{\mathrm{T}} = \Phi\left(\mathbb{E}_{\mathrm{T}}\left[\Gamma\left(\tau_\Phi^{\mathrm{S}}(X), \mu_{(0)}^{\mathrm{T}}(X)\right)\right], \mathbb{E}_{\mathrm{T}}\left[Y^{(0)}\right]\right) \qquad \textit{(transporting)} \qquad (12)$$

$$= \Phi\left(\mathbb{E}_{\mathrm{S}}\left[\frac{p_{\mathrm{T}}(X)}{p_{\mathrm{S}}(X)}\Gamma\left(\tau_\Phi^{\mathrm{S}}(X), \mu_{(0)}^{\mathrm{T}}(X)\right)\right], \mathbb{E}_{\mathrm{T}}\left[Y^{(0)}\right]\right) \qquad \textit{(weighting)} \qquad (13)$$

where $\Gamma$ is the effect function (see Definition 1). Note that for collapsible measures such as RD or RR, expressions (12) and (13) reduce to the identification results derived by [7]. Thus, our framework can be viewed as a natural extension of their approach to any first moment population causal measures even non-collapsible measures such as the OR. The detailed identification results for RD, OR, and RR are provided in Appendix C. Using the identification formula (12), we derive $\Gamma$-formula estimators, which, to the best of our knowledge, are novel contributions.

**Definition 4** (Transported $\Gamma$-formula). *For any first moment population causal measure $\Phi$, we define the transported $\Gamma$-formula estimator $\hat{\tau}_{\Phi,\mathrm{t}\Gamma}$ as follows:*

$$\hat{\tau}_{\Phi,\mathrm{t}\Gamma} = \Phi\left(\frac{1}{N-n}\sum_{S_i=0}\Gamma(\tau_\Phi^{\mathrm{S}}(X_i), \mu_{(0)}^{\mathrm{T}}(X_i)), \frac{1}{N-n}\sum_{S_i=0}\mu_{(0)}^{\mathrm{T}}(X_i)\right). \qquad (14)$$

We can also define, as before, a reweighted version of the $\Gamma$-formula (see Definition 3).

### 4.2 Semiparametric efficient estimators under exchangeability of treatment effect

Under Assumption 5, the identifiability formula (12) serves as the basis for constructing the EIF. Given access to the target baseline distribution, $\mu_{(0)}^{\mathrm{T}}$ and $\psi_0^{\mathrm{T}}$ are known. To construct one-step and estimating equation estimators (see Section 3.2), we require EIF of $\tau_\Phi^{\mathrm{T}}$, denoted $\varphi_\Phi(Z, \eta, \tau_\Phi^{\mathrm{T}})$ which is is related to $\varphi_1$, the EIF of $\psi_1^{\mathrm{T}}$, via the chain rule:

$$\varphi_\Phi(Z, \eta, \tau_\Phi^{\mathrm{T}}) = \varphi_1(Z, \eta, \psi_1^{\mathrm{T}})\partial_1\Phi(\psi_1^{\mathrm{T}}, \psi_0^{\mathrm{T}}),$$

where $\eta = (\mu_{(0)}^{\mathrm{S}}, \tau_\Phi, r)$ is the nuisance parameters.

**Proposition 6.** *The influence function of $\psi_1^{\mathrm{T}}$ at $P_{\mathrm{obs}}$ is given by*

$$\varphi_1(Z, \eta, \psi_1^{\mathrm{T}}) := \frac{Sr(X)}{\alpha}\left(\frac{A}{\pi}(Y - \Gamma(\tau_\Phi(X), \mu_{(0)}^{\mathrm{S}}(X)))\right)$$
$$- \frac{Sr(X)}{\alpha}\left(\frac{1-A}{1-\pi}(Y - \mu_{(0)}^{\mathrm{S}}(X))\partial_0\Gamma(\tau_\Phi(X), \mu_{(0)}^{\mathrm{T}}(X))\right) + \frac{1-S}{1-\alpha}\left(\Gamma(\tau_\Phi(X), \mu_{(0)}^{\mathrm{T}}(X)) - \psi_1^{\mathrm{T}}\right).$$

As in Section 3.2, we can either find the estimating equation estimator of $\psi_1$ and plug it into $\Phi$ to obtain $\hat{\tau}_\Phi^{\mathrm{EE}} = \Phi(\hat{\psi}_1^{\mathrm{EE}}, \psi_0^{\mathrm{T}})$, or apply a one step correction to a initial estimator $\hat{\psi}_1$ of the form $\hat{\tau}_\Phi^{\mathrm{OS}} = \Phi(\hat{\psi}_1, \psi_0^{\mathrm{T}}) + (1/N)\sum_{i=1}^N \varphi_\Phi(Z_i, \hat{\eta}, \hat{\tau}_\Phi^{\mathrm{T}})$. Like in Section 3.2, the estimating equation estimators and one-step estimators have no reason to coincide in general. Exact computations of the RD, RR and OR are provided in Appendix C.3.

## 5 Simulations

### 5.1 Synthetic data

We generate data $(S, X, A, Y^{(0)}, Y^{(1)})$ using the following binary outcome model: for all $a, s \in \{0, 1\}$, $\mathbb{P}(Y^{(a)} = 1 \mid X, S = s) = p_s^{(a)}(V)$ where $V^\top = [1, X^\top]$ and $X|S = s \sim \mathcal{N}(\nu_s, I_d)$. We set $d = 5$ and $S \sim \mathrm{B}(0.3)$ to reflect limited RCT data relative to the target population, and $A \mid S = 1 \sim \mathrm{B}(0.5)$. We evaluate the estimators from Section 3 and 4, estimating nuisance components—regression surfaces and density ratios—using parametric methods (linear/logistic

regression). The red (resp. gray, when displayed) dotted line represents the treatment effect in the target (resp. source) population. A basic linear setting, in which all estimators perform well, is presented in Appendix D.

**Experiment 1** (Exchangeability in mean and non-linear/non-logistic response): We consider a setting under which Assumption 3 holds and $\forall a, s \in \{0, 1\}$, $p_s^{(a)}(V) = \sigma(\beta_0^\top V \cdot (V^\top \beta_1)^a)$. Both G-formula-based estimators exhibit substantial bias across all evaluation metrics, which is expected given that the non-linear response surfaces are misspecified by using linear regression. In contrast, the estimating equation–based estimators remain unbiased across all measures, benefiting from their double robustness property. Among the one-step estimators, only the RD variant is accurate in this setting. This is also anticipated, as one-step estimators generally do not retain double robustness—except in cases where they coincide with the corresponding estimating equation estimator, which holds true for the RD variant.

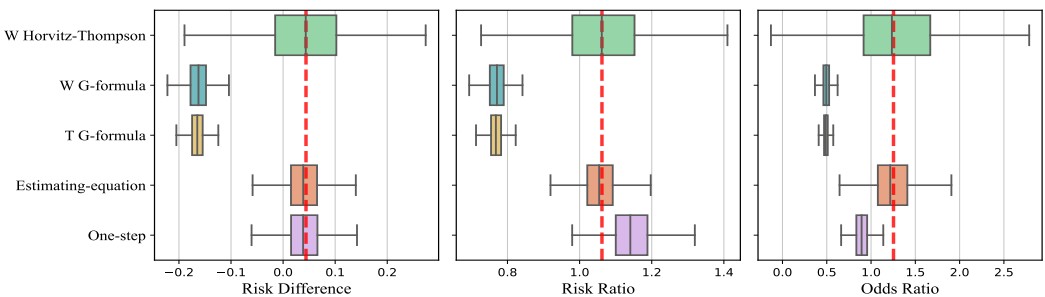

Figure 1: Comparison of estimators across different causal measures under a non-linear outcome model with a sample size of $N = 50{,}000$ and 3,000 repetitions. Source values are 0.45 / 3.2 / 7.5.

**Experiment 2** (Exchangeability in effect and linear/logistic response): We now consider a setting for which Assumption 5 holds and such that, depending on the causal measure: (RD) Model 1 and $\beta_T^{(a)} = \beta_S^{(a)} + \theta$, (RR) $p_s^{(a)}(X) = \sigma(X^\top \beta_s) \cdot \sigma(X^\top \gamma)^a$, (OR) $p_s^{(a)}(X) = \sigma(X^\top(\beta_s + a \cdot \gamma))$. In this setting, Assumption 3 is no longer satisfied, as the nuisance functions depend on $S$. However, one can verify that Assumption 5 holds for each model. Estimators introduced in Section 3 still converge for the RD, despite the violation of strong transportability, thanks to the linearity of the RD. In contrast, the RR and OR estimators fail to converge, since these measures are nonlinear. On the other hand, all estimators introduced in Section 4 remain unbiased, as expected.

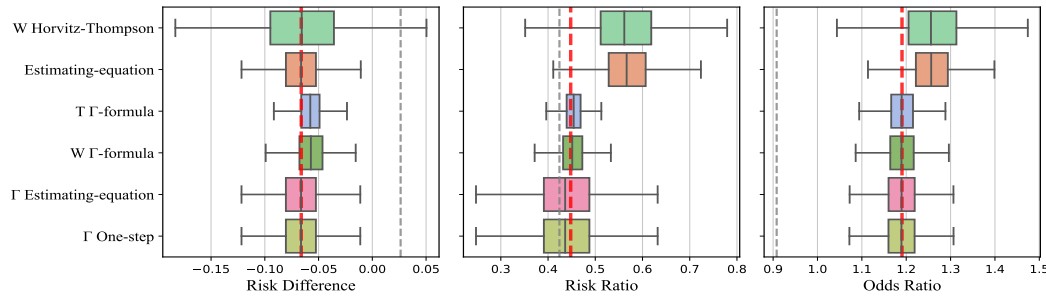

Figure 2: Comparison of estimators across different causal measures under a linear outcome model with a sample size of $N = 50{,}000$ and 3,000 repetitions.

## 5.2 Real-World Experiment

We evaluate estimators using a case study on the effectiveness of tranexamic acid on mortality for brain injury patients, combining data from the CRASH-3 trial and the Traumabase registry. CRASH-3, is a RCT with over 9,000 TBI patients from 29 countries, while Traumabase provides detailed clinical data on 8,000+ patients from 23 French trauma centers. Following [9] we consider

six covariates (age, sex, injury time, systolic BP, GCS score, pupil reactivity). Since this is a real dataset, the true treatment effect is unknown. Results are displayed in Figure 3. All estimators (except one) indicate a positive treatment effect; however, the confidence intervals are wide and include the null value (zero or one), preventing any definitive conclusions about the treatment's effectiveness.

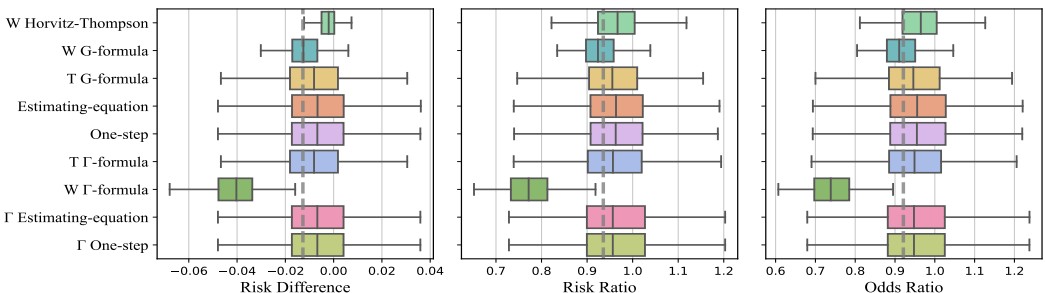

Figure 3: Comparison of estimators across different causal measures on the combined CRASH-3 and Traumabase dataset. Confidence intervals were estimated using stratified bootstrap resampling.

## 6  Conclusion

This article introduces a general framework for the generalization of first-moment, population-level causal estimands from RCTs to broader target populations. We propose different estimators (weighting based on density ratios, outcome regression methods, and approaches based on EIF) and analyze their statistical properties. A central source of complexity lies in the inherent nonlinearity of many causal estimands, which disrupts the alignment between two classical EIF techniques: one-step and estimating equations. This divergence gives rise to a diverse landscape of possible estimators, which differ depending on the underlying identification assumptions. While assuming exchangeability in effect measure is less restrictive than exchangeability in mean, few estimating methods are available in the former case, as most of them are based on CATE estimation, which is difficult for non-linear measure. In practice, we recommend to resort to EE estimators, as they are provably doubly robust and behave well in our controlled experimental setting. Further layers of complexity arise from the selection of appropriate nuisance estimators, whether parametric or nonparametric, each with its own trade-offs. Nevertheless, this work represents a step forward in enabling the computation of both absolute and relative causal measures in new target populations, thereby contributing to more informed clinical decision-making and external validity in causal research.

## 7  Acknowledgments

This work has been done in the frame of the PEPR SN SMATCH project and has benefited from a governmental grant managed by the Agence Nationale de la Recherche un- der the France 2030 programme, reference ANR-22-PESN- 0003.

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

# Technical Appendices and Supplementary Material

## Contents

# A First Moment Causal Measures

| Measure | Effect Measure $\Phi(\psi_1, \psi_0)$ | Effect Function $\Gamma(\tau, \psi_0)$ |
|---|---|---|
| **Risk Difference (RD)** | $\Phi(\psi_1, \psi_0) = \psi_1 - \psi_0$ | $\Gamma(\tau, \psi_0) = \psi_0 + \tau$ |
| **Risk Ratio (RR)** | $\Phi(\psi_1, \psi_0) = \frac{\psi_1}{\psi_0}$ | $\Gamma(\tau, \psi_0) = \tau \cdot \psi_0$ |
| **Odds Ratio (OR)** | $\Phi(\psi_1, \psi_0) = \frac{\psi_1}{1-\psi_1} \cdot \frac{1-\psi_0}{\psi_0}$ | $\Gamma(\tau, \psi_0) = \frac{\tau \cdot \psi_0}{1 + \tau \cdot \psi_0 - \psi_0}$ |
| **Number Needed to Treat (NNT)** | $\Phi(\psi_1, \psi_0) = \frac{1}{\psi_1 - \psi_0}$ | $\Gamma(\tau, \psi_0) = \frac{1}{\tau} + \psi_0$ |
| **Switch Relative Risk (GRRR)** | $\Phi(\psi_1, \psi_0) = \begin{cases} 1 - \frac{1-\psi_1}{1-\psi_0} & \text{if } \psi_1 > \psi_0 \\ 0 & \text{if } \psi_1 = \psi_0 \\ -1 + \frac{\psi_1}{\psi_0} & \text{if } \psi_1 < \psi_0 \end{cases}$ | $\Gamma(\tau, \psi_0) = \begin{cases} 1 - (1-\tau)(1-\psi_0) & \text{if } \psi_0 > 0 \\ \tau & \text{if } \psi_0 = 0 \\ \tau(1+\psi_0) & \text{if } \psi_0 < 0 \end{cases}$ |
| **Excess Risk Ratio (ERR)** | $\Phi(\psi_1, \psi_0) = \frac{\psi_1 - \psi_0}{\psi_0}$ | $\Gamma(\tau, \psi_0) = \tau(1 + \psi_0)$ |
| **Survival Ratio (SR)** | $\Phi(\psi_1, \psi_0) = \frac{1-\psi_1}{1-\psi_0}$ | $\Gamma(\tau, \psi_0) = 1 - \tau(1 - \psi_0)$ |
| **Relative Susceptibility (RS)** | $\Phi(\psi_1, \psi_0) = \frac{1-\psi_0}{1-\psi_1}$ | $\Gamma(\tau, \psi_0) = 1 - \frac{1-\tau}{\psi_0}$ |
| **Log Odds Ratio (log-OR)** | $\Phi(\psi_1, \psi_0) = \log\left(\frac{\psi_1(1-\psi_0)}{\psi_0(1-\psi_1)}\right)$ | $\Gamma(\tau, \psi_0) = \exp(\psi_0) \cdot \frac{\tau}{1 - \tau + \exp(\psi_0) \cdot \tau}$ |
| **Odds Product** | $\Phi(\psi_1, \psi_0) = \frac{\psi_1}{1-\psi_1} \cdot \frac{\psi_0}{1-\psi_0}$ | $\Gamma(\tau, \psi_0) = \frac{\sqrt{\tau \cdot \frac{\psi_0}{1-\psi_0}}}{1 + \sqrt{\tau \cdot \frac{\psi_0}{1-\psi_0}}}$ |
| **Arcsine Difference** | $\Phi(\psi_1, \psi_0) = \arcsin(\sqrt{\psi_1}) - \arcsin(\sqrt{\psi_0})$ | $\Gamma(\tau, \psi_0) = \sin(\psi_0 + \arcsin(\sqrt{\tau}))^2$ |
| **Relative Risk Reduction (RRR)** | $\Phi(\psi_1, \psi_0) = 1 - \frac{\psi_1}{\psi_0}$ | $\Gamma(\tau, \psi_0) = \tau(1 - \psi_0)$ |

# B Transporting/Reweighting a causal effect under exchangeability of conditional outcome

## B.1 Identification under exchangeability of conditional outcome

$$
\begin{aligned}
\mathbb{E}_T\left[Y^{(a)}\right] &:= \mathbb{E}_T\left[\mathbb{E}_T\left[Y^{(a)} \mid X\right]\right] \\
&= \mathbb{E}_T\left[\mathbb{E}_S\left[Y^{(a)} \mid X\right]\right] \quad \text{(Transportability)} \\
&= \mathbb{E}_S\left[\mathbb{E}_S\left[Y^{(a)} \mid X\right] \cdot \frac{P_T(X)}{P_S(X)}\right] \quad \text{(Overlap)} \\
&= \mathbb{E}_S\left[Y^{(a)} \cdot \frac{P_T(X)}{P_S(X)}\right]
\end{aligned}
$$

## B.2 Oracle weighted Horvitz-Thompson

**Proposition 7.** *For ease of notation, we let $\pi_a := \mathbb{P}_S(A = a) = \pi^a(1-\pi)^{1-a}$. Let $\psi_a = \mathbb{E}_T[Y^{(a)}]$ denote the target population mean potential outcome under treatment $a \in \{0, 1\}$. Define the oracle estimator*

$$
\psi_{a,wHT}^* = \frac{1}{n} \sum_{i=1}^N S_i \cdot r(X_i) \cdot \frac{\mathbb{1}\{A_i = a\}Y_i}{\pi_a},
$$

*where $r(X)$ denotes the density ratio between the target and source covariate distributions and $n = \sum_{i=1}^N S_i$ is the number of units in the source sample. Then, under Assumption 1 to 4 we have,*

$$
\sqrt{N}\left(\psi_{a,wHT}^* - \psi_a\right) \xrightarrow{d} \mathcal{N}(0, V_{a,wHT}^*),
$$

*where the asymptotic variance $V^*_{a,wHT}$ is given by*

$$V^*_{a,wHT} = \frac{1}{\alpha}\left(\frac{1}{\pi_a}\mathbb{E}_T\left[r(X)(Y^{(a)})^2\right] - (\psi_a)^2\right), \tag{15}$$

*Proof.* Define

$$Z_i = S_i \cdot r(X_i) \cdot \frac{\mathbb{1}\{A_i = a\}Y_i}{\pi_a},$$

so that

$$\psi^*_{a,wHT} = \frac{1}{N}\sum_{i=1}^{N} Z_i.$$

We first compute the expectations of $Z_i$ and $S_i$. By the definition of $Z_i$ and using that $S_i \sim$ Bernoulli$(\alpha)$,

$$\begin{aligned}
\mathbb{E}[Z_i] &= \mathbb{E}\left[S \cdot r(X) \cdot \frac{\mathbb{1}\{A = a\}Y}{\pi_a}\right] \\
&= \alpha\,\mathbb{E}_S\left[r(X) \cdot \frac{\mathbb{1}\{A = a\}Y}{\pi_a}\right] \\
&= \alpha\,\mathbb{E}_S\left[r(X) \cdot Y^{(a)}\right] \\
&= \alpha\,\mathbb{E}_T[Y^{(a)}] \\
&= \alpha\psi_a,
\end{aligned}$$

where we used the consistency assumption $Y = Y^{(A)}$, the randomization of $A$, and the density ratio property. Since $(Z_i, S_i)$ for $i = 1, \dots, N$ are i.i.d., one can apply the multivariate central limit theorem to

$$\left(\frac{1}{N}\sum_{i=1}^{N} Z_i, \quad \frac{1}{N}\sum_{i=1}^{N} S_i\right),$$

which leads to

$$\sqrt{N}\left(\begin{pmatrix}\frac{1}{N}\sum_{i=1}^{N} Z_i \\ \frac{1}{N}\sum_{i=1}^{N} S_i\end{pmatrix} - \begin{pmatrix}\mathbb{E}[Z_i] \\ \mathbb{E}[S_i]\end{pmatrix}\right) \xrightarrow{d} \mathcal{N}(0, \Sigma),$$

where

$$\Sigma = \begin{pmatrix}\mathbb{V}[Z_i] & \mathrm{Cov}(Z_i, S_i) \\ \mathrm{Cov}(S_i, Z_i) & \mathbb{V}[S_i]\end{pmatrix}.$$

Noting that $\psi^*_{a,wHT}$ can be written as

$$\psi^*_{a,wHT} = \frac{\frac{1}{N}\sum_{i=1}^{N} Z_i}{\frac{1}{N}\sum_{i=1}^{N} S_i},$$

one can apply the Delta method to the map $h : (x, y) \mapsto x/y$, whose gradient evaluated at $(\alpha\psi_a, \alpha)$ is

$$\nabla h(\alpha\psi_a, \alpha) = \left(\frac{1}{\alpha}, \; -\frac{\psi_a}{\alpha}\right).$$

Thus,

$$\sqrt{N}\left(\psi^*_{a,wHT} - \psi_a\right) \xrightarrow{d} \mathcal{N}(0, V^*_{a,wHT}),$$

where

$$V^*_{a,wHT} = \nabla h(\alpha\psi_a, \alpha)^\top \Sigma \nabla h(\alpha\psi_a, \alpha) \tag{16}$$

$$= \frac{1}{\alpha^2}\mathbb{V}[Z_i] + \frac{(\psi_a)^2}{\alpha^2}\mathbb{V}[S_i] - \frac{2\psi_a}{\alpha^2}\mathrm{Cov}(Z_i, S_i). \tag{17}$$

It remains to compute each term. Since $S_i \sim$ Bernoulli$(\alpha)$,

$$\mathbb{V}[S_i] = \alpha(1 - \alpha).$$

By direct computation,

$$\mathbb{V}[Z_i] = \alpha \, \mathbb{E}_T \left[ \frac{r(X)(Y^{(a)})^2}{\pi_a} \right] - (\alpha \psi_a)^2.$$

Moreover, since $S_i \times Z_i = Z_i$, we have

$$\text{Cov}(Z_i, S_i) = (1 - \alpha)\alpha \psi_a.$$

Substituting into the expression of $V_{a,\text{wHT}}^*$, we find

$$V_{a,\text{wHT}}^* = \frac{1}{\alpha} \left( \frac{1}{\pi_a} \mathbb{E}_T \left[ r(X)(Y^{(a)})^2 \right] - (\psi_a)^2 \right), \tag{18}$$

as claimed. $\qquad \square$

**Proposition 8** (Asymptotic Normality Oracle weighted Horvitz-Thompson). *Let $\tau_{\Phi,wHT}^*$ denote the oracle weighted Horvitz-Thompson estimator. Then under Assumption 1 to 4, we have:*

$$\sqrt{N} \left( \tau_{\Phi,wHT}^* - \tau_\Phi^T \right) \xrightarrow{d} \mathcal{N} \left( 0, V_{\Phi,wHT}^* \right).$$

*Proof.* Noting that $\tau_{\Phi,\text{wHT}}^* = \Phi(\psi_{1,\text{wHT}}^*, \psi_{0,\text{wHT}}^*)$ and using Proposition 7, we know that $(\psi_{1,\text{wHT}}^*, \psi_{0,\text{wHT}}^*)$ are jointly asymptotically normal. Specifically,

$$\sqrt{N} \begin{pmatrix} \psi_{1,\text{wHT}}^* - \psi_1 \\ \psi_{0,\text{wHT}}^* - \psi_0 \end{pmatrix} \xrightarrow{d} \mathcal{N} \left( 0, \Sigma_\pi^* \right),$$

where $\Sigma_\pi^*$ is the asymptotic covariance matrice. Applying the delta method to the smooth function $\Phi : \mathbb{R}^2 \to \mathbb{R}$, we have

$$\sqrt{N}(\tau_{\Phi,\text{wHT}}^* - \tau_\Phi^T) \xrightarrow{d} \mathcal{N} \left( 0, \nabla \Phi^\top \Sigma_\pi^* \nabla \Phi \right),$$

where $\nabla \Phi$ denotes the gradient of $\Phi$ evaluated at $(\psi_1, \psi_0)$. Moreover, because the treatment assignment $A$ is binary ($A \in \{0, 1\}$), we have that for each unit, $A_i(1 - A_i) = 0$, so the covariance between $\psi_1^*$ and $\psi_0^*$ is

$$\text{Cov}(\psi_{1,\text{wHT}}^*, \psi_{0,\text{wHT}}^*) = -\psi_1 \psi_0.$$

Expanding the delta method variances and fully factorizing we get:

$$V_{\Phi,\text{wHT}}^* = \frac{1}{\alpha} \left( \left( \frac{\partial \Phi}{\partial \psi_1} \right)^2 \mathbb{E}_T \left[ \frac{r(X)(Y^{(1)})^2}{\mathbb{P}_S(A = 1)} \right] + \left( \frac{\partial \Phi}{\partial \psi_0} \right)^2 \mathbb{E}_T \left[ \frac{r(X)(Y^{(0)})^2}{\mathbb{P}_S(A = 0)} \right] \right.$$
$$\left. - \left( \frac{\partial \Phi}{\partial \psi_1} \mathbb{E}_T \left[ Y^{(1)} \right] + \frac{\partial \Phi}{\partial \psi_0} \mathbb{E}_T \left[ Y^{(0)} \right] \right)^2 \right).$$

$\qquad \square$

*Example* 2.

| **Measure** | **Variance** |
|---|---|
| **Risk Difference (RD)** | $\frac{1}{\alpha} \left( \frac{\mathbb{E}_T \left[ r(X)(Y^{(1)})^2 \right]}{\pi} + \frac{\mathbb{E}_T \left[ r(X)(Y^{(0)})^2 \right]}{1 - \pi} - (\tau_{RD}^T)^2 \right)$ |
| **Risk Ratio (RR)** | $\frac{(\tau_{RR}^T)^2}{\alpha} \left( \frac{\mathbb{E}_T \left[ r(X)(Y^{(1)})^2 \right]}{\pi \mathbb{E}_T \left[ Y^{(1)} \right]^2} + \frac{\mathbb{E}_T \left[ r(X)(Y^{(0)})^2 \right]}{(1 - \pi)\mathbb{E}_T \left[ Y^{(0)} \right]^2} \right)$ |
| **Odds Ratio (OR)** | $\frac{(\tau_{OR}^T)^2}{\alpha} \left( \frac{\mathbb{E}_T \left[ r(X)(Y^{(1)})^2 \right]}{\pi (\mathbb{E}_T[Y^{(1)}])^2} + \frac{\mathbb{E}_T \left[ r(X)(Y^{(0)})^2 \right]}{(1 - \pi)(\mathbb{E}_T[Y^{(0)}])^2} - 1 \right)$ |

## B.3 Logistic weighted Horvitz-Thomson Estimator

**Proposition 9.** *Let $\psi_a = \mathbb{E}_T[Y^{(a)}]$ denote the target population mean potential outcome under treatment $a \in \{0,1\}$. Define the estimator*

$$\hat{\psi}_{a,wHT} = \frac{1}{n}\sum_{i=1}^{N} S_i \cdot r(X_i, \hat{\beta}_N) \cdot \frac{\mathbb{1}\{A_i = a\}Y_i}{\pi_a}, \quad \text{with} \quad r(x, \hat{\beta}_N) = \frac{n}{N-n} \cdot \frac{1 - \sigma(x, \hat{\beta}_N)}{\sigma(x, \hat{\beta}_N)}$$

*where $n = \sum_{i=1}^{N} S_i$ and $\hat{\beta}_N$ the maximum likelihood estimate (MLE) from logistic regression of the selection indicator $S$ on covariates $X$, and $\sigma(x, \beta) = (1 + e^{-x^\top \beta})^{-1}$ the logistic function. Then, under Assumption 1 to 3.1,*

$$\sqrt{N}\begin{pmatrix} \hat{\psi}_{1,wHT} - \psi_1 \\ \hat{\psi}_{0,wHT} - \psi_0 \end{pmatrix} \xrightarrow{d} \mathcal{N}(0, \Sigma_{wHT}), \quad \text{with} \quad \Sigma_{wHT} = \begin{pmatrix} V_{1,wHT} & C_{wHT} \\ C_{wHT} & V_{0,wHT} \end{pmatrix},$$

*where the asymptotic variance $V_{a,wHT}$ is given by*

$$V_{a,wHT} = \frac{\mathbb{E}_T[r(X)(Y^{(a)})^2]}{\alpha \pi_a} - \frac{\mathbb{E}_T[Y^{(a)}]^2}{1-\alpha} \tag{19}$$

$$- \mathbb{E}_T[Y^{(a)}V^\top]Q^{-1}\mathbb{E}_T[Y^{(a)}V] + 2\,\mathbb{E}_T[Y^{(a)}V^\top]Q^{-1}\mathbb{E}_T[\sigma(X)Y^{(a)}V], \tag{20}$$

$$\text{and} \quad C_{wHT} = -\frac{\mathbb{E}_T[Y^{(1)}]\mathbb{E}_T[Y^{(0)}]}{1-\alpha} \tag{21}$$

$$+ \mathbb{E}_T[Y^{(1)}V^\top]Q^{-1}\mathbb{E}_T[\sigma(X)Y^{(0)}V] \tag{22}$$

$$+ \mathbb{E}_T[Y^{(0)}V^\top]Q^{-1}\mathbb{E}_T[\sigma(X)Y^{(1)}V] \tag{23}$$

$$- \mathbb{E}_T[Y^{(1)}V^\top]Q^{-1}\mathbb{E}_T[Y^{(0)}V] \tag{24}$$

*with $V = (1, X)$ and $Q = (1-\alpha)\mathbb{E}_T[\sigma(X,\beta)VV^\top]$.*

*Proof.* Let $Z = (S, X, S \times A, S \times Y)$ and define the parameter vector $\theta = (\theta_0, \theta_1, \theta_2, \beta)$. We define the estimating function $\lambda(Z, \theta)$ and the estimator $\hat{\theta}_N$ as follows:

$$\lambda(Z, \theta) = \begin{pmatrix} \frac{1}{1-\theta_2}\frac{1-\sigma(X,\beta)}{\sigma(X,\beta)}\frac{S\mathbb{1}\{A=0\}Y}{\mathbb{P}_S(A=0)} - \theta_0 \\ \frac{1}{1-\theta_2}\frac{1-\sigma(X,\beta)}{\sigma(X,\beta)}\frac{S\mathbb{1}\{A=1\}Y}{\mathbb{P}_S(A=1)} - \theta_1 \\ S - \theta_2 \\ V(S - \sigma(X,\beta)) \end{pmatrix}, \quad \hat{\theta}_N = \begin{pmatrix} \hat{\psi}_{0,\text{wHT}} \\ \hat{\psi}_{1,\text{wHT}} \\ \hat{\alpha} := \frac{1}{N}\sum_{i=1}^{N} S_i \\ \hat{\beta}_N \end{pmatrix}.$$

Note that the reweighting function can be expressed as:

$$r(X_i, \hat{\beta}_N) = \frac{1 - \sigma(X_i, \hat{\beta}_N)}{\sigma(X_i, \hat{\beta}_N)} \cdot \frac{\hat{\alpha}}{1 - \hat{\alpha}}.$$

Thus, we can rewrite the estimator as:

$$\hat{\psi}_{a,\text{wHT}} = \frac{1}{N\hat{\alpha}}\sum_{j=1}^{N} S_j r(X_j, \hat{\beta}_N)\frac{\mathbb{1}\{A_j = a\}Y_j}{\pi_a} \tag{25}$$

$$= \frac{1}{N}\sum_{j=1}^{N} S_j \frac{1 - \sigma(X_j, \hat{\beta}_N)}{\sigma(X_j, \hat{\beta}_N)(1 - \hat{\alpha})}\frac{\mathbb{1}\{A_j = a\}Y_j}{\pi_a}. \tag{26}$$

Furthermore, the log-likelihood function of $\beta$ is:

$$-\ln L_N(\beta) = -\sum_{i=1}^{N} s_i \log \sigma(X_i; \beta) + (1 - s_i)\log(1 - \sigma(X_i; \beta))$$

where $\sigma(X; \beta) = (1 + \exp(-X^\top \beta_1 - \beta_0))^{-1}$. Simple calculations show that

$$\frac{\partial \ln L_N}{\partial \beta_0}(\beta) = -\sum_{i=1}^{N}(S_i - \sigma(X_i; \beta)) \quad \text{and} \quad \frac{\partial \ln L_N}{\partial \beta_1}(\beta) = -\sum_{i=1}^{N} X_i(S_i - \sigma(X_i; \beta)).$$

Recalling that $V = (1, X)$, by definition of the MLE $\hat{\boldsymbol{\beta}}_N$, we get

$$\sum_{i=1}^{N} \lambda_3(Z_i, \hat{\theta}_N) = \sum_{i=1}^{N} V_i(S_i - \sigma(X_i; \hat{\boldsymbol{\beta}}_N)) = 0.$$

Gathering the previous equality and Equation (26), we obtain

$$\sum_{i=1}^{N} \lambda(Z_i, \hat{\boldsymbol{\theta}}_n) = 0, \tag{27}$$

which proves that $\hat{\theta}_N$ is an M-estimator of type $\lambda$. Furthermore, letting $\theta_\infty = (\mathbb{E}_T\left[Y^{(0)}\right], \mathbb{E}_T\left[Y^{(1)}\right], \alpha, \beta_\infty)$, we can compute the following quantities:

$$\mathbb{E}\left[\frac{1 - \sigma(X)}{\sigma(X)(1 - \alpha)} \frac{S\mathbb{1}\{A = a\}Y}{\pi_a}\right] = \mathbb{E}\left[\frac{r(X)}{\alpha} \frac{S\mathbb{1}\{A = a\}Y^{(a)}}{\pi_a}\right]$$

$$= \frac{1}{\pi_a \alpha} \mathbb{P}(S = 1)\mathbb{E}\left[r(X)\mathbb{1}\{A = a\}Y^{(a)}|S = 1\right]$$

$$= \frac{1}{\pi_a} \mathbb{E}_S\left[r(X)\mathbb{1}\{A = a\}Y^{(a)}\right]$$

$$= \mathbb{E}_S\left[r(X)Y^{(a)}\right] \text{ since } r(X) = P_T(X)/P_S(X)$$

$$= \mathbb{E}_T\left[Y^{(a)}\right].$$

Thus, we have $\mathbb{E}\left[\lambda_0(Z, \theta_\infty)\right] = \mathbb{E}\left[\lambda_1(Z, \theta_\infty)\right] = 0$. Besides,

$$\begin{aligned}
\mathbb{E}\left[\lambda_3(Z, \theta_\infty)\right] &= \mathbb{E}\left[V(S - \sigma(X))\right] \\
&= \mathbb{E}\left[V \cdot \mathbb{E}\left[S - \sigma(X) \mid X\right]\right] && \text{(Law of Total Probability)} \\
&= \mathbb{E}\left[V \cdot (\mathbb{E}\left[S \mid X\right] - \sigma(X))\right] && (\sigma(X) \text{ is a function of } X) \\
&= 0 && \text{(Definition of } \sigma(X))
\end{aligned}$$

Therefore, we have

$$\mathbb{E}\left[\lambda(Z, \theta_\infty)\right] = 0. \tag{28}$$

Now, we show that $\theta_\infty$ defined above is the unique value that satisfies (28). We directly have that $\theta_2 = \alpha$. Let

$$L(\beta) = -\mathbb{E}\left[S \ln\left(\sigma(X, \beta)\right) + (1 - S) \ln\left(1 - \sigma(X, \beta)\right)\right].$$

A direct calculation shows that

$$\nabla_\beta L(\beta) = \mathbb{E}\left[V\left(\sigma(X, \beta) - S\right)\right] \quad \text{and} \quad \nabla_\beta^2 L(\beta) = \mathbb{E}\left[VV^\top \sigma(X, \beta)\left(1 - \sigma(X, \beta)\right)\right].$$

Since $\mathbb{E}[S \mid X] = \sigma(X, \beta_\infty)$, and $V = (1, X)$,

$$\nabla_\beta L(\beta_\infty) = \mathbb{E}\left[V\left(e(X, \beta_\infty) - S\right)\right] = 0$$

making $\beta_\infty$ a stationary point. Furthermore, using overlap we have $\sigma(X, \beta)\left(1 - \sigma(X, \beta)\right) \geq \eta^2$ therefore $\forall v \in \mathbb{R}^{p+1}$:

$$\begin{aligned}
v^\top \nabla_\beta^2 L(\beta) v &= \mathbb{E}\left[||V^\top v||_2^2 \, \sigma(X, \beta)\left(1 - \sigma(X, \beta)\right)\right] \\
&\geq \eta^2 \mathbb{E}\left[||V^\top v||_2^2\right] \\
&\geq \eta^2 v^\top \mathbb{E}\left[VV^\top\right] v.
\end{aligned}$$

Since we assumed that $\mathbb{E}[X\, X^\top]$ is positive-definite, the Hessian $\nabla^2_\beta L(\beta)$ is positive-definite, so $L(\beta)$ is strictly convex. Hence there is a unique global minimizer of $L(\beta)$; since $\beta_\infty$ is a critical point, it must be that unique minimizer. Consequently, any solution to

$$\mathbb{E}\Big[V\big(\sigma(X,\beta)-S\big)\Big]=0$$

must equal $\beta_\infty$. Since $\beta,\theta_2$ are now fixed and since the first two components of $\psi$ are linear with respect to $\theta_0$ and $\theta_1$, $\theta_\infty$ is the only value satisfying (28). We want to show that for every $\theta$ in a neighborhood of $\theta_\infty$, all the components of the second derivatives are integrable for all $k \in \{0,\dots,3\}$:

$$\left|\frac{\partial^2}{\partial^2\theta}\lambda_k(z,\theta)\right|$$

While this holds trivially for most components, the integrability of the following specific terms requires closer attention:

$$\left|\frac{\partial^2}{\partial\theta_2\partial\beta}\lambda_a(z,\theta)\right|,\qquad \left|\frac{\partial^2}{\partial\beta\partial\beta}\lambda_a(z,\theta)\right|,\qquad \left|\frac{\partial^2}{\partial\beta\partial\beta}\lambda_3(z,\theta)\right|\quad\text{for all }a\in\{0,1\}.$$

First, consider the mixed partial derivative with respect to $\theta_2$ and $\beta$:

$$\left|\frac{\partial^2}{\partial\theta_2\partial\beta}\lambda_a(z,\theta)\right|=\frac{1}{(1-\theta_2)^2}\cdot\frac{1-\sigma(X,\beta)}{\sigma(X,\beta)}\cdot\frac{S\mathbb{1}\{A=a\}Y}{\pi_a}V^\top.$$

For each coordinate $i \in \{1,\dots,d\}$, the expectation is bounded as follows:

$$\mathbb{E}\left[\left.\left|\frac{\partial^2}{\partial\theta_2\partial\beta}\lambda_a(z,\theta)\right|\right|_i\right]=\frac{1}{(1-\theta_2)^2}\mathbb{E}\left[\frac{1-\sigma(X,\beta)}{\sigma(X,\beta)}\cdot\frac{S\mathbb{1}\{A=a\}Y}{\pi_a}V_i\right]$$

$$\leq\frac{1}{(1-\theta_2)^2}\left(\mathbb{E}\left[\left(\frac{1-\sigma(X,\beta)}{\sigma(X,\beta)}\right)^2\cdot\frac{S\mathbb{1}\{A=a\}Y^2}{\pi_a^2}\right]\cdot\mathbb{E}[V_i^2]\right)^{1/2},$$

using the Cauchy–Schwarz inequality. The first expectation is finite under the assumption that $Y$ is square-integrable, and due to the exponential tail behavior of the logistic function. The sub-Gaussianity of $X$ implies that all moments of $V$ are finite, ensuring integrability of $\mathbb{E}[V_i^2]$.

Second, consider the pure second derivative with respect to $\beta$:

$$\left|\frac{\partial^2}{\partial\beta\partial\beta}\lambda_a(z,\theta)\right|=\frac{1}{1-\theta_2}\cdot\frac{1-\sigma(X,\beta)}{\sigma(X,\beta)}\cdot\frac{S\mathbb{1}\{A=a\}Y}{\pi_a}VV^\top.$$

Each entry of this matrix takes the form $C\cdot V_iV_j$ for some random coefficient $C$, and the integrability of these entries follows from the same reasoning as above.

Third, the second derivative of $\lambda_3$ is given by

$$\left|\frac{\partial^2}{\partial\beta\partial\beta}\lambda_3(z,\theta)\right|=|V_kV_lV_m\cdot\sigma(X,\beta)(1-\sigma(X,\beta))(1-2\sigma(X,\beta))|\leq|V_kV_lV_m|.$$

By applying Hölder's inequality, the following bound is obtained:

$$\mathbb{E}\left[|V_kV_lV_m|\right]\leq\mathbb{E}[V_k^2]^{1/2}\cdot\mathbb{E}[V_l^4]^{1/4}\cdot\mathbb{E}[V_m^4]^{1/4},$$

which is finite due to the sub-Gaussianity of $X$. Consequently, each second derivative component

$$\left|\frac{\partial^2}{\partial^2\theta}\lambda_k(z,\theta)\right|$$

is integrable for all $k \in \{0,\dots,3\}$, in the neighborhood of $\theta_\infty$. Define

$$A\left(\theta_\infty\right)=\mathbb{E}\left[\left.\frac{\partial\lambda}{\partial\theta}\right|_{\theta=\theta_\infty}\right]\quad\text{and}\quad B(\theta_\infty)=\mathbb{E}\left[\lambda(Z,\theta_\infty)\lambda(Z,\theta_\infty)^T\right].$$

Next, we verify the conditions of Theorem 7.2 in [42]. To do so, we compute $A(\theta_\infty)$ and $B(\theta_\infty)$. Since

$$\frac{\partial \lambda}{\partial \theta}(Z,\theta) = \begin{pmatrix} -1 & 0 & \frac{1}{(1-\theta_2)^2}\frac{1-\sigma(X,\beta)}{\sigma(X,\beta)}\frac{S\mathbb{1}\{A=0\}Y}{\mathbb{P}_S(A=0)} & \frac{-1}{1-\theta_2}\frac{1-\sigma(X,\beta)}{\sigma(X,\beta)}\frac{S\mathbb{1}\{A=0\}Y}{\mathbb{P}_S(A=0)}V^\top \\ 0 & -1 & \frac{1}{(1-\theta_2)^2}\frac{1-\sigma(X,\beta)}{\sigma(X,\beta)}\frac{S\mathbb{1}\{A=1\}Y}{\mathbb{P}_S(A=1)} & \frac{-1}{1-\theta_2}\frac{1-\sigma(X,\beta)}{\sigma(X,\beta)}\frac{S\mathbb{1}\{A=1\}Y}{\mathbb{P}_S(A=1)}V^\top \\ 0 & 0 & -1 & 0 \\ 0 & 0 & 0 & -\sigma(X,\beta)(1-\sigma(X,\beta))VV^\top \end{pmatrix},$$
(29)

We obtain with $\theta_\infty = (\mathbb{E}_T[Y^{(0)}], \mathbb{E}_T[Y^{(1)}], \alpha, \beta_\infty)$

$$A(\theta_\infty) = \begin{pmatrix} -1 & 0 & \frac{\mathbb{E}_T[Y^{(0)}]}{1-\alpha} & -\mathbb{E}_T[Y^{(0)}V^\top] \\ 0 & -1 & \frac{\mathbb{E}_T[Y^{(1)}]}{1-\alpha} & -\mathbb{E}_T[Y^{(1)}V^\top] \\ 0 & 0 & -1 & 0 \\ 0 & 0 & 0 & -Q \end{pmatrix},$$

where $Q = \mathbb{E}[\sigma(X,\beta_\infty)(1-\sigma(X,\beta_\infty))VV^\top]$, which using Schur complement leads to:

$$A(\theta_\infty)^{-1} = \begin{pmatrix} -1 & 0 & -\frac{\mathbb{E}_T[Y^{(0)}]}{1-\alpha} & \mathbb{E}_T[Y^{(0)}V^\top]Q^{-1} \\ 0 & -1 & -\frac{\mathbb{E}_T[Y^{(1)}]}{1-\alpha} & \mathbb{E}_T[Y^{(1)}V^\top]Q^{-1} \\ 0 & 0 & -1 & 0 \\ 0 & 0 & 0 & -Q^{-1} \end{pmatrix}$$

Regarding $B(\theta_\infty)$, elementary calculations show that

$$B(\theta_\infty) = \begin{pmatrix} \frac{\mathbb{E}_T[r(X)(Y^{(0)})^2]}{\alpha\mathbb{P}_S(A=0)} - \mathbb{E}_T[Y^{(0)}]^2 & -\mathbb{E}_T[Y^{(0)}]\mathbb{E}_T[Y^{(1)}] & (1-\alpha)\mathbb{E}_T[Y^{(0)}] & \mathbb{E}_T[(1-\sigma(X))V^\top Y^{(a)}] \\ -\mathbb{E}_T[Y^{(0)}]\mathbb{E}_T[Y^{(1)}] & \frac{\mathbb{E}_T[r(X)(Y^{(1)})^2]}{\alpha\mathbb{P}_S(A=1)} - \mathbb{E}_T[Y^{(1)}]^2 & (1-\alpha)\mathbb{E}_T[Y^{(1)}] & \mathbb{E}_T[(1-\sigma(X))V^\top Y^{(1)}] \\ (1-\alpha)\mathbb{E}_T[Y^{(0)}] & (1-\alpha)\mathbb{E}_T[Y^{(1)}] & \alpha(1-\alpha) & (1-\alpha)\mathbb{E}_T[\sigma(X)V^\top] \\ \mathbb{E}_T[(1-\sigma(X))VY^{(0)}] & \mathbb{E}_T[(1-\sigma(X))VY^{(1)}] & (1-\alpha)\mathbb{E}_T[\sigma(X)V] & Q \end{pmatrix}.$$

Based on the previous calculations, we have:

- $\lambda(z,\theta)$ and its first two partial derivatives with respect to $\theta$ exist for all $z$ and for all $\theta$ in the neighborhood of $\theta_\infty$.

- For each $\theta$ in the neighborhood of $\theta_\infty$, we have for all $k \in \{0,3\} \left|\frac{\partial^2}{\partial^2\theta}\lambda_k(z,\theta)\right|$ is integrable.

- $A(\theta_\infty)$ exists and is nonsingular.

- $B(\theta_\infty)$ exists and is finite.

We also have

$$\sum_{i=1}^n \lambda(Z_i, \hat{\theta}_N) = 0 \quad \text{and} \quad \hat{\theta}_N \xrightarrow{p} \theta_\infty.$$

Then the conditions of Theorem 7.2 in Stefanski and Boos [42] are satisfied, and we can conclude that

$$\sqrt{n}\left(\hat{\theta}_N - \theta_\infty\right) \xrightarrow{d} \mathcal{N}\left(0, A(\theta_\infty)^{-1}B(\theta_\infty)(A(\theta_\infty)^{-1})^\top\right),$$

Letting $\nu_0^\top$ and $\nu_1^\top$ be respectively the first and second row of $A(\theta_\infty)^{-1}$:

$$\nu_0^\top = \left(-1, 0, -\frac{\mathbb{E}_T[Y^{(0)}]}{1-\alpha}, \mathbb{E}_T[Y^{(0)}V^\top]Q^{-1}\right),$$

and

$$\nu_1^\top = \left(0, -1, -\frac{\mathbb{E}_T[Y^{(1)}]}{1-\alpha}, \mathbb{E}_T[Y^{(1)}V^\top]Q^{-1}\right).$$

Expanding the quadratic form explicitly, and using Lemma 1, we get:

$$V_{a,\text{wHT}} = \nu_a^\top B(\theta_\infty)\nu_a = \frac{\mathbb{E}_T[r(X)(Y^{(a)})^2]}{\alpha\pi_a} - \frac{\mathbb{E}_T[Y^{(a)}]^2}{1-\alpha}$$
$$- \mathbb{E}_T[Y^{(a)}V]^\top Q^{-1}\mathbb{E}_T[Y^{(a)}V] + 2\mathbb{E}_T[Y^{(a)}V]^\top Q^{-1}\mathbb{E}_T[\sigma(X)VY^{(a)}].$$

and:

$$C_{\text{wHT}} = \nu_a^\top B(\theta_\infty)\nu_{1-a} = -\frac{\mathbb{E}_T[Y^{(1)}]\mathbb{E}_T[Y^{(0)}]}{1-\alpha} + \mathbb{E}_T[Y^{(1)}V^\top]Q^{-1}\mathbb{E}_T[Y^{(0)}V]$$
$$- \mathbb{E}_T[Y^{(1)}V^\top]Q^{-1}\mathbb{E}_T[(1-\sigma(X))Y^{(0)}V]$$
$$- \mathbb{E}_T[Y^{(0)}V^\top]Q^{-1}\mathbb{E}_T[(1-\sigma(X))Y^{(1)}V]$$

$\square$

**Lemma 1.** *We have* $\mathbb{E}_T[\sigma(X)V]^\top Q^{-1} = \frac{u_1(d+1)^\top}{1-\alpha}$ *and* $Q = (1-\alpha)\mathbb{E}_T\left[\sigma(X,\beta)VV^\top\right]$.

*Proof.*

$$Q = \mathbb{E}\left[\sigma(X,\beta)(1-\sigma(X,\beta))VV^\top\right]$$
$$= \mathbb{P}(S=1)\mathbb{E}_S\left[\sigma(X,\beta)(1-\sigma(X,\beta))VV^\top\right] + \mathbb{P}(S=0)\mathbb{E}_T\left[\sigma(X,\beta)(1-\sigma(X,\beta))VV^\top\right]$$
$$= (1-\alpha)\mathbb{E}_S\left[\sigma(X,\beta)^2 r(X)VV^\top\right] + (1-\alpha)\mathbb{E}_T\left[\sigma(X,\beta)(1-\sigma(X,\beta))VV^\top\right]$$
$$= (1-\alpha)\mathbb{E}_T\left[\sigma(X,\beta)^2 VV^\top\right] + (1-\alpha)\mathbb{E}_T\left[\sigma(X,\beta)(1-\sigma(X,\beta))VV^\top\right]$$
$$= (1-\alpha)\mathbb{E}_T\left[\sigma(X,\beta)VV^\top\right]$$

Therefore using using the block inverse matrix formula:

$$Q^{-1} = \frac{1}{1-\alpha}\begin{pmatrix} S^{-1} & -S^{-1}\mathbb{E}_T[\sigma(X)X]^\top P^{-1} \\ -S^{-1}P^{-1}\mathbb{E}_T[\sigma(X)X] & P^{-1} + P^{-1}\mathbb{E}_T[\sigma(X)X]\mathbb{E}_T[\sigma(X)X]^\top P^{-1}S^{-1} \end{pmatrix},$$

where $P = \mathbb{E}_T[\sigma(X)XX^\top]$ and $S = \mathbb{E}_T[\sigma(X)] - \mathbb{E}_T[\sigma(X)X]^\top P^{-1}\mathbb{E}_T[\sigma(X)X]$.

Expanding $\begin{bmatrix} \mathbb{E}_T[\sigma(X)] & \mathbb{E}_T[\sigma(X)X]^\top \end{bmatrix}(1-\alpha)Q^{-1}$, we get:

$$\mathbb{E}_T[\sigma(X)]S^{-1} - S^{-1}\mathbb{E}_T[\sigma(X)X]^\top P^{-1}\mathbb{E}_T[\sigma(X)X] = S^{-1}(\underbrace{\mathbb{E}_T[\sigma(X)] - \mathbb{E}_T[\sigma(X)X]^\top P^{-1}\mathbb{E}_T[\sigma(X)X]}_{S}) = 1$$

and

$$-\mathbb{E}_T[\sigma(X)]S^{-1}\mathbb{E}_T[\sigma(X)X]^\top P^{-1} + \mathbb{E}_T[\sigma(X)X]^\top P^{-1} + \mathbb{E}_T[\sigma(X)X]^\top P^{-1}\mathbb{E}_T[\sigma(X)X]\mathbb{E}_T[\sigma(X)X]^\top P^{-1}S^{-1} = 0$$

Hence:

$$\mathbb{E}_T[\sigma(X)V]^\top Q^{-1} = \frac{u_1(d+1)^\top}{1-\alpha}$$

$\square$

**Proposition 10** (asymptotic normality of weighted Horvitz-Thompson estimator)**.** *Let* $\sigma(x,\beta) = \left(1+\exp(-x^\top\beta_1 - \beta_0)\right)^{-1}$ *denote the logistic function, where* $\hat{\beta}_N$ *is the maximum likelihood estimate (MLE) obtained from logistic regression of the selection indicator* $S$ *on covariates* $X$. *Define the estimated density ratio as:*

$$r(x,\hat{\beta}_N) = \frac{n}{N-n} \cdot \frac{1-\sigma(x,\hat{\beta}_N)}{\sigma(x,\hat{\beta}_N)}, \qquad \text{with} \quad n = \sum_{i=1}^N S_i.$$

*Let* $\hat{\tau}_{\Phi,wHT}$ *denote the weighted Horvitz-Thompson estimator constructed using the estimated ratio* $r(x,\hat{\beta}_N)$. *Then, under Assumption 1 to 3.1:*

$$\sqrt{N}\left(\hat{\tau}_{\Phi,wHT} - \tau_\Phi^T\right) \xrightarrow{d} \mathcal{N}\left(0, V_{\Phi,wHT}\right).$$

*Proof.* We begin by observing that the estimator of interest can be written as smooth transformations of its vector-valued estimator:

$$\hat{\tau}_{\Phi,\text{wHT}} = \Phi(\hat{\psi}_{1,\text{wHT}}, \hat{\psi}_{0,\text{wHT}}).$$

By Propositions 9, the pair $(\hat{\psi}_{1,\text{wHT}}, \hat{\psi}_{0,\text{wHT}})$ is jointly asymptotically normal. Specifically,

$$\sqrt{N} \begin{pmatrix} \hat{\psi}_{1,\text{wHT}} - \psi_1 \\ \hat{\psi}_{0,\text{wHT}} - \psi_0 \end{pmatrix} \xrightarrow{d} \mathcal{N}\left(0, \Sigma_{\text{wHT}}\right),$$

where $\Sigma_{\text{wHT}}$ is the asymptotic covariance matrices, with entries determined by the variances and covariances of the components $\hat{\psi}_{1,\text{wHT}}$ and $\hat{\psi}_{0,\text{wHT}}$. Since $\Phi : \mathbb{R}^2 \to \mathbb{R}$ is assumed to be a smooth function, we can apply the delta method to each estimator. Let $\nabla \Phi$ denote the gradient of $\Phi$, evaluated at the true parameter vector $(\psi_1, \psi_0)$. Then:

$$\sqrt{N}(\hat{\tau}_{\Phi,\text{wHT}} - \tau_\Phi^T) \xrightarrow{d} \mathcal{N}(0, V_{\Phi,\text{wHT}}),$$

where the asymptotic variance is given by the quadratic form:

$$V_{\Phi,\text{wHT}} = \nabla \Phi^\top \Sigma_{\text{wHT}} \nabla \Phi.$$

$\square$

**Proposition 11.** *Let $\psi_a = \mathbb{E}_T[Y(a)]$ denote the target population mean potential outcome under treatment $a \in \{0, 1\}$. Define the estimator*

$$\hat{\psi}_{a,N} = \frac{1}{n} \sum_{i=1}^N S_i \cdot r(X_i, \hat{\beta}_N) \cdot \frac{\mathbb{1}\{A_i = a\} Y_i}{\hat{\pi}_a}, \quad \text{with} \quad r(x, \hat{\beta}_N) = \frac{n}{N - n} \cdot \frac{1 - \sigma(x, \hat{\beta}_N)}{\sigma(x, \hat{\beta}_N)},$$

*where*

$$\hat{\pi}_a = \frac{1}{n} \sum_{S_i = 1} \mathbb{1}\{A_i = a\}, \quad n = \sum_{i=1}^N S_i,$$

*$\hat{\beta}_N$ is the MLE from logistic regression of $S$ on $X$, and $\sigma(x, \beta) = (1 + e^{-x^\top \beta})^{-1}$ is the logistic function. Then, under regularity conditions,*

$$\sqrt{N} \begin{bmatrix} \hat{\psi}_{1,N} - \psi_1 \\ \hat{\psi}_{0,N} - \psi_0 \end{bmatrix} \xrightarrow{d} \mathcal{N}\left(0, \Sigma_N\right), \quad \text{with} \quad \Sigma_N = \begin{bmatrix} V_{1,N} & C_N \\ C_N & V_{0,N} \end{bmatrix},$$

*where*

$$\begin{aligned}
V_{a,N} &= \frac{\mathbb{E}_T\left[r(X)\left(Y^{(a)}\right)^2\right]}{\alpha \, \pi_a} + \mathbb{E}_T[Y^{(a)}]^2 \left(\frac{2}{\alpha} - \frac{1}{\alpha(1 - \alpha)} - \frac{1}{\alpha \pi_a}\right) \\
&\quad - \mathbb{E}_T[Y^{(a)} V]^\top Q^{-1} \mathbb{E}_T[Y^{(a)} V] + 2\, \mathbb{E}_T[Y^{(a)} V]^\top Q^{-1} \mathbb{E}_T[\sigma(X) V Y^{(a)}], \\
C_N &= \mathbb{E}_T[Y^{(a)}] \mathbb{E}_T[Y^{(1-a)}] \cdot \frac{1 - 2\alpha}{\alpha(1 - \alpha)} \\
&\quad - \mathbb{E}_T[Y^{(a)} V^\top] Q^{-1} \mathbb{E}_T[(1 - \sigma(X)) Y^{(1-a)} V] \\
&\quad - \mathbb{E}_T[Y^{(1-a)} V^\top] Q^{-1} \mathbb{E}_T[(1 - \sigma(X)) Y^{(a)} V] \\
&\quad + \mathbb{E}_T[Y^{(a)} V^\top] Q^{-1} \mathbb{E}_T[Y^{(1-a)} V],
\end{aligned}$$

*with $V = (1, X) \in \mathbb{R}^{p+1}$ and*

$$Q = (1 - \alpha)\, \mathbb{E}_T[\sigma(X, \beta)\, V V^\top].$$

*Proof.* We follow the same initial setup and notation as in the proof of the Re-weighted Horvitz-Thomson estimator. The key difference is the empirical estimation of $\pi_a$, introducing an additional

estimating equation for $\theta_3$. The remainder of the proof—existence, uniqueness, M-estimation structure, and regularity conditions—follows identically. Define here:

$$
\lambda(Z,\boldsymbol{\theta}) =
\begin{bmatrix}
\frac{\theta_2}{1-\theta_2}\frac{1-\sigma(X,\beta)}{\sigma(X,\beta)}\frac{S\,\mathbb{1}\{A=0\}\,Y}{\theta_3} - \theta_0 \\
\frac{\theta_2}{1-\theta_2}\frac{1-\sigma(X,\beta)}{\sigma(X,\beta)}\frac{S\,\mathbb{1}\{A=1\}\,Y}{\theta_4} - \theta_1 \\
S - \theta_2 \\
S\mathbb{1}\{A=0\} - \theta_3 \\
S\mathbb{1}\{A=1\} - \theta_4 \\
V(S - \sigma(X,\beta))
\end{bmatrix}
\qquad
\hat{\boldsymbol{\theta}}_N =
\begin{bmatrix}
\hat{\psi}_{0,\mathrm{N}} \\
\hat{\psi}_{1,\mathrm{N}} \\
\hat{\alpha} := \frac{1}{N}\sum_{i=1}^N S_i \\
\frac{1}{N}\sum_{i=1}^N S_i\mathbb{1}\{A_i=0\} \\
\frac{1}{N}\sum_{i=1}^N S_i\mathbb{1}\{A_i=1\} \\
\hat{\beta}_N
\end{bmatrix}
$$

We now rewrite the estimator using this notation. Recall

$$
r(X_j, \hat{\beta}_N) = \frac{1 - \sigma(X_j, \hat{\beta}_N)}{\sigma(X_j, \hat{\beta}_N)} \cdot \frac{\hat{\alpha}}{1 - \hat{\alpha}}.
$$

Then:

$$
\hat{\psi}_{a,\hat{\pi}} = \frac{1}{N\hat{\alpha}} \sum_{j=1}^N S_j r(X_j, \hat{\beta}_N) \cdot \frac{\mathbb{1}\{A_j = a\}Y_j}{\hat{\pi}_a},
$$

and

$$
\hat{\pi}_a = \frac{1}{N\hat{\alpha}} \sum_{i=1}^N S_i \mathbb{1}\{A_i = a\}.
$$

Substituting in gives:

$$
\hat{\psi}_{a,\hat{\pi}} = \frac{1}{N} \sum_{j=1}^N S_j \cdot \frac{1 - \sigma(X_j, \hat{\beta}_N)}{\sigma(X_j, \hat{\beta}_N)} \cdot \frac{\hat{\alpha}}{1 - \hat{\alpha}} \cdot \frac{\mathbb{1}\{A_j = a\}Y_j}{\frac{1}{N}\sum_{i=1}^N S_i \mathbb{1}\{A_i = a\}}.
$$

Let $A(\theta_\infty)$ be the Jacobian of $\lambda(Z,\theta)$ at the population limit $\theta_\infty$, and $B(\theta_\infty)$ the corresponding covariance matrix where

$$
\theta_\infty = [\mathbb{E}_T[Y^{(0)}], \mathbb{E}_T[Y^{(1)}], \alpha, \alpha\pi_0, \alpha\pi_1, \beta_\infty]^T
$$

Let

$$
Q = \mathbb{E}_T[\sigma(X, \beta_\infty)(1 - \sigma(X, \beta_\infty))VV^\top].
$$

The inverse Jacobian block $A(\theta_\infty)^{-1}$ is block lower-triangular, with expressions for rows $\nu_0^\top$, $\nu_1^\top$ given by:

$$
\nu_a^\top = \begin{bmatrix} -\mathbb{1}\{a = 0\} & -\mathbb{1}\{a = 1\} & -\frac{\mathbb{E}_T[Y^{(a)}]}{(1-\alpha)\alpha} & \frac{\mathbb{E}_T[Y^{(a)}]}{\alpha\pi_a} & 0 & \mathbb{E}_T[VY^{(a)}]^\top Q^{-1} \end{bmatrix}.
$$

We then compute the asymptotic variance as

$$
V_{a,\hat{\pi}} = \nu_a^\top B(\theta_\infty)\nu_a, \quad C_{\hat{\pi}} = \nu_0^\top B(\theta_\infty)\nu_1,
$$

where the expansion of $\nu_a^\top B(\theta_\infty)\nu_b$ yields the desired closed-form expressions for $V_{a,\mathrm{N}}$ and $C_{\mathrm{N}}$ stated in the proposition. $\qquad\square$

## B.4 Weighted and transported G-formula

**Proposition 12.** *Let $\psi_a = \mathbb{E}_T[Y(a)]$ denote the target population mean potential outcome under treatment $a \in \{0,1\}$. Define the oracle estimators*

$$
\psi_{a,\mathrm{wG}}^* = \frac{1}{n}\sum_{i=1}^N S_i \cdot r(X_i) \cdot \mu_{(a)}^S(X_i), \quad \text{and} \quad \psi_{a,\mathrm{tG}}^* = \frac{1}{N-n}\sum_{i=1}^N (1 - S_i) \cdot \mu_{(a)}^S(X_i)
$$

*where $r(X)$ denotes the density ratio between the target and source covariate distributions and $n = \sum_{i=1}^N S_i$ is the number of units in the source sample. Then, under Assumption 1 to 3,*

$$
\sqrt{N}\left(\psi_{a,\mathrm{tG}}^* - \psi_a\right) \xrightarrow{d} \mathcal{N}(0, V_{a,\mathrm{tG}}^*),
$$

*Furthermore under Section 3.1:*

$$\sqrt{N}\left(\psi_{a,\mathrm{wG}}^* - \psi_a\right) \xrightarrow{d} \mathcal{N}(0, V_{a,\mathrm{wG}}^*),$$

*where the asymptotic variances are given by*

$$V_{a,\mathrm{wG}}^* = \frac{1}{\alpha}\left(\mathbb{E}_T\left[r(X)(\mu_{(a)}^S(X))^2\right] - \mathbb{E}_T[Y(a)]^2\right)$$

*and*

$$V_{a,\mathrm{tG}}^* = \frac{1}{1-\alpha}\left(\mathbb{E}_T\left[\left(\mu_{(a)}^S(X)\right)^2\right] - \mathbb{E}_T[Y^{(a)}]^2\right).$$

*Proof.* **Transported G-formula.** Define

$$Z_i = (1 - S_i)\cdot\mu_{(a)}^S(X),$$

so that

$$\psi_{a,\mathrm{tG}}^* = \frac{\frac{1}{N}\sum_{i=1}^N Z_i}{\frac{1}{N}\sum_{i=1}^N 1 - S_i}.$$

We first compute the expectations of $Z_i$ and $S_i$. By the definition of $Z_i$ and using that $S_i \sim$ Bernoulli$(\alpha)$,

$$\begin{aligned}
\mathbb{E}[Z_i] &= \mathbb{E}\left[(1-S)\cdot\mu_{(a)}^S(X)\right]\\
&= (1-\alpha)\,\mathbb{E}_T\left[\mu_{(a)}^S(X)\right]\\
&= (1-\alpha)\psi_a,
\end{aligned}$$

where we used the consistency assumption $Y = Y^{(A)}$, the randomization of $A$, and the density ratio property. Since the $(Z_i, S_i)$ for all $i = 1, \ldots, N$ are i.i.d., we can apply the multivariate central limit theorem to

$$\left(\frac{1}{N}\sum_{i=1}^N Z_i, \quad \frac{1}{N}\sum_{i=1}^N 1 - S_i\right),$$

to obtain

$$\sqrt{N}\left(\begin{pmatrix}\frac{1}{N}\sum_{i=1}^N Z_i\\\frac{1}{N}\sum_{i=1}^N 1 - S_i\end{pmatrix} - \begin{pmatrix}\mathbb{E}[Z_i]\\\mathbb{E}[1 - S_i]\end{pmatrix}\right) \xrightarrow{d} \mathcal{N}(0, \Sigma),$$

where

$$\Sigma = \begin{pmatrix}\mathbb{V}[Z_i] & \mathrm{Cov}(Z_i, S_i)\\\mathrm{Cov}(S_i, Z_i) & \mathbb{V}[S_i]\end{pmatrix}.$$

since $\psi_{a,\mathrm{tG}}^*$ can be written as

$$\psi_{a,\mathrm{tG}}^* = \frac{\frac{1}{N}\sum_{i=1}^N Z_i}{\frac{1}{N}\sum_{i=1}^N 1 - S_i},$$

we apply the Delta method to the map $h : (x, y) \mapsto x/y$, whose gradient evaluated at $((1-\alpha)\psi_a, 1-\alpha)$ is

$$u = \nabla h((1-\alpha)\psi_a, 1-\alpha) = \left(\frac{1}{1-\alpha}, -\frac{\psi_a}{1-\alpha}\right).$$

Thus,

$$\sqrt{N}\left(\psi_{a,\mathrm{tG}}^* - \psi_a\right) \xrightarrow{d} \mathcal{N}(0, V_{a,\mathrm{tG}}^*),$$

where

$$V_{a,\mathrm{tG}}^* = u^\top \Sigma u.$$

Expanding, we obtain

$$V_{a,\mathrm{tG}}^* = \frac{1}{(1-\alpha)^2}\mathbb{V}[Z_i] + \frac{(\psi_a)^2}{(1-\alpha)^2}\mathbb{V}[S_i] - \frac{2\psi_a}{(1-\alpha)^2}\mathrm{Cov}(Z_i, S_i).$$

It remains to compute each term. Since $S_i \sim \text{Bernoulli}(\alpha)$,

$$\mathbb{V}[S_i] = \alpha(1-\alpha).$$

By direct computation,

$$\mathbb{V}[Z_i] = (1-\alpha)\,\mathbb{E}_T\left[\left(\mu_{(a)}^S(X)\right)^2\right] - ((1-\alpha)\psi_a)^2.$$

Moreover, since $(1-S_i) \times Z_i = Z_i$, we have

$$\text{Cov}(Z_i, S_i) = (1-\alpha)\alpha\psi_a.$$

Substituting into the expression for $V_{a,\text{tG}}^*$, we find

$$V_{a,\text{tG}}^* = \frac{1}{1-\alpha}\left(\mathbb{E}_T\left[\left(\mu_{(a)}^S(X)\right)^2\right] - (\psi_a)^2\right).$$

**weighted G-formula.** Define

$$Z_i = S_i \cdot r(X_i) \cdot \mu_{(a)}^S(X_i),$$

so that

$$\psi_{a,\text{wG}}^* = \frac{\frac{1}{N}\sum_{i=1}^N Z_i}{\frac{1}{N}\sum_{i=1}^N S_i} = \frac{\bar{Z}_N}{\bar{S}_N}.$$

We first compute the expectations of $Z_i$ and $S_i$. Using the definition of $Z_i$ and that $S$ is binary,

$$\begin{aligned}
\mathbb{E}[Z_i] &= \mathbb{E}\left[S_i \cdot r(X) \cdot \mu_{(a)}^S(X)\right] \\
&= \alpha\,\mathbb{E}_S\left[r(X) \cdot \mu_{(a)}^S(X)\right] \\
&= \alpha\,\mathbb{E}_T\left[\mu_{(a)}^S(X)\right] = \alpha\psi_a,
\end{aligned}$$

where we used the importance sampling identity $\mathbb{E}_S[r(X)f(X)] = \mathbb{E}_T[f(X)]$. By the multivariate central limit theorem,

$$\sqrt{N}\left(\begin{pmatrix}\bar{Z}_N \\ \bar{S}_N\end{pmatrix} - \begin{pmatrix}\alpha\psi_a \\ \alpha\end{pmatrix}\right) \xrightarrow{d} \mathcal{N}(0, \Sigma),$$

where

$$\Sigma = \begin{pmatrix} \mathbb{V}[Z_i] & \text{Cov}(Z_i, S_i) \\ \text{Cov}(Z_i, S_i) & \mathbb{V}[S_i] \end{pmatrix}.$$

The gradient of the map $h : (x, y) \mapsto x/y$ evaluated at $(\alpha\psi_a, \alpha)$ is

$$u = \nabla h(\alpha\psi_a, \alpha) = \left(\frac{1}{\alpha}, -\frac{\psi_a}{\alpha}\right).$$

By the Delta method,

$$\sqrt{N}\left(\psi_{a,\text{wG}}^* - \psi_a\right) \xrightarrow{d} \mathcal{N}(0, V_{a,\text{wG}}^*),$$

where

$$V_{a,\text{wG}}^* = u^\top \Sigma u = \frac{1}{\alpha^2}\mathbb{V}[Z_i] + \frac{\psi_a^2}{\alpha^2}\mathbb{V}[S_i] - \frac{2\psi_a}{\alpha^2}\text{Cov}(Z_i, S_i).$$

We compute each term:

$$\begin{aligned}
\mathbb{V}[Z_i] &= \mathbb{E}[Z_i^2] - (\mathbb{E}[Z_i])^2 = \alpha\,\mathbb{E}_T\left[r(X)\left(\mu_{(a)}^S(X)\right)^2\right] - \alpha^2\psi_a^2, \\
\mathbb{V}[S_i] &= \alpha(1-\alpha), \\
\text{Cov}(Z_i, S_i) &= \mathbb{E}[Z_i S_i] - \mathbb{E}[Z_i]\mathbb{E}[S_i] = \alpha\,\mathbb{E}_T\left[\mu_{(a)}^S(X)\right] - \alpha^2\psi_a = \alpha(1-\alpha)\psi_a.
\end{aligned}$$

Substituting into the expression for $V_{a,\mathrm{wG}}^*$, we get:

$$
\begin{aligned}
V_{a,\mathrm{wG}}^* &= \frac{1}{\alpha^2}\left(\alpha\,\mathbb{E}_T\left[r(X)(\mu_{(a)}^S(X))^2\right] - \alpha^2\psi_a^2\right) + \frac{\psi_a^2}{\alpha^2}\alpha(1-\alpha) - \frac{2\psi_a}{\alpha^2}\alpha(1-\alpha)\psi_a \\
&= \frac{1}{\alpha}\mathbb{E}_T\left[r(X)(\mu_{(a)}^S(X))^2\right] - \psi_a^2 + \frac{\psi_a^2(1-\alpha)}{\alpha} - \frac{2\psi_a^2(1-\alpha)}{\alpha} \\
&= \frac{1}{\alpha}\mathbb{E}_T\left[r(X)(\mu_{(a)}^S(X))^2\right] - \frac{\psi_a^2}{\alpha}.
\end{aligned}
$$

$\square$

**Proposition 13.** *Grant Assumption 1 to 1 defining $\beta^{(a)} := [c^{(a)}, \gamma^{(a)}]$, $V := [1, X]$. We rearrange the source $Y_i$ and $V_i$ so that the first $n_1$ observations of correspond to $A = 1$. We then define $\mathbf{Y}_1 = (Y_1, \ldots, Y_{n_1})^\top$ and $\mathbf{Y}_0 = (Y_{n_1+1}, \ldots, Y_n)^\top$, as well as $\mathbf{V}_1 = (V_1, \ldots, V_{n_1})^\top$ and $\mathbf{V}_0 = (V_{n_1+1}, \ldots, V_n)^\top$. Letting $\hat{\alpha} = (\sum_{i=1}^n S_i)/N$ and for all $a \in \{0, 1\}$,*

$$
\bar{V}^{(0)} = \frac{1}{\sum_{i=1}^n \mathbb{1}_{S_i=0}}\sum_{i=1}^n \mathbb{1}_{S_i=0}V_i \quad and \quad \hat{\beta}^{(a)} = \left(\frac{1}{n_a}\mathbf{V}_a^\top\mathbf{V}_a\right)^{-1}\frac{1}{n_a}\mathbf{V}_a^\top\mathbf{Y}_a.
$$

*Defining $\nu = \mathbb{E}_S[X]$ and $\Sigma = Var(X|S = 1)$, we have*

$$
\sqrt{N}(\hat{\theta}_N - \theta_\infty) \xrightarrow{d} \mathcal{N}(0, \Sigma),
$$

*where*

$$
\hat{\theta}_N = \begin{pmatrix}\bar{V}_{(0)} \\ \hat{\beta}^{(0)} \\ \hat{\beta}^{(1)}\end{pmatrix}, \quad \theta_\infty = \begin{pmatrix}\mathbb{E}_T[V] \\ \beta^{(0)} \\ \beta^{(1)}\end{pmatrix}, \quad \Sigma = \begin{pmatrix}\frac{\mathrm{Var}[V|S=0]}{(1-\alpha)} & 0 & 0 \\ 0 & \frac{\sigma^2 M^{-1}}{\alpha(1-\pi)} & 0 \\ 0 & 0 & \frac{\sigma^2 M^{-1}}{\alpha\pi}\end{pmatrix},
$$

*with $M^{-1} = \begin{bmatrix}1 + \nu^T\Sigma^{-1}\nu & -\nu^T\Sigma^{-1} \\ -\Sigma^{-1}\nu & \Sigma^{-1}\end{bmatrix}.$*

*Proof.* Using M-estimation theory to prove asymptotic normality of the $\theta_N$, we first define the following:

$$
\lambda(Z, \theta) = \begin{pmatrix}\lambda_1(Z, \theta) \\ \lambda_2(Z, \theta) \\ \lambda_3(Z, \theta)\end{pmatrix} := \begin{pmatrix}(1 - S)(V - \theta_0) \\ S(1 - A)\left(V\epsilon^{(0)} - VV^\top\left(\theta_1 - \beta^{(0)}\right)\right) \\ SA\left(V\epsilon^{(1)} - VV^\top\left(\theta_2 - \beta^{(1)}\right)\right)\end{pmatrix}
$$

where $\theta = (\theta_0, \theta_1, \theta_2, \theta_3)$. We still have that $\hat{\theta}_N$ is an M-estimator of type $\lambda$ [see 42] since

$$
\sum_{i=1}^N \lambda(Z_i, \hat{\theta}_N) = 0.
$$

Note that

$$
\begin{aligned}
\mathbb{E}\left[\lambda_1(Z, \theta_\infty)\right] &= \mathbb{E}\left[(1 - S)(V - \mathbb{E}_T[V])\right] \\
&= (1 - \alpha)\mathbb{E}_T\left[(V - \mathbb{E}_T[V])\right] \\
&= 0.
\end{aligned}
$$

We also have

$$
\begin{aligned}
\mathbb{E}\left[\lambda_2(Z, \theta_\infty)\right] &= \mathbb{E}\left[S(1 - A)V\epsilon^{(0)}\right] \\
&= \alpha\mathbb{E}_S\left[(1 - A)V\epsilon^{(0)}\right] \\
&= \alpha(1 - \pi)\mathbb{E}_S\left[V\epsilon^{(0)}\right] \\
&= \alpha(1 - \pi)\mathbb{E}_S\left[V\mathbb{E}_S\left[\epsilon^{(0)}|V\right]\right] \\
&= 0.
\end{aligned}
$$

Similarly, we can show that $\mathbb{E}\left[\lambda_3(Z, \theta_\infty)\right] = 0$. Since $\lambda(Z, \theta)$ is a linear function of $\theta$, $\theta_\infty$ is the only value of $\theta$ such that $\mathbb{E}\left[\lambda(Z, \theta)\right] = 0$ Define

$$A\left(\theta_\infty\right) = \mathbb{E}\left[\frac{\partial \lambda}{\partial \theta}\Big|_{\theta=\theta_\infty}\right] \quad \text{and} \quad B(\theta_\infty) = \mathbb{E}\left[\lambda(Z, \theta_\infty)\lambda(Z, \theta_\infty)^T\right].$$

Next, we check the conditions of Theorem 7.2 in Stefanski and Boos [42]. First, we compute $A\left(\theta_\infty\right)$ and $B\left(\theta_\infty\right)$. Since

$$\frac{\partial \lambda}{\partial \theta}(Z, \theta) = \begin{pmatrix} -(1-S) & 0 & 0 \\ 0 & -S(1-A)VV^\top & 0 \\ 0 & 0 & -SAVV^\top \end{pmatrix},$$

we obtain

$$A\left(\theta_\infty\right) = \begin{pmatrix} -(1-\alpha) & 0 & 0 \\ 0 & -\alpha(1-\pi)M & 0 \\ 0 & 0 & -\alpha\pi M \end{pmatrix},$$

where $M = \mathbb{E}_S\left[VV^\top\right]$, which leads to

$$A^{-1}\left(\theta_\infty\right) = \begin{pmatrix} -\frac{1}{1-\alpha} & 0 & 0 \\ 0 & -\frac{M^{-1}}{\alpha(1-\pi)} & 0 \\ 0 & 0 & -\frac{M^{-1}}{\alpha\pi} \end{pmatrix}.$$

Regarding $B(\theta_\infty)$, since we have $A(1 - A) = 0$ and $S(1 - S) = 0$, elementary calculations show that:

$$B(\theta_\infty)_{2,3} = B(\theta_\infty)_{3,2} = 0 \quad \text{and} \quad \begin{array}{l} B(\theta_\infty)_{1,2} = B(\theta_\infty)_{2,1} = 0 \\ B(\theta_\infty)_{1,3} = B(\theta_\infty)_{3,1} = 0. \end{array}$$

Besides

$$B(\theta_\infty)_{1,1} = \mathbb{E}\left[(1-S)^2(V - \mathbb{E}_T\left[V\right])(V - \mathbb{E}_T\left[V\right])^\top\right]$$
$$= (1-\alpha)\operatorname{Var}\left[V|S=0\right],$$

We can also note that:

$$B(\theta_\infty)_{3,3} = \mathbb{E}\left[S^2 A^2 VV^\top(\epsilon^{(1)})^2\right]$$
$$= \alpha\pi\mathbb{E}_S\left[VV^\top(\epsilon^{(1)})^2|A=1\right]$$
$$= \alpha\pi\sigma^2 M$$

and similarly,

$$B(\theta_\infty)_{2,2} = \alpha(1-\pi)\sigma^2 M.$$

Gathering all calculations, we have

$$B(\theta_\infty) = \begin{pmatrix} (1-\alpha)\operatorname{Var}\left[V|S=0\right] & 0 & 0 \\ 0 & \alpha(1-\pi)\sigma^2 M & 0 \\ 0 & 0 & \alpha\pi\sigma^2 M \end{pmatrix},$$

Based on the previous calculations, we have:

- $\lambda(z, \theta)$ and its first two partial derivatives with respect to $\theta$ exist for all $z$ and for all $\theta$ in the neighborhood of $\theta_\infty$.

- For each $\theta$ in the neighborhood of $\theta_\infty$, we have for all $i, j, k \in \{1, 3\}$:

$$\left|\frac{\partial^2}{\partial\theta_i\partial\theta_j}\lambda_k(z, \theta)\right| \leq 1.$$

- $A(\theta_\infty)$ exists and is nonsingular.

- $B(\theta_\infty)$ exists and is finite.

Since we have:

$$\sum_{i=1}^{n} \lambda(T_i, Z_i, \hat{\theta}_N) = 0 \quad \text{and} \quad \hat{\theta}_N \xrightarrow{p} \theta_\infty.$$

Then, the conditions of Theorem 7.2 in Stefanski and Boos [42] are satisfied, we have:

$$\sqrt{n} \left( \hat{\theta}_N - \theta_\infty \right) \xrightarrow{d} \mathcal{N} \left( 0, A(\theta_\infty)^{-1} B(\theta_\infty)(A(\theta_\infty)^{-1})^\top \right),$$

where:

$$A(\theta_\infty)^{-1} B(\theta_\infty)(A(\theta_\infty)^{-1})^\top = \begin{pmatrix} \frac{\mathrm{Var}[V|S=0]}{(1-\alpha)} & 0 & 0 \\ 0 & \frac{\sigma^2 M^{-1}}{\alpha(1-\pi)} & 0 \\ 0 & 0 & \frac{\sigma^2 M^{-1}}{\alpha \pi} \end{pmatrix}.$$

$\square$

**Corollary 1.** *For all $a \in \{0, 1\}$, let $\hat{\tau}_{a,\mathrm{tG}}^{OLS}$ denote the transported G-formula estimator where linear regressions are used to estimate $\mu_{(a)}^S$. Then, under Assumption 1 to 3 and 1:*

$$\sqrt{N} \begin{pmatrix} \hat{\tau}_{1,\mathrm{tG}}^{OLS} - \psi_1 \\ \hat{\tau}_{0,\mathrm{tG}}^{OLS} - \psi_0 \end{pmatrix} \xrightarrow{d} \mathcal{N} \left( 0, \Sigma_{\mathrm{tG}}^{OLS} \right),$$

*with*

$$\Sigma_{\mathrm{tG}}^{OLS} = \begin{pmatrix} (\beta^{(1)})^\top \frac{\mathrm{Var}[V|S=0]}{1-\alpha} \beta^{(1)} + \mathbb{E}_T[V]^\top \frac{\sigma^2 M^{-1}}{\alpha \pi} \mathbb{E}_T[V] & (\beta^{(1)})^\top \frac{\mathrm{Var}[V|S=0]}{1-\alpha} \beta^{(0)} \\ (\beta^{(1)})^\top \frac{\mathrm{Var}[V|S=0]}{1-\alpha} \beta^{(0)} & (\beta^{(0)})^\top \frac{\mathrm{Var}[V|S=0]}{1-\alpha} \beta^{(0)} + \mathbb{E}_T[V]^\top \frac{\sigma^2 M^{-1}}{\alpha(1-\pi)} \mathbb{E}_T[V] \end{pmatrix},$$

*Proof.* Recall that for $a \in \{0, 1\}$, we have:

$$\hat{\tau}_{a,\mathrm{tG}}^{\mathrm{OLS}} = (\hat{\beta}^{(a)})^\top \bar{V}_{(0)}$$

From Proposition 13, we know that:

$$\sqrt{N}(\hat{\theta}_N - \theta_\infty) \xrightarrow{d} \mathcal{N}(0, \Sigma),$$

with $\hat{\theta}_N = (\bar{V}_{(0)}^\top, (\hat{\beta}^{(0)})^\top, (\hat{\beta}^{(1)})^\top)^\top \in \mathbb{R}^{3p+3}$, and where $\Sigma$ is the block-diagonal covariance matrix. By applying the delta method to the map

$$g(\bar{V}_{(0)}, \hat{\beta}^{(0)}, \hat{\beta}^{(1)}) = \begin{pmatrix} (\hat{\beta}^{(1)})^\top \bar{V}_{(0)} \\ (\hat{\beta}^{(0)})^\top \bar{V}_{(0)} \end{pmatrix},$$

the asymptotic distribution of $\sqrt{N}(\hat{\tau}_{a,\mathrm{tG}}^{\mathrm{OLS}} - \psi_a)$ is multivariate normal with covariance matrix:

$$\Sigma_{\mathrm{tG}}^{\mathrm{OLS}} = J \Sigma J^\top,$$

where $J$ is the Jacobian of $g$ evaluated at $(\mathbb{E}_T[V], \beta^{(0)}, \beta^{(1)})$:

$$J = \begin{pmatrix} (\beta_{(1)})^\top & 0 & \mathbb{E}_T[V]^\top \\ (\beta_{(0)})^\top & \mathbb{E}_T[V]^\top & 0 \end{pmatrix}.$$

we obtain:

$$\Sigma_{\mathrm{tG}}^{\mathrm{OLS}} = \begin{pmatrix} (\beta^{(1)})^\top \frac{\mathrm{Var}[V|S=0]}{1-\alpha} \beta^{(1)} + \mathbb{E}_T[V]^\top \frac{\sigma^2 M^{-1}}{\alpha \pi} \mathbb{E}_T[V] & (\beta^{(1)})^\top \frac{\mathrm{Var}[V|S=0]}{1-\alpha} \beta^{(0)} \\ (\beta^{(1)})^\top \frac{\mathrm{Var}[V|S=0]}{1-\alpha} \beta^{(0)} & (\beta^{(0)})^\top \frac{\mathrm{Var}[V|S=0]}{1-\alpha} \beta^{(0)} + \mathbb{E}_T[V]^\top \frac{\sigma^2 M^{-1}}{\alpha(1-\pi)} \mathbb{E}_T[V] \end{pmatrix},$$

(30)

$\square$

**Proposition 14.** *For all $a \in \{0, 1\}$, let $\hat{\tau}_{a,\mathrm{wG}}^{OLS}$ denote the weighted G-formula estimators where the density ratio is estimated using a logistic regression and linear regressions are used to estimate $\mu_{(a)}^S$. Then, under Assumption 1 to 1:*

$$\sqrt{N}\left(\begin{pmatrix} \hat{\psi}_{0,\mathrm{wG}}^{OLS} \\ \hat{\psi}_{1,\mathrm{wG}}^{OLS} \end{pmatrix} - \begin{pmatrix} \psi_0 \\ \psi_1 \end{pmatrix}\right) \xrightarrow{d} \mathcal{N}(0, \Sigma_{\mathrm{wG}}^{OLS}).$$

*Proof.* Let $Z = (S, X, S \times A, S \times Y)$ and define $\theta = (\theta_1, \theta_2, \theta_3, \beta, \theta_4, \theta_5)$. Consider the estimating function $\lambda(Z, \theta)$ defined as:

$$\lambda(Z, \theta) = \begin{pmatrix} \frac{1-\sigma(X,\beta)}{\sigma(X,\beta)} \frac{S\theta_4^\top V}{1-\theta_3} - \theta_1, \\ \frac{1-\sigma(X,\beta)}{\sigma(X,\beta)} \frac{S\theta_5^\top V}{1-\theta_3} - \theta_2, \\ S - \theta_3, \\ V(S - \sigma(X,\beta)), \\ S(1-A)\left(V\epsilon(0) - VV^\top(\theta_4 - \beta^{(0)})\right), \\ SA\left(V\epsilon(1) - VV^\top(\theta_5 - \beta^{(1)})\right) \end{pmatrix}$$

Define $\hat{\theta}_N = (\hat{\psi}_{0,\mathrm{wG}}^{OLS}, \hat{\psi}_{1,\mathrm{wG}}^{OLS}, \hat{\alpha}, \hat{\beta}_N, \hat{\beta}^{(0)}, \hat{\beta}^{(1)})$, where $\hat{\beta}_N$ is the MLE from the logistic regression of $S$ on $X$, and $r(X_i, \hat{\beta}_N) = \frac{1-\sigma(X_i,\hat{\beta}_N)}{\sigma(X_i,\hat{\beta}_N)} \cdot \frac{\hat{\alpha}}{1-\hat{\alpha}}$. Then, the estimators $\hat{\psi}_{a,\mathrm{wG}}^{OLS}$ take the form:

$$\hat{\psi}_{a,\mathrm{wG}}^{OLS} = \frac{1}{N} \sum_{i=1}^{N} S_i \cdot \frac{1 - \sigma(X_i, \hat{\beta}_N)}{\sigma(X_i, \hat{\beta}_N)(1 - \hat{\alpha})} \cdot (\hat{\beta}^{(a)})^\top V_i.$$

The estimator $\hat{\theta}_N$ solves the estimating equation:

$$\sum_{i=1}^{N} \lambda(Z_i, \hat{\theta}_N) = 0.$$

This setup is structurally identical to that of proposition 9 and 13, and the M-estimation theory in Stefanski and Boos [42] applies directly. Specifically, the regularity conditions (e.g., smoothness of $\lambda$, identifiability, uniqueness of root) are satisfied.

As before, the asymptotic distribution of the M-estimator is:

$$\sqrt{N}(\hat{\theta}_N - \theta_\infty) \xrightarrow{d} \mathcal{N}(0, A^{-1}B(A^{-1})^\top),$$

where $A$ and $B$ are the Jacobian of the estimating function and its variance, respectively, evaluated at $\theta_\infty = (\psi_0, \psi_1, \alpha, \beta_\infty, \beta^{(0)}, \beta^{(1)})$. Focusing on the top-left $2 \times 2$ block of the sandwich covariance matrix—corresponding to $(\hat{\psi}_{0,\mathrm{wG}}^{OLS}, \hat{\psi}_{1,\mathrm{wG}}^{OLS})$—we denote this block by $\Sigma_{\mathrm{wG}}^{OLS}$, and conclude:

$$\sqrt{N}\begin{pmatrix} \hat{\psi}_{0,\mathrm{wG}}^{OLS} - \psi_0 \\ \hat{\psi}_{1,\mathrm{wG}}^{OLS} - \psi_1 \end{pmatrix} \xrightarrow{d} \mathcal{N}(0, \Sigma_{\mathrm{wG}}^{OLS}) \quad \text{where} \quad \Sigma_{\mathrm{wG}}^{OLS} = \begin{pmatrix} V_{0,\mathrm{wG}}^{OLS} & C_{\mathrm{wG}}^{OLS} \\ C_{\mathrm{wG}}^{OLS} & V_{1,\mathrm{wG}}^{OLS} \end{pmatrix},$$

where using Lemma 1, we simplify the expression to:

$$V_{a,\mathrm{wG}}^{OLS} = \frac{(\beta^{(a)})^\top \Delta \beta^{(a)}}{\alpha} - \frac{\psi_a^2}{1-\alpha} + 2\frac{(\beta^{(a)})^\top \mathbb{E}_T[VV^\top]\beta^{(a)}}{1-\alpha} - (\beta^{(a)})^\top \mathbb{E}_T[VV^\top]Q^{-1}\mathbb{E}_T[VV^\top]\beta^{(a)}$$
(31)

$$+ \sigma^2 \cdot \frac{\mathbb{E}_T[V]^\top M^{-1}\mathbb{E}_T[V]}{\alpha\pi_a}.$$
(32)

For the covariance terms we get:

$$C_{\mathrm{wG}}^{OLS} = \frac{(\beta^{(0)})^\top \Delta \beta^{(1)}}{\alpha} - \frac{\psi_1\psi_0}{1-\alpha} + 2\frac{(\beta^{(1)})^\top \mathbb{E}_T[VV^\top]\beta^{(0)}}{1-\alpha} - (\beta^{(0)})^\top \mathbb{E}_T[VV^\top]Q^{-1}\mathbb{E}_T[VV^\top]\beta^{(1)},$$
(33)

where $\Delta = \mathbb{E}_T[r(X)VV^\top]$. $\qquad \square$

**Lemma 2.** *Grant Assumption 1 to Linear model 1, then:*

- *the matrix $\Sigma_{\text{wG}}^{OLS} - \Sigma_{\text{tG}}^{OLS}$ is semi-definite positive,*

- *the matrix $\Sigma_{wHT} - \Sigma_{\text{wG}}^{OLS}$ is semi-definite positive.*

*Consequently, $V_{\Phi,\text{tG}}^{OLS} \le V_{\Phi,\text{wG}}^{OLS} \le V_{\Phi,wHT}$.*

*Proof.* **First statement** We first start with $\Sigma_{\text{wHT}} - \Sigma_{\text{wG}}^{\text{OLS}}$. Under Linear model 1

$$Y^{(a)} = (\beta^{(a)})^\top V + \epsilon^{(a)}, \quad \text{where} \quad \mathbb{E}[\epsilon^{(a)} \mid X] = 0, \quad \text{Var}(\epsilon^{(a)} \mid X) = \sigma^2,$$

we can express the variances $V_{a,\text{wHT}}$ and the covariance term $C_{\text{wHT}}$ defined in 19 of the weighted Horvitz-thompson in terms of the model parameters and the distribution of $X$. Starting with $V_{a,\text{wHT}}$, we expand each expectation by substituting the linear form of $Y^{(a)}$. The first term of $V_{a,\text{wHT}}$ becomes

$$\mathbb{E}_T[r(X)(Y^{(a)})^2] = \mathbb{E}_T\left[r(X)\left(((\beta^{(a)})^\top V)^2 + 2(\beta^{(a)})^\top V \epsilon^{(a)} + (\epsilon^{(a)})^2\right)\right].$$

Taking expectations and applying the assumptions $\mathbb{E}[\epsilon^{(a)} \mid X] = 0$ and $\text{Var}(\epsilon^{(a)} \mid X) = \sigma^2$, this simplifies to

$$\mathbb{E}_T[r(X)(Y^{(a)})^2] = \mathbb{E}_T[r(X)((\beta^{(a)})^\top V)^2] + \sigma^2 \mathbb{E}_T[r(X)].$$

The second term, becomes $(\mathbb{E}_T[(\beta^{(a)})^\top X])^2$, since the error has mean zero. Moving to the third term, we use linearity to write

$$\mathbb{E}_T[Y^{(a)} V^\top] = (\beta^{(a)})^\top \mathbb{E}_T[V V^\top],$$

and thus the quadratic form becomes

$$\mathbb{E}_T[Y^{(a)} V^\top] Q^{-1} \mathbb{E}_T[Y^{(a)} V] = (\beta^{(a)})^\top \mathbb{E}_T[V V^\top] Q^{-1} \mathbb{E}_T[V V^\top] \beta^{(a)}.$$

For the fourth term, we compute

$$\mathbb{E}_T[\sigma(X) Y^{(a)} V] = \mathbb{E}_T[\sigma(X) V V^\top] \beta^{(a)},$$

which leads to

$$2 \mathbb{E}_T[Y^{(a)} V^\top] Q^{-1} \mathbb{E}_T[\sigma(X) Y^{(a)} V] = 2 (\beta^{(a)})^\top \mathbb{E}_T[V V^\top] Q^{-1} \mathbb{E}_T[\sigma(X) V V^\top] \beta^{(a)}.$$

Noting that $Q = (1 - \alpha)\mathbb{E}_T\left[\sigma(X, \beta) V V^\top\right]$, this further simplifies to:

$$2 \mathbb{E}_T[Y^{(a)} V^\top] Q^{-1} \mathbb{E}_T[\sigma(X) Y^{(a)} V] = \frac{2 (\beta^{(a)})^\top \mathbb{E}_T[V V^\top] \beta^{(a)}}{1 - \alpha}.$$

Combining all components, using linearity and the definition of $\Delta$, we have

$$V_{a,\text{wHT}} = \frac{1}{\alpha \pi_a} \left((\beta^{(a)})^\top \Delta \beta^{(a)} + \sigma^2 \mathbb{E}_T[r(X)]\right) - \frac{(\mathbb{E}_T[(\beta^{(a)})^\top V])^2}{1 - \alpha}$$
$$- (\beta^{(a)})^\top \mathbb{E}_T[V V^\top] Q^{-1} \mathbb{E}_T[V V^\top] \beta^{(a)} + 2 \frac{(\beta^{(a)})^\top \mathbb{E}_T[V V^\top] \beta^{(a)}}{1 - \alpha}.$$

Turning to the cross-covariance term $C_{\text{wHT}}$, we proceed similarly. Noting that

$$\mathbb{E}_T[Y^{(a)}] = \mathbb{E}_T[(\beta^{(a)})^\top V],$$
$$\mathbb{E}_T[Y^{(a)} V^\top] = (\beta^{(a)})^\top \mathbb{E}_T[V V^\top],$$
$$\mathbb{E}_T[\sigma(X) Y^{(a)} V] = \mathbb{E}_T[\sigma(X) V V^\top] \beta^{(a)},$$

we substitute these into the original definition to obtain

$$C_{\text{wHT}} = -\frac{\mathbb{E}_T[(\beta^{(1)})^\top V] \cdot \mathbb{E}_T[(\beta^{(0)})^\top V]}{1 - \alpha} + 2\frac{(\beta^{(1)})^\top \mathbb{E}_T[V V^\top] \beta^{(0)}}{1 - \alpha}$$
$$- (\beta^{(1)})^\top \mathbb{E}_T[V V^\top] Q^{-1} \mathbb{E}_T[V V^\top] \beta^{(0)}.$$

Now, we can compute the following quantities:

$$V_{a,\text{wHT}} - V_{a,\text{wG}}^{\text{OLS}} = \frac{1 - \pi_a}{\alpha \pi_a} (\beta^{(a)})^\top \Delta \beta^{(a)} + \frac{\sigma^2}{\alpha \pi_a} \left( \mathbb{E}_T[r(X)] - \mathbb{E}_T[V]^\top M^{-1} \mathbb{E}_T[V] \right)$$

and

$$C_{\text{wHT}} - C_{\text{wG}}^{\text{OLS}} = -\frac{(\beta^{(0)})^\top \Delta \beta^{(1)}}{\alpha}.$$

Since $0 < \pi_a < 1$, we immediately have that

$$\frac{1 - \pi_a}{\alpha \pi_a} (\beta^{(a)})^\top \Delta \beta^{(a)} \geq 0.$$

Now, turning to the second term in the expression for $V_{a,\text{wHT}} - V_{a,\text{wG}}^{\text{OLS}}$, observe that

$$\mathbb{E}_T[r(X)] - \mathbb{E}_T[V]^\top M^{-1} \mathbb{E}_T[V]$$

can be rewritten using the fact that the target distribution $T$ is defined via reweighting from the source distribution $S$, with $r(X)$. This yields:

$$\mathbb{E}_T[r(X)] - \mathbb{E}_T[V]^\top M^{-1} \mathbb{E}_T[V] = \mathbb{E}_S[r(X)^2] - \mathbb{E}_S[r(X)V]^\top \mathbb{E}_S[VV^\top]^{-1} \mathbb{E}_S[r(X)V].$$

To interpret this expression, we recognize it as a variance-type quantity. In particular, we can rewrite it as:

$$\mathbb{E}_S[r(X)^2] - 2\mathbb{E}_S[r(X)V]^\top \mathbb{E}_S[VV^\top]^{-1} \mathbb{E}_S[r(X)V]$$
$$+ \mathbb{E}_S[r(X)V]^\top \mathbb{E}_S[VV^\top]^{-1} \mathbb{E}_S[VV^\top] \mathbb{E}_S[VV^\top]^{-1} \mathbb{E}_S[r(X)V],$$

which simplifies to:

$$\mathbb{E}_S \left[ \left( r(X) - \mathbb{E}_S[r(X)V]^\top \mathbb{E}_S[VV^\top]^{-1} V \right)^2 \right].$$

This is the expected squared residual from projecting $r(X)$ onto the linear span of $V$, and is therefore nonnegative. Hence,

$$\mathbb{E}_T[r(X)] - \mathbb{E}_T[V]^\top M^{-1} \mathbb{E}_T[V] \geq 0.$$

Combining this result with the earlier inequality, we conclude that

$$V_{a,\text{wHT}} - V_{a,\text{wG}}^{\text{OLS}} \geq 0.$$

Since this holds for both $a = 0$ and $a = 1$, and noting that

$$C_{\text{wHT}} - C_{\text{wG}}^{\text{OLS}} = -\frac{(\beta^{(0)})^\top \Delta \beta^{(1)}}{\alpha},$$

we now analyze the overall matrix difference. The trace satisfies:

$$\text{tr}(\Sigma_{\text{wHT}} - \Sigma_{\text{wG}}^{\text{OLS}}) = (V_{1,\text{wHT}} - V_{1,\text{wG}}^{\text{OLS}}) + (V_{0,\text{wHT}} - V_{0,\text{wG}}^{\text{OLS}}) \geq 0,$$

and the determinant becomes:

$$\det(\Sigma_{\text{wHT}} - \Sigma_{\text{wG}}^{\text{OLS}}) = (V_{1,\text{wHT}} - V_{1,\text{wG}}^{\text{OLS}})(V_{0,\text{wHT}} - V_{0,\text{wG}}^{\text{OLS}}) - (C_{\text{wHT}} - C_{\text{wG}}^{\text{OLS}})^2$$
$$\geq \frac{1}{\alpha^2} \left( (\beta^{(1)})^\top \Delta \beta^{(1)} \cdot (\beta^{(0)})^\top \Delta \beta^{(0)} - ((\beta^{(1)})^\top \Delta \beta^{(0)})^2 \right) \geq 0,$$

where the final inequality follows from Cauchy–Schwarz. Since both the trace and determinant of $\Sigma_{\text{wHT}} - \Sigma_{\text{wG}}^{\text{OLS}}$ are nonnegative, we conclude that the matrix $\Sigma_{\text{wHT}} - \Sigma_{\text{wG}}^{\text{OLS}}$ is positive semi-definite.

**Second statement** Now, we want to prove that $\Sigma_{\text{wG}}^{\text{OLS}} - \Sigma_{\text{tG}}^{\text{OLS}}$ is semi-definite positive. Observe that using the expression defined in 19, 31 and 30 we have

$$V_{a,\text{wG}}^{\text{OLS}} - V_{a,\text{tG}}^{\text{OLS}} = (\beta^{(a)})^\top H \beta^{(a)} \quad \text{and} \quad C_{\text{wG}}^{\text{OLS}} - C_{\text{tG}}^{\text{OLS}} = (\beta^{(1)})^\top H \beta^{(0)},$$

where the matrix $H$ is defined as

$$H := \frac{1}{1 - \alpha} \left( \mathbb{E}_T \left[ \frac{1 - \sigma(X, \beta)}{\sigma(X, \beta)} VV^\top \right] + \mathbb{E}_T[VV^\top] - \mathbb{E}_T[VV^\top] \left( \mathbb{E}_T[\sigma(X, \beta)VV^\top] \right)^{-1} \mathbb{E}_T[VV^\top] \right).$$

This can be rewritten more compactly as:

$$H = \frac{1}{1-\alpha}(C - AB^{-1}A),$$

where we define:

$$C = \mathbb{E}_T\left[\frac{1}{\sigma(X,\beta)}VV^\top\right], \quad A = \mathbb{E}_T[VV^\top], \quad B = \mathbb{E}_T[\sigma(X,\beta)VV^\top].$$

First, we define the vector:

$$\tilde{V} := \begin{bmatrix} \sqrt{\sigma(X,\beta)}V \\ \frac{1}{\sqrt{\sigma(X,\beta)}}V \end{bmatrix} \in \mathbb{R}^{2d}.$$

(Note that here we changed the definition of $\tilde{V}$ to simplify later steps.)

Then the outer product $\tilde{V}\tilde{V}^\top$ is:

$$\tilde{V}\tilde{V}^\top = \begin{bmatrix} \sigma(X,\beta)VV^\top & VV^\top \\ VV^\top & \frac{1}{\sigma(X,\beta)}VV^\top \end{bmatrix}.$$

Taking expectation, we define the matrix:

$$M := \mathbb{E}_T[\tilde{V}\tilde{V}^\top] = \begin{bmatrix} B & A \\ A^\top & C \end{bmatrix}.$$

We now show that $M$ is PSD. For any $z \in \mathbb{R}^{2d}$, we have:

$$z^\top M z = \mathbb{E}_T\left[z^\top \tilde{V}\tilde{V}^\top z\right] = \mathbb{E}_T\left[(z^\top \tilde{V})^2\right] \geq 0,$$

since each term in the expectation is a square. Therefore, $M \succeq 0$. By the Schur Complement Lemma for block matrices, provided that $B$ is invertible and square and $B \succeq 0$ and $M \succeq 0$, we get that

$$H = \frac{1}{1-\alpha}(C - AB^{-1}A) \succeq 0.$$

Now, by definition:

$$\Sigma_{\text{wG}}^{\text{OLS}} - \Sigma_{\text{tG}}^{\text{OLS}} = \begin{bmatrix} (\beta^{(0)})^\top H\beta^{(0)} & (\beta^{(1)})^\top H\beta^{(0)} \\ (\beta^{(1)})^\top H\beta^{(0)} & (\beta^{(1)})^\top H\beta^{(1)} \end{bmatrix}.$$

Since this matrix is a Gram matrix induced by $H \succeq 0$, it follows that $\Sigma_{\text{wG}}^{\text{OLS}} - \Sigma_{\text{tG}}^{\text{OLS}} \succeq 0$. $\quad\square$

**Proposition 15.** *Grant Assumption 1 to 1, and let $\hat{\tau}_{\Phi,\text{tG}}^{OLS}$ denote the transported G-formula estimators where linear regressions are used to estimate $\mu_{(a)}^S$. Then, $\hat{\tau}_{\Phi,\text{tG}}^{OLS}$ is asymptotically normal:*

$$\sqrt{N}\left(\hat{\tau}_{\Phi,\text{tG}}^{OLS} - \tau_\Phi^T\right) \xrightarrow{d} \mathcal{N}\left(0, V_{\Phi,\text{tG}}^{OLS}\right),$$

*where*

$$V_{\Phi,\text{tG}}^{OLS} = \frac{1}{1-\alpha}\left\|\frac{\partial\Phi}{\partial\psi_1}\beta^{(1)} + \frac{\partial\Phi}{\partial\psi_0}\beta^{(0)}\right\|_{Var[V|S=0]}$$

$$+ \frac{\sigma^2 \mathbb{E}_T[V]^\top M^{-1}\mathbb{E}_T[V]}{\alpha}\left[\frac{1}{1-\pi}\left(\frac{\partial\Phi}{\partial\psi_0}\right)^2 + \frac{1}{\pi}\left(\frac{\partial\Phi}{\partial\psi_1}\right)^2\right].$$

*Proof.* We begin by analyzing the transported OLS estimator, defined as

$$\hat{\tau}_{\Phi,\text{tG}}^{OLS} = \Phi\left((\hat{\beta}^{(1)})^\top \bar{V}_{(0)}, (\hat{\beta}^{(0)})^\top \bar{V}_{(0)}\right),$$

where $\bar{V}_{(0)}$ is the empirical mean of covariates in the target population. Under Linear model 1, the corresponding population estimand is

$$\tau_\Phi^T = \Phi\left((\beta^{(1)})^\top \mathbb{E}_T[V], (\beta^{(0)})^\top \mathbb{E}_T[V]\right),$$

where $\mathbb{E}_T[V]$ is the expectation of covariates under the target distribution.

To study the asymptotic behavior of the estimator, we perform a first-order Taylor expansion of $\Phi$ around the point $(\psi_1^*, \psi_0^*) := \left((\beta^{(1)})^\top \mathbb{E}_T[V],\ (\beta^{(0)})^\top \mathbb{E}_T[V]\right)$. This yields:

$$
\begin{aligned}
\hat{\tau}_{\Phi,\mathrm{tG}}^{\mathrm{OLS}} - \tau_\Phi^T &= \Phi\left((\hat{\beta}^{(1)})^\top \bar{V}_{(0)},\ (\hat{\beta}^{(0)})^\top \bar{V}_{(0)}\right) - \Phi\left((\hat{\beta}^{(1)})^\top \mathbb{E}_T[V],\ (\beta^{(0)})^\top \mathbb{E}_T[V]\right) \\
&= \left.\frac{\partial \Phi}{\partial \psi_1}\right|_{(\psi_1^*,\psi_0^*)} \left((\hat{\beta}^{(1)})^\top \bar{V}_{(0)} - (\beta^{(1)})^\top \mathbb{E}_T[V]\right) \\
&\quad + \left.\frac{\partial \Phi}{\partial \psi_0}\right|_{(\psi_1^*,\psi_0^*)} \left((\hat{\beta}^{(0)})^\top \bar{V}_{(0)} - (\beta^{(0)})^\top \mathbb{E}_T[V]\right) \\
&\quad + o_p\left(\left\| \begin{pmatrix} (\hat{\beta}^{(1)})^\top \bar{V}_{(0)} - (\beta^{(1)})^\top \mathbb{E}_T[V] \\ (\hat{\beta}^{(0)})^\top \bar{V}_{(0)} - (\beta^{(0)})^\top \mathbb{E}_T[V] \end{pmatrix}\right\|\right).
\end{aligned}
$$

To further decompose the linear terms, note that for each $a \in \{0,1\}$, we can write:

$$
(\hat{\beta}^{(a)})^\top \bar{V}_{(0)} - (\beta^{(a)})^\top \mathbb{E}_T[V] = (\hat{\beta}^{(a)} - \beta^{(a)})^\top \mathbb{E}_T[V] + (\hat{\beta}^{(a)})^\top \left(\bar{V}_{(0)} - \mathbb{E}_T[V]\right).
$$

Combining these expressions, we obtain:

$$
\begin{aligned}
\hat{\tau}_{\Phi,\mathrm{tG}}^{\mathrm{OLS}} - \tau_\Phi^T &= \left.\frac{\partial \Phi}{\partial \psi_1}\right|_{(\psi_1^*,\psi_0^*)} \left[(\hat{\beta}^{(1)} - \beta^{(1)})^\top \mathbb{E}_T[V] + (\hat{\beta}^{(1)})^\top \left(\bar{V}_{(0)} - \mathbb{E}_T[V]\right)\right] \\
&\quad + \left.\frac{\partial \Phi}{\partial \psi_0}\right|_{(\psi_1^*,\psi_0^*)} \left[(\hat{\beta}^{(0)} - \beta^{(0)})^\top \mathbb{E}_T[V] + (\hat{\beta}^{(0)})^\top \left(\bar{V}_{(0)} - \mathbb{E}_T[V]\right)\right] + o_p(N^{-1/2}).
\end{aligned}
$$

By the multivariate Central Limit Theorem, the Law of Large Numbers, and Slutsky's theorem—along with 13—we conclude:

$$
\sqrt{N}\left(\hat{\tau}_{\Phi,\mathrm{tG}}^{\mathrm{OLS}} - \tau_\Phi^T\right) \xrightarrow{d} \mathcal{N}\left(0,\ \alpha_\infty^\top \Sigma\, \alpha_\infty\right),
$$

where $\Sigma$ is defined in 30 and the influence vector $\alpha_\infty$ is given by

$$
\alpha_\infty = \left.\frac{\partial \Phi}{\partial \psi_1}\right|_{(\psi_1^*,\psi_0^*)} \alpha_{1,\infty} + \left.\frac{\partial \Phi}{\partial \psi_0}\right|_{(\psi_1^*,\psi_0^*)} \alpha_{0,\infty},
$$

and each $\alpha_{a,\infty} \in \mathbb{R}^p$ is defined as

$$
\alpha_{a,\infty}^\top = \left((\beta^{(a)})^\top,\ \mathbb{E}_T[V]^\top \cdot \mathbb{1}_{\{a=0\}},\ \mathbb{E}_T[V]^\top \cdot \mathbb{1}_{\{a=1\}}\right) \in \mathbb{R}^{3p+3}.
$$

Combining this with the block-diagonal form of $\Sigma$, we obtain the explicit asymptotic variance:

$$
\begin{aligned}
\alpha_\infty^\top \Sigma \alpha_\infty &= \frac{1}{1-\alpha} \left\| \frac{\partial \Phi}{\partial \psi_1} \beta^{(1)} + \frac{\partial \Phi}{\partial \psi_0} \beta^{(0)} \right\|_{\mathrm{Var}[V|S=0]} \\
&\quad + \frac{\sigma^2 \mathbb{E}_T[V]^\top M^{-1} \mathbb{E}_T[V]}{\alpha} \left[ \frac{1}{1-\pi} \left(\frac{\partial \Phi}{\partial \psi_0}\right)^2 + \frac{1}{\pi} \left(\frac{\partial \Phi}{\partial \psi_1}\right)^2 \right].
\end{aligned}
$$

$\square$

**Proposition 16.** *Let $\hat{\tau}_{\Phi,\mathrm{wG}}^{OLS}$ denote the weighted G-formula estimators where the density ratio is estimated using a logistic regression and linear regressions are used to estimate $\mu_{(a)}^S$. Then, under Assumption 1 to 1, $\hat{\tau}_{\Phi,\mathrm{wG}}^{OLS}$ is asymptotically normal:*

$$
\sqrt{N}\left(\hat{\tau}_{\Phi,\mathrm{wG}}^{OLS} - \tau_\Phi^T\right) \xrightarrow{d} \mathcal{N}\left(0, V_{\Phi,\mathrm{wG}}^{OLS}\right),
$$

*where*

$$
V_{\Phi,\mathrm{wG}}^{OLS} = \left(\frac{\partial \Phi}{\partial \psi_1}\right)^2 V_{1,\mathrm{wG}}^{OLS} + \left(\frac{\partial \Phi}{\partial \psi_0}\right)^2 V_{0,\mathrm{wG}}^{OLS} + 2\left(\frac{\partial \Phi}{\partial \psi_1}\right)\left(\frac{\partial \Phi}{\partial \psi_0}\right) C_{\mathrm{wG}}^{OLS}.
$$

*Proof.* Using 14, we apply the delta method to $\hat{\tau}_{\Phi,\mathrm{wG}}^{\mathrm{OLS}} = \Phi(\hat{\psi}_{1,\mathrm{wG}}^{\mathrm{OLS}}, \hat{\psi}_{0,\mathrm{wG}}^{\mathrm{OLS}})$ of the two-dimensional asymptotically normal vector. By the delta method:

$$\sqrt{N}\left(\hat{\tau}_{\Phi,\mathrm{wG}}^{\mathrm{OLS}} - \Phi(\psi_1, \psi_0)\right) \xrightarrow{d} \mathcal{N}\left(0, \nabla\Phi(\psi_1, \psi_0)^\top \Sigma_{\mathrm{wG}}^{\mathrm{OLS}} \nabla\Phi(\psi_1, \psi_0)\right),$$

where $\nabla\Phi(\psi_1, \psi_0) = \left(\frac{\partial\Phi}{\partial\psi_1}, \frac{\partial\Phi}{\partial\psi_0}\right)^\top$ is the gradient of $\Phi$ evaluated at the population means. Therefore we get:

$$V_{\Phi,\mathrm{wG}}^{\mathrm{OLS}} = \left(\frac{\partial\Phi}{\partial\psi_1}\right)^2 V_{1,\mathrm{wG}}^{\mathrm{OLS}} + \left(\frac{\partial\Phi}{\partial\psi_0}\right)^2 V_{0,\mathrm{wG}}^{\mathrm{OLS}} + 2\left(\frac{\partial\Phi}{\partial\psi_1}\right)\left(\frac{\partial\Phi}{\partial\psi_0}\right) C_{\mathrm{wG}}^{\mathrm{OLS}}.$$

$\square$

## B.5 Semiparametric Efficient Estimators under Exchangeability of conditional outcome

*Proof of Proposition 3.* In this proof, we use the usual machinery of influence function computation, as described for instance in [26]. In particular, we will for the sake of the computation, assume that $X$ is a categorical variable taking value in a countable space. We first recall that if

$$\psi(P_{\mathrm{obs}}) := \mathbb{E}_{P_{\mathrm{obs}}}[h(Z)|\mathcal{A}] \tag{34}$$

for some measurable function $h$ and some event of positive mass $\mathcal{A}$, then the influence function of $\psi$ at $P_{\mathrm{obs}}$ is given by

$$\mathrm{IF}(\psi)(Z) := \frac{\mathbb{1}\{\mathcal{A}\}}{P_{\mathrm{obs}}(\mathcal{A})}(h(Z) - \psi). \tag{35}$$

We now rewrite $\psi_a$ using functional of the form (34):

$$\begin{aligned}
\psi_a &:= \mathbb{E}_P[Y^{(a)}|S = 0] \\
&= \mathbb{E}_{P_{\mathrm{obs}}}[\mathbb{E}_P[Y^{(a)}|X, S = 0]|S = 0] \\
&= \mathbb{E}_{P_{\mathrm{obs}}}[\mathbb{E}_P[Y^{(a)}|X, S = 1]|S = 0] \\
&= \mathbb{E}_{P_{\mathrm{obs}}}[\mathbb{E}_{P_{\mathrm{obs}}}[Y|X, S = 1, A = a]|S = 0] \\
&= \sum_{x \in \mathcal{X}} \mathbb{E}_{P_{\mathrm{obs}}}[\mathbb{1}\{X = x\}|S = 0] \times \mathbb{E}_{P_{\mathrm{obs}}}[Y|X = x, S = 1, A = a].
\end{aligned}$$

Using (35), and usual properties of the influence functions [26, Sec 3.4.3], we find

$$\mathrm{IF}(\psi_a)(Z) = \sum_{x \in \mathcal{X}} \frac{1 - S}{1 - \alpha}(\mathbb{1}\{X = x\} - \mathbb{E}_{P_{\mathrm{obs}}}[\mathbb{1}\{X = x\}|S = 0]) \times \mathbb{E}_{P_{\mathrm{obs}}}[Y|X = x, S = 1, A = a]$$

$$+ \sum_{x \in \mathcal{X}} \mathbb{E}_{P_{\mathrm{obs}}}[\mathbb{1}\{X = x\}|S = 0] \times \frac{\mathbb{1}\{X = x, S = 1, A = a\}}{P_{\mathrm{obs}}(X = x, S = 1, A = a)}(Y - \mathbb{E}_{P_{\mathrm{obs}}}[Y|X = x, S = 1, A = a]).$$

The first sum simply rewrites

$$\frac{1 - S}{1 - \alpha}(\mu_{(a)}(X) - \Psi_a(P)),$$

and for the second, notice that, using that $A$ and $S$ are independent:

$$\frac{\mathbb{E}_{P_{\mathrm{obs}}}[\mathbb{1}\{X = x\}|S = 0]}{P(X = x, S = 1, A = a)} = \frac{r(X)}{\alpha P(A = a)},$$

so that the sum rewrites

$$\frac{\mathbb{1}\{S = 1, A = a\}}{\alpha P(A = a)}r(X)(Y - \mu_{(a)}(X)).$$

In the end, we indeed find

$$\mathrm{IF}(\psi_a)(Z) = \frac{1 - S}{1 - \alpha}(\mu_{(a)}(X) - \psi_a) + \frac{S\mathbb{1}\{A = a\}}{\alpha P(A = a)}r(X)(Y - \mu_{(a)}(X)).$$

$\square$

*Proof of Proposition 5.* By conditioning with respect to the randomness of $\widehat{\mu}_{(a)}$ and $\widehat{r}$, we can treat these functions as deterministic. By using the law of large number, we see that $m/N$ goes to $1 - \alpha$ and that $\widehat{\psi}_{(a)}$ converges towards

$$\frac{1}{1-\alpha}\mathbb{E}[(1-S)\widehat{\mu}_{(a)}(X)] + \frac{1}{\alpha}\mathbb{E}\left[S\widehat{r}(X)(Y^a - \widehat{\mu}_{(a)}(X))\right].$$

If $\widehat{\mu}_{(a)} = \mu_{(a)}$, then the second term in the above formula cancels and we are left with

$$\frac{1}{1-\alpha}\mathbb{E}[(1-S)\widehat{\mu}_{(a)}(X)] = \mathbb{E}[\widehat{\mu}_{(a)}(X)|S=0] = \psi_a.$$

If $\widehat{r} = r$, then the second term yields

$$\mathbb{E}\left[r(X)(Y - \widehat{\mu}_{(a)}(X)|S=1\right] = \mathbb{E}\left[Y^a - \widehat{\mu}_{(a)}(X)|S=0\right] = \psi_a - \mathbb{E}\left[\widehat{\mu}_{(a)}(X)|S=0\right],$$

which also yields the results. Using continuity of $\Phi$ allows to conclude. $\qquad\square$

### B.6 Computations of one-step estimators and estimating equation estimators for the OR.

Note that $\widehat{\psi}_a^{\mathrm{EE}}$ is solution to $\sum \varphi_a(Z_i, \widehat{\eta}, \widehat{\psi}_a^{\mathrm{EE}}) = 0$. Then, given the expression of $\varphi_a$ in Proposition 3, it holds that for any other estimator $\widehat{\psi}_a$:

$$\sum_{i=1}^{N} \varphi_a(Z_i, \widehat{\eta}, \widehat{\psi}_a) = -\frac{m}{1-\alpha}(\widehat{\psi}_a - \widehat{\psi}_a^{\mathrm{EE}}).$$

We will use this observation in the subsequent computations.

(RD) For the risk difference, the estimating equation approach yields:

$$\widehat{\tau}_{\mathrm{RD}}^{\mathrm{EE}} = \widehat{\psi}_1^{\mathrm{EE}} - \widehat{\psi}_0^{\mathrm{EE}}.$$

The one step approach, based on initial estimators $\widehat{\psi}_1$ and $\widehat{\psi}_0$ yields an estimate of the form:

$$\widehat{\tau}_{\mathrm{RD}}^{\mathrm{OS}} = \widehat{\psi}_1 - \widehat{\psi}_0 + \frac{m}{N(1-\alpha)}(\widehat{\psi}_1^{\mathrm{EE}} - \widehat{\psi}_1) - \frac{m}{N(1-\alpha)}(\widehat{\psi}_0^{\mathrm{EE}} - \widehat{\psi}_0)$$
$$= \frac{m}{N(1-\alpha)}\widehat{\tau}_{\mathrm{RD}}^{\mathrm{EE}} + \left(1 - \frac{m}{N(1-\alpha)}\right)\widehat{\tau}_{\mathrm{RD}}$$

In particular, we see that starting from estimators $\widehat{\psi}_a$ of the form $\sum_{S_i=0} \widehat{\mu}_{(a)}(X_i)$ yields a final estimator that has the same structure as the estimating equation estimator, up to scaling factors depending on $\alpha, N$ and $m$ that are asymptotically close to 1.

(RR) For the risk ratio, the estimating equation approach yields:

$$\widehat{\tau}_{\mathrm{RR}}^{\mathrm{EE}} = \frac{\widehat{\psi}_1^{\mathrm{EE}}}{\widehat{\psi}_0^{\mathrm{EE}}}.$$

In contrast, the one step approach, based on initial estimators $\widehat{\psi}_1$ and $\widehat{\psi}_0$ yields an estimate of the form:

$$\widehat{\tau}_{\mathrm{RR}}^{\mathrm{OS}} = \frac{\widehat{\psi}_1}{\widehat{\psi}_0} + \frac{1}{\widehat{\psi}_0}\frac{m}{N(1-\alpha)}(\widehat{\psi}_1^{\mathrm{EE}} - \widehat{\psi}_1) - \frac{\widehat{\psi}_1}{\widehat{\psi}_0^2}\frac{m}{N(1-\alpha)}(\widehat{\psi}_0^{\mathrm{EE}} - \widehat{\psi}_0).$$

In particular, we see that in general, $\widehat{\tau}_{\mathrm{RR}}^{\mathrm{OS}} \neq \widehat{\tau}_{\mathrm{OR}}^{\mathrm{EE}}$, unless of course we initially picked $\widehat{\psi}_a^{\mathrm{EE}}$ to apply the one-step correction.

(OR) For the odds ratio, the estimating equation approach yields:

$$\widehat{\tau}_{\mathrm{OR}}^{\mathrm{EE}} = \frac{\widehat{\psi}_1^{\mathrm{EE}}}{1 - \widehat{\psi}_1^{\mathrm{EE}}}\frac{1 - \widehat{\psi}_0^{\mathrm{EE}}}{\widehat{\psi}_0^{\mathrm{EE}}}.$$

In contrast, the one step approach, based on initial estimators $\widehat{\psi}_1$ and $\widehat{\psi}_0$ yields an estimate of the form:

$$\widehat{\tau}_{\mathrm{OR}}^{\mathrm{OS}} = \frac{\widehat{\psi}_1}{1 - \widehat{\psi}_1} \frac{1 - \widehat{\psi}_0}{\widehat{\psi}_0} + \frac{1}{(1 - \widehat{\psi}_1)^2} \frac{1 - \widehat{\psi}_0}{\widehat{\psi}_0} \frac{m}{N(1 - \alpha)} (\widehat{\psi}_1 - \widehat{\psi}_1^{\mathrm{EE}})$$
$$- \frac{\widehat{\psi}_1}{1 - \widehat{\psi}_1} \frac{1}{\widehat{\psi}_0^2} \frac{m}{N(1 - \alpha)} (\widehat{\psi}_0 - \widehat{\psi}_0^{\mathrm{EE}}).$$

In particular, we see that in general, $\widehat{\tau}_{\mathrm{OR}}^{\mathrm{OS}} \neq \widehat{\tau}_{\mathrm{OR}}^{\mathrm{EE}}$, unless of course we initially picked $\widehat{\psi}_a^{\mathrm{EE}}$ to apply the one-step correction.

## C Transporting/Reweighting a causal effect under exchangeability of CATE

### C.1 Identification under exchangeability of conditional outcome

**Risk Difference (RD)**

$$\tau_{\mathrm{RD}}^{\mathrm{T}} = \mathbb{E}_{\mathrm{T}} \left[ \tau_{\mathrm{RD}}^{\mathrm{S}}(X) \right]$$

**Risk Ratio (RR)**

$$\tau_{\mathrm{RR}}^{\mathrm{T}} = \frac{\mathbb{E}_{\mathrm{T}} \left[ \tau_{\mathrm{RR}}^{\mathrm{S}}(X) \cdot \mu_{(0)}^{\mathrm{T}}(X) \right]}{\mathbb{E}_{\mathrm{T}} \left[ Y^{(0)} \right]}$$

**Odds Ratio (OR)**

$$\tau_{\mathrm{OR}}^{\mathrm{T}} = \left( \frac{\mathbb{E}_{\mathrm{T}} \left[ Y^{(0)} \right]}{1 - \mathbb{E}_{\mathrm{T}} \left[ Y^{(0)} \right]} \right)^{-1} \cdot \frac{\mathbb{E}_{\mathrm{T}} \left[ \frac{\tau_{\mathrm{OR}}^{\mathrm{S}}(X) \cdot \mu_{(0)}^{\mathrm{T}}(X)}{1 + \tau_{\mathrm{OR}}^{\mathrm{S}}(X) \cdot \mu_{(0)}^{\mathrm{T}}(X) - \mu_{(0)}^{\mathrm{T}}(X)} \right]}{1 - \mathbb{E}_{\mathrm{T}} \left[ \frac{\tau_{\mathrm{OR}}^{\mathrm{S}}(X) \cdot \mu_{(0)}^{\mathrm{T}}(X)}{1 + \tau_{\mathrm{OR}}^{\mathrm{S}}(X) \cdot \mu_{(0)}^{\mathrm{T}}(X) - \mu_{(0)}^{\mathrm{T}}(X)} \right]}$$

### C.2 Semiparametric efficient estimators under exchangeability of treatment effect

*Proof of Proposition 6.* We use the same tricks as in the proof of Proposition 3. We first notice that

$$\psi_1^{\mathrm{T}} = \sum_{x \in \mathcal{X}} P_{\mathrm{obs}}(X = x | S = 0) \Gamma(\tau_{\Phi}(x), \mu_{(0)}^{\mathrm{T}}(x)),$$

so that, with a slight abuse of notation

$$\mathrm{IF}(\psi_1^{\mathrm{T}}) = \sum_{x \in \mathcal{X}} \frac{1 - S}{1 - \alpha} (\mathbb{1}\{X = x\} - P_{\mathrm{obs}}(X = x | S = 0)) \Gamma(\tau_{\Phi}(x), \mu_{(0)}^{\mathrm{T}}(x))$$
$$+ \sum_{x \in \mathcal{X}} P_{\mathrm{obs}}(X = x | S = 0) \mathrm{IF}(\tau_{\Phi}(x)) \partial_1 \Gamma(\tau_{\Phi}(x), \mu_{(0)}^{\mathrm{T}}(x)).$$

Using that $\tau_{\Phi}(x) = \Phi(\mathbb{E}[Y | A = 1, S = 1, X = x], \mathbb{E}[Y | A = 0, S = 1, X = x])$, we further find that

$$\mathrm{IF}(\tau_{\Phi}(x)) = \frac{\mathbb{1}\{A = 1, S = 1, X = x\}}{P(A = 1, S = 1, X = x)} (Y - \mu_{(1)}^{\mathrm{S}}(x)) \partial_1 \Phi(\mu_{(1)}^{\mathrm{S}}(x), \mu_{(0)}^{\mathrm{S}}(x))$$
$$+ \frac{\mathbb{1}\{A = 0, S = 1, X = x\}}{P(A = 0, S = 1, X = x)} (Y - \mu_{(0)}^{\mathrm{S}}(x)) \partial_0 \Phi(\mu_{(1)}^{\mathrm{S}}(x), \mu_{(0)}^{\mathrm{S}}(x))$$

Again, we know that

$$P(A = a, X = x, S = 1) = \alpha \pi P(X = x | S = 1) = \alpha \pi r(X) P(X = x | S = 0).$$

Furthermore, since $\Gamma(\Phi(a, b), b) = a$ for all $a, b$, differentiating with respect to $a$ or $b$ yields $\partial_0 \Phi(a, b) \partial_1 \Gamma(\Phi(a, b), b)) + \partial_0 \Gamma(\Phi(a, b), b) = 0$ and $\partial_1 \Phi(a, b) \partial_1 \Gamma(\Phi(a, b), b) = 1$. Patching all of this together yields the result. $\square$

## C.3 Computations of one-step estimators and estimating equation estimators under exchangeability of CATE

*Example* 3 (Application to the usual causal measures.). We give the expression of $\widehat{\psi}_1^{\mathrm{EE}}$ for the most usual causal measures.

(RD) For the risk difference, we find

$$\widehat{\psi}_1^{\mathrm{EE}} = \frac{1}{m} \sum_{S_i=0} \mu_{(0)}^{\mathrm{T}}(X_i) + \widehat{\tau}_\Phi(X_i)$$
$$+ \frac{1-\alpha}{\alpha m} \sum_{S_i=1} \widehat{r}(X_i) \left( \frac{A_i}{\pi}(Y_i - \widehat{\tau}_\Phi(X_i) - \widehat{\mu}_{(0)}^{\mathrm{S}}(X_i)) - \frac{1-A_i}{1-\pi}(Y_i - \widehat{\mu}_{(0)}^{\mathrm{S}}(X_i)) \right).$$

(RR) For the risk ratio, we find:

$$\widehat{\psi}_1^{\mathrm{EE}} = \frac{1}{m} \sum_{S_i=0} \mu_{(0)}^{\mathrm{T}}(X_i)\widehat{\tau}_\Phi(X_i)$$
$$+ \frac{1-\alpha}{\alpha m} \sum_{S_i=1} \widehat{r}(X_i) \left( \frac{A_i}{\pi}(Y_i - \widehat{\mu}_{(0)}^{\mathrm{S}}(X_i)\widehat{\tau}_\Phi(X_i)) - \frac{1-A_i}{1-\pi}(Y_i - \widehat{\mu}_{(0)}^{\mathrm{S}}(X_i))\widehat{\tau}_\Phi(X_i) \right).$$

(OR) For the odds ratio, we find:

$$\widehat{\psi}_1^{\mathrm{EE}} = \frac{1}{m} \sum_{S_i=0} \frac{\mu_{(0)}^{\mathrm{T}}(X_i)\widehat{\tau}_\Phi(X_i)}{1 - \mu_{(0)}^{\mathrm{T}}(X_i) + \mu_{(0)}^{\mathrm{T}}(X_i)\widehat{\tau}_\Phi(X_i)}$$
$$+ \frac{1-\alpha}{\alpha m} \sum_{S_i=1} \widehat{r}(X_i) \left( \frac{A_i}{\pi}\left(Y_i - \frac{\mu_{(0)}^{\mathrm{S}}(X_i)\widehat{\tau}_\Phi(X_i)}{1 - \mu_{(0)}^{\mathrm{S}}(X_i) + \mu_{(0)}^{\mathrm{S}}(X_i)\widehat{\tau}_\Phi(X_i)} \right) \right.$$
$$\left. - \frac{1-A_i}{1-\pi}(Y_i - \widehat{\mu}_{(0)}^{\mathrm{S}}(X_i)) \frac{\widehat{\tau}_\Phi(X_i)}{(1 - \widehat{\mu}_{(0)}^{\mathrm{S}}(X_i) + \widehat{\mu}_{(0)}^{\mathrm{S}}(X_i)\widehat{\tau}_\Phi(X_i))^2} \right).$$

# D   Simulation

For the simulations we have implemented all estimators in Python using Scikit-Learn for our regression and classification models. All our experiments were run on a 8GB M1 Mac.

**Linear setting under Assumption 3:** we evaluate estimators under a linear response surface:

$$\mu_s^{(a)}(V) = \beta_a^\top V \quad \text{with} \quad \beta_1 = (0.5, 1.2, 1.1, 3.3, -0.6) \quad \text{and} \quad \beta_0 = (-0.2, -0.6, 0.6, 1.7, 0.3).$$

Since $\beta_0$ and $\beta_1$ remain unchanged across the source and target domains, Assumption 3 is satisfied, results are depicted in Figure 4. As expected from the linear generative process, all estimators perform well across all measures, with the transported and weighted G-formula exhibiting particularly low variance in this setting—outperforming the influence function–based estimators.

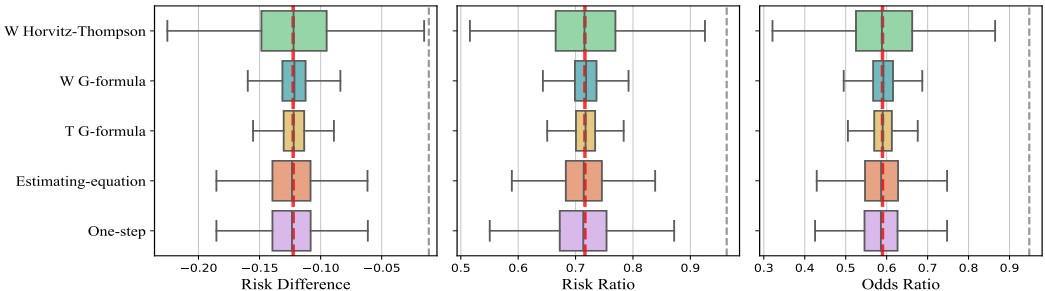

Figure 4: Comparison of estimators across different causal measures under a linear outcome model with a sample size of $N = 50{,}000$ and 3,000 repetitions.

