# OpenReview forum: "A Unified Framework for the Transportability of Population-Level Causal Measures"
_NeurIPS.cc/2025/Conference — NeurIPS 2025 poster_

### Official Review · Reviewer_nTva · 2025-06-22

**Clarity:** 3
**Significance:** 3
**Originality:** 3
**Rating:** 5
**Confidence:** 3

**Summary:**

This paper introduces a unified framework for transporting a wide range of population causal effect measures from randomized controlled trials (RCTs) to target populations under covariate shift. It provides identification results under two key assumptions (exchangeability in mean and in effect measure), and derives classical, semiparametric, and doubly-robust estimators, revealing that estimators often differ across causal measures due to nonlinearity.

**Questions:**

Do you expect significant performance differences between your estimators in small or moderate sample sizes, especially when nuisance functions are estimated with limited data?

**Ethical Concerns:**

["NO or VERY MINOR ethics concerns only"]

**Final Justification:**

I stand by my original recommendation to accept the paper. The authors have addressed my concerns about small-sample behavior with additional simulations. Assuming the correctness of Lemma 2 (which I reviewed based on their rebuttal and found reasonable) the contributions of this paper remain solid.

**Limitations:**

yes

**Quality:**

3

**Strengths And Weaknesses:**

**Strengths**:
- The paper generalizes causal transportability to a wide class of first-moment estimands, which is a notable expansion beyond existing methods
- The theoretical claims are convincingly supported
- The paper is well written and easy to follow



**Weaknesses**:

- All the simulations use large sample sizes ($ n=50,000$), masking potential pathologies under finite or moderate samples, such as weight instability, overfitting of nuisance models, or violation of positivity. Arguably, most trials have significantly smaller sample sizes and therefore it would be valuable to evaluate the proposed estimators in a small sample setting

---

> ### Author Rebuttal · Authors · 2025-07-30
>
> Thank you for your thoughtful and constructive review. We are glad that you found the contributions of the paper valuable and the exposition clear.
>
> Regarding your comment on small or moderate sample sizes: while our theoretical results are asymptotic, the methodology itself is not restricted to large samples. In practice, when RCT data is limited and nuisance functions (e.g., outcome or propensity models) must be estimated from this same limited data, a pragmatic approach would be to use parametric models to reduce variance and mitigate overfitting.
>
> To assess finite-sample behaviour empirically, we conducted simulations with total sample size \$N = 5000\$ and \$500\$, of which respectively 1,500 and 150 for the RCT group. The results in Figure 4 (first table) correspond to the setting under exchangeability in mean, and those in Figure 2 (second table) to the setting under exchangeability in effect measure. Across both settings, the **Estimating-equation** estimator—whether standard or Γ-adjusted—shows the best empirical performance, achieving the lowest combination of bias and variance across RD, RR, and OR. While in some specific cases it may be marginally outperformed by a G-formula or a One-step estimator, the **Estimating-equation remains doubly robust**, offering consistent performance even under model misspecification. For this reason, we strongly recommend its use in practice, particularly when nuisance functions are estimated from limited data.
>
> That said, we agree that further exploration of finite-sample properties would be valuable. In particular, understanding how the sizes of both the source (RCT) and target populations affect the performance of different estimators—especially those based on efficient influence functions—is a promising direction for future research. We have added a sentence to the conclusion to reflect this.
>
> **N=5000**
>
> | Estimator           | RD                  | RR                  | OR                  |
> | :------------------ | :------------------ | :------------------ | :------------------ |
> | Estimating-equation | **0.0033 ± 0.0867** | 0.0098 ± 0.1916     | **0.0021 ± 0.2762** |
> | One-step            | 0.0049 ± 0.0868     | **0.0027 ± 0.2003** | 0.0024 ± 0.2771     |
> | T G-formula         | 0.0115 ± 0.0798     | 0.0198 ± 0.1780     | 0.0208 ± 0.2606     |
> | W G-formula         | 0.0604 ± 0.0559     | 0.0991 ± 0.1555     | 0.1449 ± 0.2172     |
> | W Horvitz-Thompson  | 0.0669 ± 0.0581     | 0.0001 ± 0.2894     | 0.0718 ± 0.3586     |
> | W Neyman        | 0.0759 ± 0.0557     | 0.0438 ± 0.2698     | 0.1261 ± 0.3320     |
>
>
> | Estimator             | RD                  | RR                  | OR                   |
> | :-------------------- | :------------------ | :------------------ | :------------------- |
> | Estimating-equation   | **0.0015 ± 0.0383** | 0.1293 ± 0.0648     | 0.0536 ± 0.1317      |
> | W Horvitz-Thompson    | 0.0435 ± 0.0349     | -0.0288 ± 0.0555    | -0.0885 ± 0.1501     |
> | W Neyman              | 0.0448 ± 0.0296     | -0.0314 ± 0.0493    | -0.0963 ± 0.1315     |
> | W Γ-formula           | 0.1573 ± 0.0278     | -0.1327 ± 0.0328    | -0.1343 ± 0.0884     |
> | Γ Estimating-equation | **0.0015 ± 0.0383** | **0.0113 ± 0.0678** | **-0.0119 ± 0.1109**     |
> | Γ One-step            | **0.0015 ± 0.0383** | **0.0113 ± 0.0678** | -0.0124 ± 0.1108 |
>
>
> **N=500**
>
> | Estimator           | RD                   | RR                   | OR                  |
> | :------------------ | :------------------- | :------------------- | :------------------ |
> | Estimating-equation | **-0.0153 ± 0.1354** | **-0.0664 ± 0.5310**     | 0.1662 ± 1.7489     |
> | One-step            | -0.0157 ± 0.1356     | -0.0956 ± 0.4791     | 0.1307 ± 1.7063 |
> | T G-formula         | -0.0210 ± 0.1233     | -0.1130 ± 0.4482 | **-0.0072 ± 1.4014**    |
> | W G-formula         | -0.2270 ± 0.0876     | -0.7003 ± 0.1600     | -1.5368 ± 0.3619    |
> | W Horvitz-Thompson  | -0.2100 ± 0.0811     | -0.6106 ± 0.4439     | -1.4052 ± 0.6020    |
> | W Neyman            | -0.2073 ± 0.0707     | -0.6106 ± 0.4124     | -1.4104 ± 0.5589    |
>
>
>
> | Estimator             | RD                  | RR                  | OR                   |
> | :-------------------- | :------------------ | :------------------ | :------------------- |
> | Estimating-equation   | **0.0062 ± 0.0999** | 0.0154 ± 0.1457     | 0.1330 ± 0.5173      |
> | W Horvitz-Thompson    | 0.0647 ± 0.0879     | -0.0835 ± 0.1988    | 0.0416 ± 0.5925      |
> | W Neyman              | 0.0569 ± 0.0793     | -0.0968 ± 0.1759    | -0.0207 ± 0.4897     |
> | W Γ-formula           | 0.1586 ± 0.0754     | 0.0688 ± 0.2283     | -0.1641 ± 0.2978     |
> | T Γ-formula           | 0.0097 ± 0.0892     | 0.0971 ± 0.2766     | -0.0157 ± 0.3539 |
> | Γ Estimating-equation | **0.0063 ± 0.0998**     | **0.0871 ± 0.3054** | **-0.0044 ± 0.3654** |
> | Γ One-step            | **0.0062 ± 0.0998**     | **0.0870 ± 0.3053** | -0.0157 ± 0.3603 |

---

> > ### Comment · Area_Chair_9pc5 · 2025-08-02
> >
> > @reviewer: Could you please review the authors’ rebuttal and share your feedback? The other reviews and rebuttals also contain useful information that may help you refine your assessment of the paper, so it would be helpful if you could look at them as well when you have a chance.

---

> > > ### Comment · Reviewer_nTva · 2025-08-03
> > >
> > > I stand by my original recommendation to accept the paper. The authors have addressed my concerns about small-sample behavior with additional simulations. Assuming the correctness of Lemma 2 (which I reviewed based on their rebuttal and found reasonable) the contributions of this paper remain solid.

---

### Official Review · Reviewer_247y · 2025-06-25

**Clarity:** 1
**Significance:** 3
**Originality:** 2
**Rating:** 3
**Confidence:** 3

**Summary:**

This paper aims to generalize transportability theory from average treatment effects to a broader class of first-moment population-level causal measures. The authors derive identification results under two regimes—exchangeability in mean and in effect measure—and propose several estimators (classical, regression-based, semiparametric). They show empirically that estimator performance diverges sharply across nonlinear estimands and assumptions.

**Questions:**

- What if the observed covariates are not fully overlapping among training and target experiment?  What if some covariates are changing distribution on target but they are not directly observed? (see PPCI formulation [1] for reference).
- Are the assumptions and estimators scaling to an higher number of observations? The real world experiments only consider 6 covariates.

[1] Causal Lifting of Neural Representations: Zero-Shot Generalization for Causal Inferences, arxiv, Cadei et al., 2025

**Ethical Concerns:**

["NO or VERY MINOR ethics concerns only"]

**Final Justification:**

The rebuttal addresses some theoretical points and acknowledges important limitations. However, the empirical evidence remains very limited and underspecified, with no demonstration on realistic or robust settings. Additionally, while the authors mention providing code, no link has been made available, which hinders reproducibility and transparency. Given these unresolved issues, especially the weak empirical support, I maintain my recommendation to reject.

**Limitations:**

See questions.

**Paper Formatting Concerns:**

Type error:
- l55: 'NNTetc.)  '
- l172: exists \exists

Citations:
- use \citet{} for inline references.

**Quality:**

2

**Strengths And Weaknesses:**

*Strengths*
- Motivation: The paper tackles an under-addressed problem (generalising nonlinear causal estimands across populations) motivated by several real-world application, far beyond medicine.
- Method: The paper is methodologically rich, providing numerous instantiations of classical and doubly robust estimators.

*Weaknesses*:
- The introduction is difficult to follow. Several concepts are not properly introduced, e.g.:
   - l27 the risk difference is introduced with no definition or assumptions on the setting (binary outcome?).
   - l54-55 'collapsible and not collapsible measures' are not defined.

- Imprecisions/Error (?):
  - Risk ratio assumes continuous outcome for interpretation, but I cannot find where/if assumed in Section 2. Similarly Odds ratio etc.
  - l110 m and n are DEPENDENT binomials (use multimodal distribution instead).
  - Finishing the intro was still unclear if some transportability assumptions are required. I find them crucial and I would suggest to better stress them in the introduction too.
- Experiments
  - Synthetic experiments are under-defined and no code is provided. How many covariates are considered (d)? What are the source values in Figure 1 caption?
  - I find difficult to interpret the objective of the experiments. Additionally I find unclear how to evaluate the real world experiment results in Figure 3.

Overall I appreciate the attempt of this work in formalising transportability of population-level causal measures but I find the writing still immature, containing imprecisions with unclear story conclusions.

---

> ### Author Rebuttal · Authors · 2025-07-30
>
> We thank you for your carefull review. We will fix the several typos you pointed out.
>
> **Covariate Non-Overlap Between Source and Target Populations**
>
> The overlap assumption is critical for identifying causal effects in a target population. It requires that for any covariate pattern present in the target population, there is sufficient representation in the trial population ($P(S = 1 | X = x) > 0$). When this assumption is violated—i.e., when the observed covariates are not fully overlapping between training (RCT) and target populations—standard identification results no longer hold, and estimators proposed in the paper may become biased or unstable due to extreme or poorly estimated weights.
>
> There are two main strategies to address this:
>
> 1. **Restricting the target population**: One approach is to redefine the generalization target to individuals in the target population who are within the support of the RCT sample—that is, those who could plausibly have been enrolled in the RCT. This amounts to generalizing the effect only to the subpopulation with overlapping covariates, thereby restoring identifiability at the cost of external validity (the target population has changed).
>
> 2. **Performing sensitivity analyses**: An alternative is to formally account for violations of overlap through sensitivity analyses, which quantify the robustness of conclusions under deviations from the assumption. See for instance Huang (2025) for recent developments in this direction.
>
>
> **Unobserved Shifting Covariates**
>
> If unobserved covariates drive differences in outcome distributions between the source and target populations, the assumption of exchangeability in mean is violated, undermining identification based on outcome modeling. In such cases, a weaker but potentially more realistic assumption—exchangeability in the effect measure—can still hold. Indeed, if the unobserved covariates influence only the outcome level but not the treatment effect heterogeneity, transportability of the effect measure is still possible. However, identification under this assumption requires access to control outcomes in the target population, which may not be feasible in many  scenarios. If shifted treatment effect modifiers are not observed, then, this boils down to an identifiability issue and consequently, one has to resort to traditional methods to give bounds on treatment effects, for instance using sensitivity analysis. We added a paragraph in the discussion section on limitations, and the verification of assumptions.
>
>
> **Scalability of Estimators with Increasing Covariate Dimensions**
>
> It’s true that the example uses only six covariates, but this reflects the limited data collected in the clinical trial. In randomized trials, covariates are often less critical, since treatment effects can be estimated directly by comparing average outcomes between treated and control groups. The six covariates used are those shared between the trial and observational data, and which show distributional differences across the two datasets.
>
> That said, there’s no theoretical limitation on the number of covariates. The estimators rely on the estimation of nuisance functions—such as regressions of $Y$ on $X$—which can be modeled using parametric (e.g., linear regression) or nonparametric methods (e.g., random forests). When using g-formula-type estimators, more data is required as the number of covariates increases, especially with nonparametric methods due to slower convergence rates. However, doubly robust estimators can recover parametric convergence rates even in high-dimensional settings.
>
>
> ### Weaknesses
>
>
> **1. "The introduction is difficult to follow. Several concepts are not properly introduced, e.g.: l27 the risk difference is introduced with no definition or assumptions on the setting (binary outcome?). l54-55 'collapsible and not collapsible measures' are not defined."**
>
> We thank the reviewer for this comment. We have revised the introduction for clarity and now explicitly define key concepts such as the *risk difference* and *collapsible vs. non-collapsible measures*, including references for further reading.
>
> **2. "Risk ratio assumes continuous outcome for interpretation, but I cannot find where/if assumed in Section 2. Similarly Odds ratio etc."**
>
> The *risk ratio* can be interpreted for both binary and continuous outcomes (as long as the outcome remains positive), while the *odds ratio* applies only to binary outcomes. Our results hold in the continuous and binary setting, which we have now clarified in Section 2.
>
> **3. "l110 m and n are DEPENDENT binomials (use multimodal distribution instead)."**
>
> We infer you meant multinomial and not multimodal. You are right , $(n, m)$ follow a **multinomial distribution** with parameter $N$ and probabilities $(\alpha, 1 - \alpha)$.
>
> **5. "Synthetic experiments are under-defined and no code is provided. How many covariates are considered (d)? What are the source values in Figure 1 caption?"**
>
> Thank you for the feedback. We now also provide a link to the experimental code for reproducibility. We have updated the experiments section to clarify that:
>
> * The number of covariates (*d*) is **5**, consistent with real-world RCTs that often involve few covariates.
> * The *source value* in Figure 1 denotes the **true treatment effect in the source population**, shown to illustrate the shift in effect when applied to the target population.

---

### Official Review · Reviewer_naME · 2025-06-28

**Clarity:** 3
**Significance:** 3
**Originality:** 2
**Rating:** 4
**Confidence:** 4

**Summary:**

This paper studies the generalizability of several nonlinear causal estimands under distribution shift, proposing a family of estimators under mean exchangeability and effect measure exchangeability conditions. When mean exchangeability holds, the authors propose weighted, regression, and doubly robust strategies for transporting an estimand that is a known mapping of the mean of two potential outcomes. In addition, the authors discuss generalization strategies for such estimands based on the inverse mapping. Theoretical analysis shows the inferential guarantees of the proposed estimators, and the performance of them is demonstrated via simulations and real data analysis.

**Questions:**

1. How would you interpret the discrepancy in the messages in the real experiments? Since the truth is not available, it would be helpful to investigate why there is a discrepancy to decide whether it makes sense.

2. I think connections to existing results should be more explicitly discussed to more clearly position the contributions. In particular, [1] discusses quite a few identification strategies for nonlinear causal estimands. How does this work compare to them? If all identification strategies have been studied and this work just focuses on estimators, then the estimators would look not very novel/surprising given the existing results, and this work might be better suited to a (bio)stats journal with more in-depth simulations and real data analysis to show the performance of various estimators in diverse conditions.


[1] Colnet, B., J. Josse, G. Varoquaux, and E. Scornet (2023). Risk ratio, odds ratio, risk difference... which causal measure is easier to generalize? arXiv preprint arXiv:2303.16008.

**Ethical Concerns:**

["NO or VERY MINOR ethics concerns only"]

**Final Justification:**

The author's response clarified the contributions which I had concerns about.

**Limitations:**

Yes.

**Quality:**

3

**Strengths And Weaknesses:**

Strength:

1. This paper studies an important problem of generalizing causal estimands.
2. The results are solid and rich, covering identification, estimation, and inference.
3. Numerical results show promising performance.

Weakness:

1. The contributions need to be more clearly stated by comparing to relevant literature (see Questions).
2. The real data analysis has vague message.

---

> ### Author Rebuttal · Authors · 2025-07-30
>
> ## Clarification on discrepancy in the messages in the real experiments
>
> Thank you for this comment. We have clarified the real data analysis in the revised manuscript. While the RCT analysis (not included in the paper; see Colnet et al., 2024) suggested a slightly positive effect of the treatment, the estimators applied to the target population suggest a slightly deleterious—but non-significant—effect. This discrepancy is expected due to differences in the underlying populations: the RCT population differs from the target population used in generalization.
>
> All causal estimators applied to the target population—whether based on generalizing from the RCT or using purely observational data—converge on the same conclusion: a non-significant effect in the Traumabase cohort. As you rightly point out, the true effect is unknown. However, because we observe outcomes in the target population, it is possible to directly estimate treatment effects from the observational data (Mayer et al., 2020). While this relies on different assumptions—particularly, the absence of unmeasured confounding—it is reassuring that both approaches (generalization from RCT and causal inference from observational data) lead to the same conclusion. This is true for the risk difference as well as relative measures (Boughdiri et al., 2025).
>
> In addition, if one suspects that some identifiability assumptions are not met, it is possible to resort to sensitivity analysis.
>
>
> ## Position with Colnet et al. (2023)
>
> We thank the reviewer for this comment and appreciate the opportunity to clarify the novelty and scope of our contributions relative to prior work, particularly to Colnet et al. (2023).
>
> While Colnet et al. (2023) explores identification strategies for causal estimands—most notably the risk ratio and the risk difference—in the context of generalization, **our work extends the identification and estimation landscape in several important directions**:
>
> ---
>
> ### **1. Unified Framework for Transportability**
>
> **We introduce a general identification and estimation framework for transporting a broad class of first-moment causal estimands**—including more than a dozen measures, both **collapsible** (e.g., RD, RR) and **non-collapsible** (e.g., OR)—across covariate shifts. Specifically:
>
> * Under exchangeability in mean, we **recover and extend existing identification results** and **derive asymptotic properties** for all proposed estimators. For instance, under a logistic selection and linear outcome model, we show that the **transported G-formula is asymptotically more efficient than the Horvitz–Thompson estimator**.
>
> * Under exchangeability in effect measure, a **strictly weaker assumption**, we provide **a different result from Colnet, which is more general**  to cover **any first-moment causal estimand**, including **non-collapsible ones like the odds ratio (OR)**. To our knowledge, these are **novel identification results** in the generalization context and are especially relevant given the widespread use of the OR in applied research.
>
> ---
>
> ### **2. New Estimators for Transported Estimands**
>
> Whereas Colnet et al. focused on identification **we propose new estimators designed for generalization from randomized trials**, including:
>
> * Semiparametric efficient estimators (one-step and estimating equation), and we **demonstrate that their behavior diverges for nonlinear estimands** such as RR and OR—**a key theoretical and empirical contribution**. Notably, we show that the **estimating equation (EE) estimator is doubly robust** for any causal measure (unlike the one-step estimator), and **consistently outperforms it empirically**. **This difference had not been previously emphasized** in the literature.
>
> * **Γ-formula estimators under exchangeability in effect measure**, which are, to our knowledge, novel contributions, especially in the context of **generalizing non-collapsible estimands**.
>
> ---
>
> ### **3. Complementary Goals to Prior Work**
>
> While Colnet et al. emphasize which causal estimands are easier to identify and generalize, our work offers a unified framework that supports the generalization of all first-moment causal measures, with a strong focus on estimation strategies and efficiency (which allows practical deployment of such methods with theoretical guarantees in contrats to their work). Their focus is on separating treatment effect from baseline; our focus reveals how many estimands share a common structure, enabling unified treatment in both identification and estimation. We see these as complementary contributions but different that together helps the understanding of causal transportability.
>
> ---
>
> ### **4. Relevance to NeurIPS**
>
> Although we agree that real-world validation is valuable, our primary goal is to offer a principled, broadly applicable framework for researchers and practitioners at the interface of machine learning, statistics, and application. Specifically, our use of influence functions, robustness analysis, and complex simulation designs (e.g., under model misspecification and nonlinear responses) directly targets challenges common in machine learning applications that require both generalizability and estimator efficiency.
>
> ---
>
> We hope to have convinced you that our contribution is both theoretical and methodological, encompassing a unified transportability framework and novel results at both the identification and estimation levels, especially for **nonlinear estimands**. We believe this goes well beyond a routine application or minor extension and is a strong fit for a generalist ML venue like NeurIPS. Furthermore, **our framework is broadly applicable across domains**—biostatistics, econometrics, and public policy—underscoring its potential impact.
>
>
> ---
>
> ### References
>
> * Colnet, B., Mayer, I., Chen, G., Dieng, A., Li, R., Varoquaux, G., Vert, J.-P., Josse, J., & Yang, S. (2024). *Causal Inference Methods for Combining Randomized Trials and Observational Studies: A Review*. Statistical Science, 39(1), 165–191.
>
> * Mayer, I., Sverdrup, E., Gauss, T., Moyer, J.-D., Wager, S., & Josse, J. (2020). *Doubly robust treatment effect estimation with missing attributes*. arXiv preprint
>
> * Boughdiri, A., Josse, J., & Scornet, E. (2025). *Quantifying Treatment Effects: Estimating Risk Ratios in Causal Inference*. ICML2025

---

> > ### Comment · Reviewer_naME · 2025-08-02
> >
> > Thanks for the response! I've updated the score to 4.

---

### Official Review · Reviewer_LpQj · 2025-07-06

**Clarity:** 3
**Significance:** 3
**Originality:** 3
**Rating:** 4
**Confidence:** 3

**Summary:**

This paper studies the problem of causal meta analysis, and proposes a unified framework for transporting a board class of first moment causal effect measures. It aims to generalize treatment effect estimates from RCTs to target populations under covariate shifts. The authors extend existing methods for generalizing causal effects by deriving identification results under conditional exchangeability assumptions. The framework incorporates both classical estimators and semiparametric efficiency theory to provide doubly-robust estimators for various causal measures. The method is evaluated through synthetic simulations and real-world applications, showcasing its utility in generalizing both absolute and relative causal effects to broader populations.

**Questions:**

Could the authors please clarify the justification for my questions on lemma2 proof? As proposition 2 seems crucially assumes correctness of lemma 2.

**Ethical Concerns:**

["NO or VERY MINOR ethics concerns only"]

**Final Justification:**

Most of my concerns has been addressed

**Limitations:**

See above.

**Quality:**

3

**Strengths And Weaknesses:**

Strength:
- In general I think formulating meta analysis under a causal framework is an important direction. One way of approaching this is to reframe the problem as generalizing treatment effects from controlled settings to some target populations that may differ in their distribution of covariates. Then, the paper's focus on a diverse set of causal measures beyond the risk difference, which is a major strength.
- The work is quite thorough. The authors systematically derived identification formulas and a complete suite of estimators (weighting, G-formula, one-step, EE) under two clearly articulated transportability assumptions. This is very organized and I like it a lot.

Weakness:
There are a few minor weaknesses especially on discussions, evaluation and potential mistakes in proofs.

-  The paper presents two parallel frameworks based on exchangeability in mean vs effect measure. The latter is weaker but imposes the strong practical requirement of having access to control outcomes for the target. The paper would be strengthened by a more detailed discussion of applied scenarios where this condition is met and more explicit guidance on how a practitioner should choose between these two sets of assumptions and corresponding estimators

-  I am not sure if lemma 2 proof is correct. Its correctness depends on the component matrix
$$
E_{T}[VV^{\top}] - E_{T}[VV^{\top}] E_{T}[\sigma(X,\beta)VV^{\top}])^{-1}E_{T}[VV^{\top}]
$$
being PSD. As the authors suggests, this matrix has the form $A-AB^{-1}A$. But I think the authors might have missed an additional condition for $A-AB^{-1}A$ to be PSD which is $B \succeq A$ should hold.

However, given the definitions $A=E_{T}[VV^{\top}]$ and $B=E_{T}[\sigma(X,\beta)VV^{\top}]$, and the fact that $\sigma(X, \beta) \in (0,1)$ (if I understand correctly, the authors can elaborate), it would seem the reverse inequality, $A \succeq B$, holds true instead.

- on eval:  In the real-world application (sec 5.2), the authors conclude that the treatment appears to have "little to no impact on the target population". However, the confidence intervals presented in Figure 3 are quite wide. The claim of "little to no impact" seems stronger than what can be concluded from the point estimates and their associated uncertainty.





**Please note: I do find I enjoy this paper, I am giving borderline reject simply due to those concerns. If the authors can address my concerns, especially the one on lemma 2, I will consider increasing the score.**

---

> ### Author Rebuttal · Authors · 2025-07-30
>
> We are grateful for the reviewer’s positive remarks. In particular, we appreciate the recognition of our effort to present a unified causal framework that accommodates different effect measures and transportability assumptions. We aimed to make the methodology both general and practically accessible. We are encouraged that this structure was found to be clear and well-organized.
>
> ---
> ## Response to Comment on Exchangeability Assumptions and Applied Guidance
>
> We thank the reviewer for this thoughtful and constructive comment. We fully agree that providing concrete examples and clearer practical guidance on when to prefer each assumption significantly improves the applied relevance of the work.
>
> ### Applied Scenarios Where Exchangeability in Effect Measure Is Plausible
>
> In pre-treatment settings, the intervention has not yet been introduced, so outcomes under usual care can be observed for all individuals. We are currently collaborating with an Inserm team to generalize the effect of paracetamol (acetaminophen) for preterm infants to prevent patent ductus arteriosus  from the RCT **TREOCAPA trial** to 23 cohorts from the **RECAP preterm** project. In this setting, outcome data under the control condition are available for the cohorts, as the infants in these cohorts did not receive paracetamol. The aim is to predict the treatment effect across these different populations. These untreated cohorts provide high-quality data on clinically important outcomes—including survival without severe morbidity, defined as the absence of major complications such as bronchopulmonary dysplasia (BPD), necrotizing enterocolitis (NEC), severe intraventricular hemorrhage (IVH), and cystic periventricular leukomalacia (cPVL).
>
> We have revised the discussion section to highlight a real-world setting in which control outcomes are available in the target population.
>
>
> ### Understanding the Assumptions and Their Plausibility
>
> We also clarify the structural differences between the two assumptions:
>
> - Exchangeability in effect measure assumes that treatment effects are portable but requires access to control outcomes in the target population and access to shifted treatment effect modifiers.
>
> - Exchangeability in mean, is stronger and assumes that conditional outcome models are stable across populations, but relies on  shifted prognostic covariates.
>
> To assess which assumption is plausible, it is possible to perform some statistical tests. To help practitioners we now include the following guidance:
>
> - **Model-based diagnostics**: To test exchangeability in mean, one can regress the outcome on covariates and a study indicator. If the study coefficient is significant, it suggests the conditional mean function differs across populations (Robertson et al., 2021).
>
> - **Illustrative example**: Suppose the potential outcomes in the source population follow
>   $$
>   \mathbb{E}_S[Y^{(a)} | X] = b(X) + a \cdot m(X),
>   $$
>   while in the target population they follow
>   $$
>   \mathbb{E}_T[Y^{(a)} | X] = \gamma + b(X) + a \cdot m(X).
>   $$
>
>   where $m(.)$ and  $b(.)$ respectively correspond to the treatment effect and the baseline. The constant $\gamma$ modifies the baseline of the target population and may be due to the fact that the target population is more likely than the source population to undergo undesirable events, due to exogeneous variables whose information is not fully contained in the covariates $X$. In this case, the local effect of the Risk Difference can be generalized but the distribution of the potential outcomes cannot.
>
> - **Identifying effect modifiers** is an ongoing research challenge, but recent methods (e.g., Hines et al., 2022; Benard et al., 2023; Paillard et al., 2025) provide tools for detecting covariates that drive treatment effect heterogeneity.
>
>
> To conclude, the choice among different approaches often depends on data availability—particularly whether outcomes in the target population are accessible. When outcome data are available, both identification assumptions can be empirically tested. Close collaboration with clinicians is essential to develop a DAG and to identify key variables: those likely to act as shifted prognostic factors, and potential treatment effect modifiers. In the latter case, specific statistical tests can also be conducted to support the analysis.
>
>
>
> We believe these additions help clarify when each assumption is likely to be valid and how practitioners can evaluate them in applied settings.
>
> ### Estimator Recommendation
>
> Regardless of which identification assumption is used, we recommend the corresponding estimating equation (EE) estimator (and not the one-step estimators): either the standard version (under exchangeability in mean) or the Γ-formula-based version (under exchangeability in effect measure). These estimators are doubly robust, providing consistent estimates if either the outcome model or the density ratio model is correctly specified. Our theoretical and empirical results show that EE estimators offer favorable bias–variance trade-offs under both assumptions.
>
> ---
>
> ## Clarification on Lemma 2: Positive Semi-Definiteness Argument
>
> You're absolutely right that the expression
>
> $$
> \mathbb{E}_T[VV^\top] - \mathbb{E}_T[VV^\top] \left( \mathbb{E}_T[\sigma(X,\beta) VV^\top] \right)^{-1} \mathbb{E}_T[VV^\top]
> $$
>
> is **not guaranteed to be PSD** unless additional structural assumptions (e.g., $B \succeq A$) are imposed which does not hold in our setting. However, this can be easily solved. The second statement holds if the matrix H is symmetric and positive semidefinite. We can rewrite H up to a positive scaling factor as the regrouped form:
>
> $$
> H := \mathbb{E}_T\left[\frac{1}{\sigma(X,\beta)} VV^\top\right] - \mathbb{E}_T[VV^\top] \left( \mathbb{E}_T[\sigma(X,\beta) VV^\top] \right)^{-1} \mathbb{E}_T[VV^\top].
> $$
>
> Let us define:
>
> * $A := \mathbb{E}_T[VV^\top]$,
> * $B := \mathbb{E}_T[\sigma(X,\beta) VV^\top]$,
> * $C := \mathbb{E}_T\left[\frac{1}{\sigma(X,\beta)} VV^\top\right]$.
>
> Then we can write:
>
> $$
> H = C - A B^{-1} A.
> $$
>
> This expression is PSD **if and only if** the following block matrix is PSD (by the Schur complement condition):
>
> $$
> M := \begin{bmatrix}
> C & A \\\\
> A & B
> \end{bmatrix}.
> $$
>
> Now observe that this matrix can be written as an expectation over rank-one PSD matrices:
>
> $$
> M = \mathbb{E}_T[\tilde{V} \tilde{V}^\top],
> $$
>
> where
>
> $$
> \tilde{V}^T := \begin{bmatrix}
> \frac{1}{\sqrt{\sigma(X,\beta)}} V^T  \\
> \sqrt{\sigma(X,\beta)} V^T
> \end{bmatrix} \in \mathbb{R}^{2d}.
> $$
>
> Since each outer product $\tilde{V} \tilde{V}^\top$ is PSD and expectations preserve positive semi-definiteness, it follows that $M \succeq 0$. By the Schur complement identity, this implies that $H = C - A B^{-1} A \succeq 0$ as well.
>
> We have updated the manuscript to reflect this corrected structure and clarify the PSD argument. This correction ensures the validity of the key step used in Proposition 2.
>
> ---
>
> ## Clarification on Interpretation of Real-World Results
>
> We agree that the phrase “little to no impact” was unclear. We have revised the sentence as follows:
> >*All estimators (except one) indicate a positive treatment effect; however, the confidence intervals are wide and include the null value (zero or one), preventing any definitive conclusions about the treatment’s effectiveness.*
>
> This revised phrasing better captures the uncertainty in the estimates and avoids overinterpreting the results. We thank the reviewer for pointing this out.
>
> ---
>
> We thank the reviewer again for their feedback. We have addressed all raised concerns, including the clarification of Lemma 2, improved interpretation of the real-world results, and the addition of practical guidance and examples to support assumption selection. We believe these revisions substantially strengthen both the theoretical clarity and applied usability of the paper.
>
>
> ---
>
> ### References
>
> * Hines, O., Diaz-Ordaz, K., & Vansteelandt, S. (2022). *Variable importance measures for heterogeneous causal effects*. arXiv preprint arXiv:2204.06030.
> * Benard, C., & Josse, J. (2023). *Variable importance for causal forests: breaking down the heterogeneity of treatment effects*. arXiv preprint arXiv:2308.03369.
> * Paillard, J., Reyero Lobo, A. D., Kolodyazhniy, V., Thirion, B., & Engemann, D. A. (2025). *Measuring variable importance in heterogeneous treatment effects with confidence*. ICML 2025.
> * Robertson, S. E., Steingrimsson, J. A., Joyce, N. R., Stuart, E. A., & Dahabreh, I. J. (2021). *Center-specific causal inference with multicenter trials: Reinterpreting trial evidence in the context of each participating center*. arXiv preprint arXiv:2104.05905.

---

> > ### Comment · Area_Chair_9pc5 · 2025-08-02
> >
> > @reviewer: Please take a look at the authors’ rebuttal and the other reviews, and let us know your thoughts. In particular, I am curious how your concern on lemma 2 has been dealt with from your perspective.

---

> > > ### Author Response · Authors · 2025-08-02
> > >
> > > Due to space limitations, we previously presented a condensed version of the proof that may have been difficult to follow. Here is a more detailed explanation.
> > >
> > > ---
> > >
> > > We aim to show that the matrix $H$ is positive semi-definite (PSD). The matrix is defined as:
> > >
> > > $$
> > > H = \frac{1}{1-\alpha} \left( \mathbb{E}_T\left[\frac{1}{\sigma(X,\beta)} VV^\top\right] - \mathbb{E}_T[VV^\top] \left( \mathbb{E}_T[\sigma(X,\beta) VV^\top] \right)^{-1} \mathbb{E}_T[VV^\top] \right).
> > > $$
> > >
> > > This can be rewritten more compactly as:
> > >
> > > $$
> > > H = \frac{1}{1-\alpha} (C - A B^{-1} A),
> > > $$
> > >
> > > where we define:
> > >
> > > * $C = \mathbb{E}_T\left[\frac{1}{\sigma(X, \beta)} VV^\top\right]$,
> > > * $A = \mathbb{E}_T[VV^\top]$,
> > > * $B = \mathbb{E}_T[\sigma(X, \beta) VV^\top]$.
> > >
> > > First, we define the vector:
> > >
> > > \\[
> > > \tilde{V} :=
> > > \begin{bmatrix}
> > > \sqrt{\sigma(X, \beta)} V \\\\
> > > \frac{1}{\sqrt{\sigma(X, \beta)}} V
> > > \end{bmatrix}
> > > \in \mathbb{R}^{2d}.
> > > \\]
> > >
> > > (Note that here we changed the definition of $\tilde{V}$ to simplify later steps.)
> > >
> > > Then the outer product $\tilde{V} \tilde{V}^\top$ is:
> > >
> > > \\[
> > > \tilde{V} \tilde{V}^\top =
> > > \begin{bmatrix}
> > > \sigma(X, \beta) VV^\top & VV^\top \\\\
> > > VV^\top & \frac{1}{\sigma(X, \beta)} VV^\top
> > > \end{bmatrix}.
> > > \\]
> > >
> > > Taking expectation, we define the matrix:
> > >
> > > \\[
> > > M := \mathbb{E}_T[\tilde{V} \tilde{V}^\top] =
> > > \begin{bmatrix}
> > > B & A \\\\
> > > A^\top & C
> > > \end{bmatrix}.
> > > \\]
> > >
> > > We now show that $M$ is PSD. For any $z \in \mathbb{R}^{2d}$, we have:
> > >
> > > $$
> > > z^\top M z = \mathbb{E}_T \left[ z^\top \tilde{V} \tilde{V}^\top z \right] = \mathbb{E}_T \left[ (z^\top \tilde{V})^2 \right] \geq 0,
> > > $$
> > >
> > > since each term in the expectation is a square. Therefore, $M \succeq 0$.
> > >
> > > By the **Schur Complement Lemma** for block matrices (see Theorem 1.12 in \[1]), provided that $B$ is invertible and square and $B \succeq 0$ and $M \succeq 0$, we get that
> > >
> > > $$
> > > H = \frac{1}{1-\alpha}(C - A B^{-1} A) \succeq 0.
> > > $$
> > >
> > > This completes the proof.
> > >
> > > ---
> > >
> > > ### Reference
> > >
> > > \[1] F. Zhang, *The Schur Complement and Its Applications*, Vol. 4, Springer Science & Business Media, 2006.

---

> > > > ### Comment · Reviewer_LpQj · 2025-08-04
> > > >
> > > > Thanks for the detailed rebuttal. It addresses most of my concerns and I will increase my score.

---

### Decision · Program_Chairs · 2025-09-17

**Decision:**

Accept (poster)

**Comment:**

This paper develops a unified framework for transporting a broad range of first-moment population causal effect measures under covariate shift. Unlike most prior work that focuses on the risk difference, the authors address both absolute and relative effect measures, which are commonly reported in clinical and policy research. The paper presents identification results, develops classical and semiparametric estimators, and supports the framework with simulations and empirical applications.

While the reviewer team did not reach full agreement, the concern regarding Lemma 2 appears to have been adequately resolved during the discussion. The remaining issues (specifically, the need to acknowledge the limitations of the empirical analysis and to release code) can be addressed in the camera-ready version.

Given the strength of the contribution and the practical relevance of the framework, I recommend acceptance.